# RIGNO: A Graph-based Framework For Robust And Accurate Operator Learning For PDEs On Arbitrary Domains

**Sepehr Mousavi**[1]  **Shizheng Wen**[2]  **Levi Lingsch**[2,3]

**Maximilian Herde**[2]  **Bogdan Raonić**[2,3]  **Siddhartha Mishra**[2,3]

[1] Computational Mechanics Group, ETH Zurich, Switzerland
[2] Seminar for Applied Mathematics, ETH Zurich, Switzerland
[3] ETH AI Center, Switzerland

## Abstract

Learning the solution operators of PDEs on arbitrary domains is challenging due to the diversity of possible domain shapes, in addition to the often intricate underlying physics. We propose an end-to-end graph neural network (GNN) based neural operator to learn PDE solution operators from data on point clouds in arbitrary domains. Our multi-scale model maps data between input/output point clouds by passing it through a downsampled regional mesh. The approach includes novel elements aimed at ensuring spatio-temporal resolution invariance. Our model, termed RIGNO, is tested on a challenging suite of benchmarks composed of various time-dependent and steady PDEs defined on a diverse set of domains. We demonstrate that RIGNO is significantly more accurate than neural operator baselines and robustly generalizes to unseen resolutions both in space and in time. Our code is publicly available at github.com/camlab-ethz/rigno.

## 1 Introduction

Partial Differential Equations (PDEs) mathematically model a wide variety of interesting phenomena in physics and engineering, making their study a matter of great scientific importance [9]. Given the paucity of explicit solution formulas, *numerical methods* [36], such as finite difference, finite element and spectral methods constitute the main tools in the quantitative study of PDEs. The computational cost of existing numerical methods or solvers for PDEs can be prohibitively expensive, particularly in the context of problems such as uncertainty quantification, inverse problems, control, and design, where PDE solvers have to be called multiple times [35]. This necessitates the design of fast and accurate *surrogates* for PDE solvers.

In this context, machine learning (ML) algorithms are increasingly being used as surrogates for PDE solvers, see [31] and the references therein. In particular, *operator learning*, i.e., where the task at hand is to learn the underlying *solution operator* of a PDE from data, has emerged as the dominant paradigm in the application of ML techniques for PDEs. A large variety of operator learning algorithms have recently been proposed, with examples being DeepONets [7, 28], Fourier Neural Operator (FNO) [20], Low-rank Neural Operators [22], Convolutional Neural Operators (CNO) [38] and transformer-based operator learning algorithms [6, 34, 12, 13, 1], among many others. The success of these approaches may largely be contributed to their multi-scale learning. For example, the FNO learns frequencies of different scales within the data, CNO learns by down- and up-sampling

39th Conference on Neural Information Processing Systems (NeurIPS 2025).

via convolutions, and transformer-based algorithms often use multi-scale windowing. As a result, these algorithms learn both short- and long-range physical dependencies of the solutions of PDEs.

An overwhelming majority of these operator learning algorithms or *neural operators* [16] assume that the underlying domain is *Cartesian* and is discretized on a regular grid; however, in practice, and particularly in engineering, PDEs are posed on domains with very complex and highly non-Cartesian geometries. Thus, the direct application of many of the aforementioned operator learning algorithms is difficult or impractical. Therefore, we require *algorithms that can learn the solution operators of PDEs on arbitrary domains.* Various approaches have been proposed to extend neural operators to arbitrary domains. To start, methods such as the FNO and the CNO, originally designed for Cartesian domains, can be extended to non-Cartesian domains via domain masking [38, 13]. Another approach for extending FNO to arbitrary domains is to directly evaluate the discrete Fourier transform rather than FFT [24]. Moreover, one can transform the underlying problem to a Cartesian domain either through extensions [15], learned diffeomorphisms as in Geo-FNO [19], or through learned interpolations between non-Cartesian physical domain and a Cartesian latent domain via graph neural operators [21] as in GINO [23] and references therein. Applying attention mechanisms to a downsampled representation, as demonstrated in UPT [1] and Transolver [45], offers another viable approach. A conceptually distinct line of work leverages implicit neural representations [41, 42], which accommodate arbitrary geometries by encoding input and output functions into a shared latent vector that can be continuously queried at any spatial coordinate. The mapping between input and output functions is then learned in this latent space through a separate training stage.

An alternative approach to learn operators for PDEs on arbitrary domains is provided by *graph neural networks* (GNNs) [5]. This is an ideal ML framework for handling relational data such as molecules, networks, or unstructured computational grids. The use of graph neural networks for simulating PDEs has received widespread attention [40, 39, 33, 4, 8], to name a few. Particularly, approaches such as MeshGraphNet [33] and its derivatives [10, 17] continue to demonstrate success in learning physical simulations. Graphs are able to implicitly capture the challenging collocation point distributions of unstructured meshes and arbitrary point clouds, making methods such as MeshGraphNet ideal for the non-Cartesian domain. However, genuine learning of operators, rather than a specific finite-dimensional discretization of them, requires the ability of the underlying algorithm to display some form of *space-time resolution invariance* [16, 2], i.e., the operator learning algorithm should be able to process inputs and outputs on different spatial and temporal resolutions, both during training and testing, without any decrease in its overall accuracy. Existing graph-based approaches such as MeshGraphNet cannot satisfy this key property, and even many neural operators on Cartesian grids do not inherently capture temporal resolution invariance. Moreover, due to the local nature of their graph construction, MeshGraphNets struggle to capture long-range dependencies within the data. Therefore, given the success of multi-scale neural operators such as FNO and CNO, the question naturally arises: *does there exist a complementary, multi-scale, graph-based approach which is capable of learning space-time resolution invariant operators on complicated distributions of input points?*

In this work, we answer this question affirmatively by proposing RIGNO, a *Region Interacting Graph Neural Operator*. RIGNO is an end-to-end graph neural network based on the *encode-process-decode* paradigm [40, 39], relying on an algorithm to construct a multi-scale *regional mesh*. We also incorporate several novel elements which, in combination with the algorithmic developments, push this algorithm to achieve state-of-the-art performance. The contributions of this work are as follows:

- The introduction of a flexible *regional mesh* which simultaneously incorporates resolution invariance and multi-scale feature learning.

- An *edge masking* technique for GNNs in the context of scientific machine learning, which improves resolution invariance, facilitates optimization during training, and can provide model uncertainty estimates at inference.

- The introduction of *temporal fractional pairing*, a fine-tuning strategy for time-continuous neural operators that unlocks generalization to higher time resolutions.

- Extensive experimental results that demonstrate significant improvements to the state-of-the-art across a variety of benchmarks with domain geometries including structured and unstructured meshes, random point clouds, and Cartesian grids.

## 2  Methods

**Problem formulation.**   We consider the following generic time-dependent PDE with $\Omega_t = (0, T)$ the time domain, $\Omega_x \subset \mathbb{R}^d$ the $d$-dimensional spatial domain, $u \in \mathcal{C}(\overline{\Omega}_t; \mathcal{X})$ the solution of the PDE with $\mathcal{X} \subset L^p(\Omega_x; \mathbb{R}^s)$ for some $1 \le p < \infty$, $c \in \mathcal{Q}$ a spatially varying coefficient, $\mathcal{F}$ a differential operator, $\mathcal{B}$ a boundary operator, and $a \in \mathcal{X}$ the initial condition,

$$
\begin{aligned}
\partial_t u &= \mathcal{F}(c, t, u, \nabla_x u, \nabla_x^2 u, \dots), & \forall (t, x) &\in \Omega_t \times \Omega_x, \\
\mathcal{B}u(t, x) &= 0, & \forall (t, x) &\in \Omega_t \times \partial\Omega_x, \\
u(0, x) &= a(x), & \forall x &\in \Omega_x.
\end{aligned}
\tag{1}
$$

The solutions of the PDE (1) are given in terms of the *solution operator* $\mathcal{S}^t : \mathcal{X} \times \mathcal{Q} \times \Omega_t \to \mathcal{X}$, which maps the initial conditions and coefficients to the solution $u(t)$ at any time. With the semi-group property for solutions of time-dependent PDEs [9], we can define an extended solution operator $\mathcal{G}^\dagger : \mathcal{X} \times \mathcal{Q} \times \Omega_t \times \mathbb{R}^+ \to \mathcal{X}$, which maps the solution $u(t)$ at any time $t$ and the coefficient $c$ to the solution $u(t + \tau)$ at a later time $t + \tau$ through the action,

$$
\mathcal{G}^\dagger(u(t), c, t, \tau) = \mathcal{S}^\tau(u(t), c, \tau) = u(t + \tau).
\tag{2}
$$

Our goal here is to learn this extended solution operator $\mathcal{G}^\dagger$ from data. Note that plugging $t = 0, \tau = t, u(0) = a$ yields $\mathcal{S}^t$, recovering the solution operator of the PDE (1). We would also like to point that the (extended) solution operator depends implicitly on the underlying spatial domain $\Omega_x$, which in turn, can change in some operator learning tasks.

**Model architecture.**   As shown in Figure 1, our model is based on the well-known *encode-process-decode* paradigm [40, 39]. All the three components, i.e., encoder, processor, and decoder are graph neural networks (GNNs). The methods presented in this section are implemented using JAX [3] and are available on GitHub[1]. As the aim is to learn the (extended) solution operator $\mathcal{G}^\dagger$ (2), the inputs to our model are, in general, i) a set of coordinates in the domain $\mathbf{x} := \{\mathbf{x}_i \in \Omega_x\}_i$, where $\Omega_x$ is the spatial domain, ii) $\boldsymbol{u} := \{\boldsymbol{u}_i \in \mathbb{R}^s\}_i$, discretized values of a function at known coordinates $\mathbf{x}$, iii ) $\boldsymbol{c} := \{\boldsymbol{c}_i\}_i$, discretized values of the spatial coefficient at $\mathbf{x}$ coordinates, iv) the time $t$ and v) the *lead time* $\tau$. The model's output $\boldsymbol{r} := \{\boldsymbol{r}_i \in \mathbb{R}^s\}_i$ represent discretized values of a parameterized map at the coordinates $\mathbf{x}$ and is obtained by $\boldsymbol{r} = \mathcal{T}_\theta(\boldsymbol{u}, \mathbf{x}, \boldsymbol{c}, t, \tau)$.

**Underlying graphs.**   The map $\mathcal{T}_\theta$ is a GNN and will process data that are defined on graphs. As shown in Figure 1 (for an idealized two-dimensional domain) and following ideas in [17], we consider two underlying sets of nodes. The first set of nodes corresponds to the unstructured point cloud on which the input (and output) data is defined. We call this point cloud as the *physical mesh* and the nodes that constitute it as *physical nodes*. We do not perform any explicit information processing on these physical nodes. Most of our processing will take place in another bespoken set of nodes that we call as *regional nodes* and the point cloud constituted by them as the *regional mesh*. Here, the regional nodes are constructed by randomly sub-sampling the set of physical nodes with a sampling ratio which is a hyperparameter of our model. A graph is constructed to connect the regional nodes (see Figure 1) via bidirectional *regional mesh edges*. Moreover, we need to pass information between the physical nodes and the regional nodes, necessitating the construction of graphs between them. To this end, we connect physical nodes to regional nodes (Figure 1) by unidirectional edges. These edges are constructed such that each regional node constitutes a circular *sub-region* containing a subset of nearby physical nodes and serves to accumulate information from them. Based on the local density of regional nodes, a minimum support radius is calculated for each of them such that their union covers the whole domain. Information from the regional mesh can be transmitted to the physical mesh in a similar way by constructing a graph with unidirectional edges. As shown in Figure 1, these edges are constructed to accumulate information from multiple regional nodes into one physical node. Thus, construction of these unidirectional edges provides a very flexible upscaling and downscaling strategy for our model.

Needless to say, when possible, the graph construction must be adapted to the boundary conditions of the PDE (1). Once the underlying graphs have been constructed, we must also provide geometric information to the GNNs. We normalize the given coordinates and prepare them for all nodes and all

---

[1]`https://github.com/camlab-ethz/rigno`

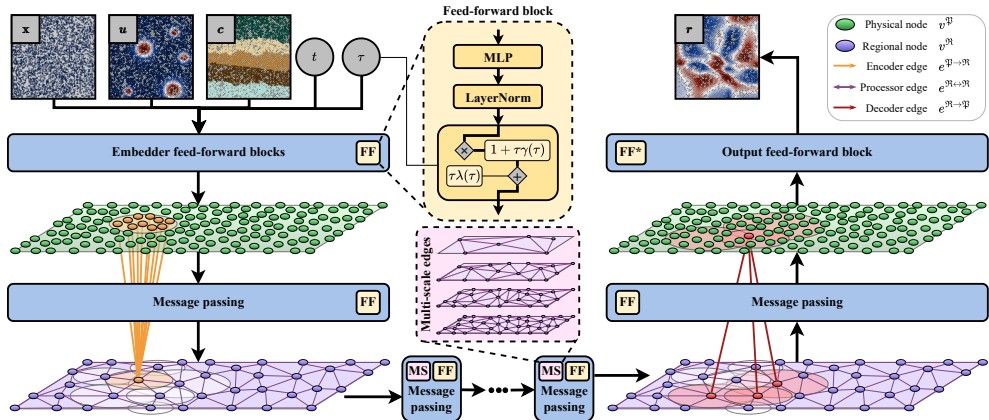

Figure 1: General schematic of the RIGNO architecture for an idealized two-dimensional domain. The inputs are first independently projected to a latent space by feed-forward blocks. The information on the original discretization (*physical nodes*) is then locally aggregated to a coarser discretization (*regional nodes*). Regional nodes are connected to each other by edges with multiple length scales. Several message-passing steps are then applied on the regional nodes which constitute the processor. The processed features are then transmitted back to the original discretization by using similar edges as in the encoder, before being independently projected back to the desired output dimension via a feed-forward block without normalization layers.

edges accordingly. We postpone the rather technical description of graph construction, geometric features, and incorporation of boundary conditions, particularly the novel additions to the model design to enforce periodic boundary conditions, to **SM** Section A. Once the inputs, graphs and geometric information are available, the first step is to lift the initial features of all nodes and all edges to suitable high-dimensional latent spaces. To this end, feed-forward blocks are applied independently on the initial features of each node and edge. Each feed-forward block consists of a shallow multi-layer perceptron (MLP) followed by normalization layers (see Figure 1). Next, our model processes this high-dimensional latent information by employing the following constituent GNNs.

**Main components.**    The encoder is designed to transmit information from the underlying physical mesh (representing a discretization of the domain $\Omega_x$) to the coarser regional mesh. This is done by performing a single message passing step with residual updates. The processor then operates on the latent information defined over the regional mesh by performing $P$ sequential message passing steps. We incorporate residual updates and skip connections to maintain gradient flow and promote stable training for large $P$. The decoder serves to transfer processed information from the regional mesh back to the physical mesh. The first step in the decoder is a single message-passing block. In the second step of the decoder, the physical node feature vectors are independently scaled down to the desired dimension to form the output $r$. Summarizing, as our model involves the learned interaction between nodes that each represent aggregated information from an underlying sub-region, we call it as Region Interaction Graph Neural Operator or RIGNO. More details about the GNN components are provided in **SM B**.

**Edge masking.**    Considering the graph structure in the encoder as an example, each regional node aggregates information from multiple physical nodes. For high-resolution inputs, many of these physical nodes can be redundant, as the underlying function can often be accurately reconstructed (e.g., through interpolation) using a smaller subset of nodes. During training, the model may implicitly disregard some of these redundant nodes and instead focus on a subset of dominant or central connections. However, at test time, when the input resolution differs, these dominant nodes may no longer exist, resulting in degraded performance. We identify this phenomenon as a primary cause of the resolution sensitivity commonly observed in GNN-based operator learning models. To address this, we use random masking of edges in all message-passing blocks of RIGNO. Before each message passing step (e.g., step $p$ of the processor), each edge is masked with a given probability and does not participate in that specific message-passing block. In the next message passing step

(e.g., step $p + 1$ of the processor), a different set of edges is randomly masked. Through random masking of the edges during training, the model cannot consistently rely on the same high-correlation connections. In the absence of these edges, the model is forced to use the weaker correlations. Additionally, learning from weaker or secondary correlations in the processor enables the model to better integrate information across multiple spatial scales, rather than overfitting to dominant patterns in the training resolution. Although loosely inspired by dropout [43], edge masking is conceptually very different from it. While dropout masks trainable parameters of MLPs, edge masking disables pairwise interaction of spatial entities (nodes). This allows the model to discover weaker spatial correlations in the data. The set of the masked edges randomly changes in every epoch, which allows a single message-passing block to learn from all the edges during training. Since the messages computed from the neighbors are averaged, there is no need for any further configuration at training or inference. It is also possible to use a different masking probability or to completely disable it at any point during training or inference. As shown in the next section, edge masking enables us to i) make RIGNO (approximately) resolution invariant, ii) obtain significantly lower errors, iii) reduce computational complexity by sparsely sampling edges during each step, and iv) provide a method for quantifying uncertainty in the predictions of RIGNO.

**Temporal processing.** We recall that our goal is to approximate the extended solution operator $\mathcal{G}^\dagger(u(t), c, t, \tau)$ (2), which is a function of the input time $t$ and the lead time $\tau$. Both of these scalars are provided to the encoder of RIGNO as inputs and embedded into suitable high-dimensional latent spaces. However, merely providing lead time as an input may not be sufficient to learn the solution operator in deep models. Following [13], we introduce lead time as an additional *conditioning* in each hidden layer of RIGNO (see Figure 1). We do this by adapting the *normalization layers* employed within our model. Let $\mathbf{y}$ be an output vector for any layer and normalize it by an appropriate normalization layer (e.g., *BatchNorm*, *LayerNorm*) to obtain the normalized output $\widetilde{\mathbf{y}}$. We further condition $\widetilde{\mathbf{y}}$ with the lead time $\tau$ by setting $\hat{\mathbf{y}} = (1 + \tau\gamma(\tau)) \odot \widetilde{\mathbf{y}} + \tau\lambda(\tau)$, with $\gamma, \lambda$ being small learnable MLPs.

Finally, we need to provide a time marching strategy to update our model in time. In this context, RIGNO can be employed in a very general time marching mode by setting,

$$\mathcal{G}_\theta(u^t, c^t, t, \tau) = \alpha u^t + \beta \mathcal{T}_\theta(u^t, c^t, t, \tau), \tag{3}$$

where $\alpha, \beta$ are scalar parameters, $\mathcal{T}_\theta$ is the output of RIGNO, and $\mathcal{G}_\theta \approx \mathcal{G}^\dagger$ is the final approximation to the extended solution operator (2) of the PDE (1). Setting $\alpha = 0$ and $\beta = 1$ directly recovers an approximation to the *output* of the extended solution operator. In this case, RIGNO can be interpreted to directly provide the solution of the PDE (1) at time $t + \tau$, given the solution at time $t$, coefficient $c$ and the lead time $\tau$ as inputs. Similarly, setting $\alpha = \beta = 1$ yields the *residual* of the solution at the later time, given the solution at current time. On the other hand, setting $\alpha = 1$ and $\beta = \tau$ leads to the model output $\mathcal{T}_\theta$ approximating the *time derivative* of the solution operator as (3) is simply an approximation of the Taylor expansion in time of the solution operator, truncated at leading order. A clear advantage of this time-derivative update is that it automatically enforces consistency for $\tau = 0$. We will explore these choices for time updates in our numerical experiments to determine which ones are the most suitable.

**Loss functions and training strategy.** For simplicity of exposition, here, we fix a final time $T$ and consider a fixed discretization $\Omega_t = \cup_{n=1}^N [t_{n-1}, t_n]$. We assume that training samples are provided as trajectories $\{u_m(t_n)\}_{n=0}^N$, for $1 \le m \le M$, with each $u_m$ being the solution to the PDE (1) with initial condition $a_m$ and coefficient $c_m$. Given these training trajectories and the fact that our model can be evaluated at any time, there are multiple ways in which the RIGNO model, denoted here by $\mathcal{G}_\theta$ can be trained. We follow the recently proposed *all2all* training strategy [13] as training input-output pairs scale quadratically with respect to the number of time steps for time-dependent problems. Hence, the goal of training is to find parameters $\theta \in \Theta$ such that the following loss function can be be minimized,

$$\mathcal{J} := \frac{1}{M\widehat{N}} \sum_{m,n,t_k \le t_n} \mathcal{L}\left[u_m(t_n), \widehat{u}_m^{t_k}(t_n)\right], \quad \widehat{u}_m^{t_k}(t_n) = \mathcal{G}_\theta\left(u_m(t_k), c_m, \ t_k, \ t_n - t_k\right), \tag{4}$$

with $\widehat{N} = \frac{1}{2}(N+1)(N+2)$ and $\mathcal{L}$ being a measure of the mismatch between the ground truth and the model prediction, for instance either the mean absolute or the mean squared error.

A problem with this approach is that the smallest prediction lead time $\tau_{min}$ in the training set is given by the minimum of $t_n - t_{n-1}$ for all $n$. Hence, at inference, predicting for lead times $\tau < \tau_{min}$ could lead to distribution shifts as smaller lead times have not been seen during training. Yet, a fundamental requirement for a time-dependent neural operator is to predict arbitrarily small times correctly in order to ensure *time continuity*. We tackle this problem by proposing to further *fine-tune* a RIGNO model, trained with the loss function (4), by using a modified loss function where $\widehat{u}_m^{t_k}(t_n)$ in (4) is obtained by starting from an intermediate input function: $\mathcal{G}_\theta(\widetilde{u}_m(t_\star), c_m, t_\star, t_n - t_\star)$. Thus, our aim is to use solutions at intermediate time steps, which are not available in the training set, to make predictions at later time steps, available in the training data. However, we must find an approximation $\widetilde{u}_m(t_\star) \approx u_m(t_\star)$ as $u_m(t_\star)$ is not available during training. Our solution is to use the pretrained RIGNO itself by setting $\widetilde{u}(t_\star) = \mathcal{G}_\theta(u(t_k), c, t_k, t_\star - t_k)$. This self-consistent approach is reasonable as the modified loss function is only used during fine-tuning. Recycling model outputs as inputs has been employed in previous implementations for the fixed-step setting [4, 20] to improve rollout stability. Here, we extend this concept to the time-continuous setting, aiming to generalize to arbitrarily small time steps. As shown in **SM** Figure C.3, the flow of the gradient can be stopped at the intermediate time $t_\star$ for training efficiency. The utility of this *fractional pairing* strategy will be tested subsequently.

Given that the output time $t$ and lead time $\tau$ are both inputs to the trained RIGNO, a variety of approaches can be used for temporal processing at inference. As in [13], to obtain the solution $u(t)$ at time $t$, from an initial datum $u(0) = a$, we can either directly apply RIGNO with lead time $\tau = t$ or we can employ different autoregressive rollout strategies as described in the **SM** Section C.2.

## 3   Results

In this section, we demonstrate the performance of RIGNO on a challenging suite of numerical experiments by comparing it with well-established baselines. We also highlight the factors that underpin the observed performance of RIGNO.

**RIGNO accurately learns the solution operators of PDEs on arbitrary domains.**   To test the performance of RIGNO for PDEs on arbitrary domains, we choose two types of datasets. First, we consider 4 datasets on point clouds obtained from structured and unstructured meshes, namely Heat-L-Sines (heat equation on a L-shaped domain), Wave-C-Sines (wave equation, propagation of waves in a homogeneous medium of circular shape), AF (compressible transonic flow past airfoils taken from [19]), and Elasticity (hyper-elastic deformations of varying domain shapes, also taken from [19]). Next, we test how well RIGNO learns operators defined on random point clouds. To set this up, we choose 9 datasets from the *PDEGym* database [13], which deal with four different PDEs; the incompressible Navier-Stokes equations, the compressible Euler equations, the Allen-Cahn equation, the wave equation on a spatially varying medium, and the Poisson equation. The datasets are all originally defined on uniform Cartesian grids. To obtain a random point cloud, we randomly subsample 9216 points from the available coordinates on a $128 \times 128$ uniform grid, see **SM** Figure D.1. Our dataset collection spans 8 PDEs and comprises 3 time-independent and 10 time-dependent problems. See **SM** Section D for more details about the datasets and their abbreviations used here.

In Table 1, we present the test errors on these 13 *unstructured* datasets [32] with two sizes of RIGNO, namely RIGNO-12 and RIGNO-18, with 12 and 18 layers in their processor modules, respectively. To contextualize the results, we consider five baselines, namely MeshGraphNet [33], Geo-FNO [19], FNO DSE [24], GINO [23], and UPT [33]. All of these baselines are capable of handling inputs and outputs on arbitrary point clouds, yet use different routes to achieve this. More details about the baselines, including the selection of hyperparameters, are outlined in **SM** Section G. We observe from Table 1 that RIGNO-18 outperforms all the baselines in all 13 datasets. Likewise, RIGNO-12 is the second best-performing model for 10 of the 13 datasets. For Wave-C-Sines, Poisson-Gauss, and Elasticity, other architectures claim second best; yet, RIGNO-12 remains comparable in performance to RIGNO-18. Moreover, in some cases, the RIGNO variants are almost an order of magnitude more accurate than the baselines, showing excellent performance on these benchmarks.

Often, models are designed to have flexible inputs/outputs by potentially reducing their performance on Cartesian domains. To further illustrate the expressive power of RIGNO, we consider 9 datasets from the *PDEGym* database on their original Cartesian grids. In this case, we can compare RIGNO with state-of-the-art neural operators such as FNO, CNO, and scOT, which are tailored for Cartesian

Table 1: Benchmarks on datasets with unstructured and Cartesian-grid space discretizations. **Lowest** and **second lowest** errors are highlighted. The errors for time-dependent problems are the lowest among three autoregressive strategies (see **SM** Section C.2). All training datasets have 1024 trajectories/samples except for the AF dataset (2048). MGN corresponds to the MeshGraphNet baseline.

| Dataset | Median relative $L^1$ error [%] | | | | | | |
|---|---|---|---|---|---|---|---|
| (point cloud) | RIGNO-18 | RIGNO-12 | MGN | Geo-FNO | FNO DSE | GINO | UPT |
| Heat-L-Sines | 0.04 | 0.05 | 0.51 | 0.15 | 0.53 | 0.19 | 20.2 |
| Wave-C-Sines | 5.35 | 6.25 | 17.3 | 13.1 | 5.52 | 5.82 | 12.7 |
| NS-Gauss | 2.29 | 3.80 | 76.4 | 41.1 | 38.4 | 13.1 | 92.5 |
| NS-PwC | 1.58 | 2.03 | 33.1 | 26.0 | 56.7 | 5.85 | 100 |
| NS-SL | 1.28 | 1.91 | 18.8 | 24.3 | 29.6 | 4.48 | 51.5 |
| NS-SVS | 0.56 | 0.73 | 7.53 | 9.75 | 26.0 | 1.19 | 4.20 |
| CE-Gauss | 6.90 | 7.44 | 76.4 | 42.1 | 30.8 | 25.1 | 64.2 |
| CE-RP | 3.98 | 4.92 | 13.9 | 18.4 | 27.7 | 12.3 | 26.8 |
| ACE | 0.01 | 0.01 | 0.09 | 1.09 | 1.29 | 3.33 | 100 |
| Wave-Layer | 6.77 | 9.01 | 84.8 | 11.1 | 28.3 | 19.2 | 19.6 |
| Poisson-Gauss | 2.26 | 2.52 | 30.9 | 8.16 | 2.27 | 7.57 | 48.4 |
| AF | 1.00 | 1.09 | 10.1 | 4.48 | 1.99 | 2.00 | 45.7 |
| Elasticity | 4.31 | 4.63 | 11.9 | 5.53 | 4.81 | 4.38 | 12.6 |

Table 2: Benchmarks on datasets with Cartesian-grid space discretizations. **Lowest** and **second lowest** errors are highlighted. The errors for time-dependent problems are the lowest among three autoregressive strategies (see **SM** Section C.2). All training datasets have 1024 trajectories/samples.

| Dataset | Median relative $L^1$ error [%] | | | | |
|---|---|---|---|---|---|
| (Cartesian grid) | RIGNO-18 | RIGNO-12 | CNO | scOT | FNO |
| NS-Gauss | 2.74 | 3.78 | 10.9 | 2.92 | 14.2 |
| NS-PwC | 1.12 | 1.82 | 5.03 | 7.11 | 11.2 |
| NS-SL | 1.13 | 1.82 | 2.12 | 2.49 | 2.08 |
| NS-SVS | 0.56 | 0.75 | 0.70 | 0.99 | 6.21 |
| CE-Gauss | 5.47 | 7.56 | 22.0 | 9.44 | 28.7 |
| CE-RP | 3.49 | 4.43 | 18.4 | 9.74 | 31.2 |
| ACE | 0.01 | 0.01 | 0.28 | 0.21 | 0.60 |
| Wave-Layer | 6.75 | 8.97 | 8.28 | 13.4 | 28.0 |
| Poisson-Gauss | 1.80 | 2.44 | 2.04 | 0.68 | 11.5 |

grids. The resulting test errors in Table 2 show that RIGNO is very accurate on Cartesian grids too and RIGNO-18 is the best performing model on 8 out of 9 datasets, often with several times lower errors than the baselines. We would also like to emphasize that RIGNO is the only framework that displays acceptably low errors for every single dataset that we consider here, demonstrating its unprecedented robustness.

**RIGNO scales with data and model size.** An essential requirement for modern ML models is their ability to *scale with data*, i.e., the test error should reduce at a consistent rate as the number of training samples increases [14, 18, 13]. To test this aspect, we plot the test error with respect to the number of training samples in Figure 2a, observing that RIGNO-12 shows a consistent scaling for the datasets. Another critical aspect of ML models is their ability to *scale with model size* [14, 13]. Compared to data scaling, scaling with model size is more subtle as a number of hyperparameters influence model size and changing each of them, while keeping the others fixed, might lead to different outcomes. In **SM** Section F.3, we present a detailed discussion and results on how RIGNO scales with respect to each of the hyperparameters that influences model size. Here, in Figure 2b, we only consider two (correlated) indications of model size, i.e., total number of model parameters and *inference time*. We see from Figure 2b that RIGNO-12 consistently scales with respect to both the total number of parameters and inference run time.

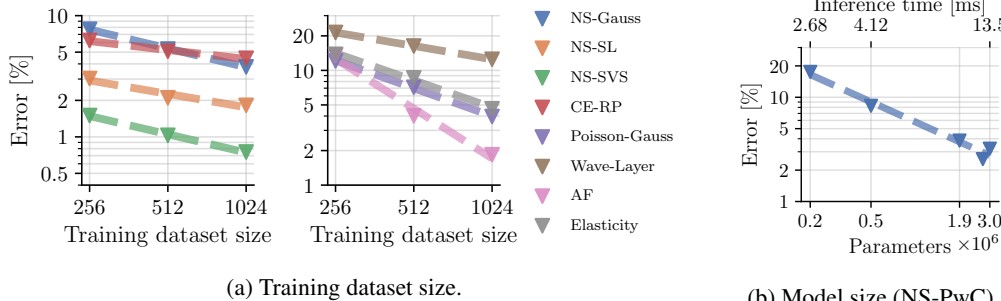

(a) Training dataset size.

(b) Model size (NS-PwC).

Figure 2: Scaling behavior of RIGNO with respect to dataset and model size. All values correspond to autoregressive (`AR-2`, see **SM** Section C.2) test errors at $t = t_{14}$.

**Resolution invariance and RIGNO.** As mentioned before, a key requirement for a robust neural operator is to enforce an appropriate form of resolution invariance [2]. To test the abilities of RIGNO in this regard, we consider the NS-PwC dataset on random point clouds. The training resolution for RIGNO is $42^2$. At inference, we test it on this and several other resolutions, presenting the test errors in Figure 3a. We see from this figure that the super-resolution test errors with the base version of RIGNO are very small and indicate that the proposed local aggregation scheme enables RIGNO to generalize well across higher spatial resolutions. The results of the same test on MeshGraphNet (see **SM** Section F.4) provide further evidence that local aggregations are essential for obtaining super-resolution invariance. Furthermore, the proposed *edge masking* technique allows us to enforce resolution invariance to a greater extent, particularly in the sub-resolution regime. Moreover, we observe that edge masking substantially improves the optimization process by allowing the model to learn weaker spatial correlations, while simultaneously cutting the computational cost by a factor of 2 when $50\%$ of the edges are masked.

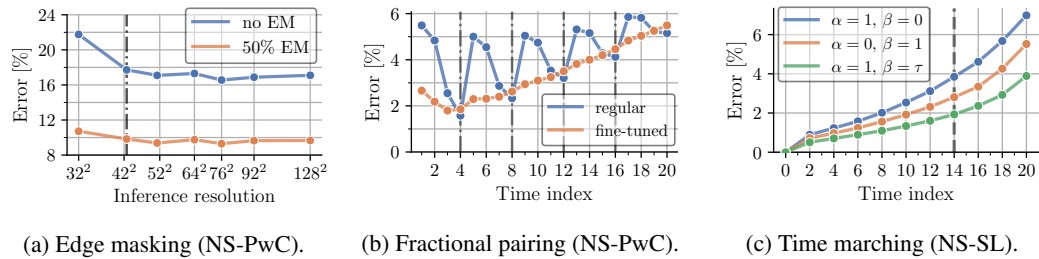

(a) Edge masking (NS-PwC).

(b) Fractional pairing (NS-PwC).

(c) Time marching (NS-SL).

Figure 3: Ablation of the proposed training strategies. a) The effect of edge masking on resolution invariance. EM stands for edge masking. The y-axis shows the relative test error at $t = t_{14}$ when the spatial resolution of the inputs and outputs varies from the training resolution $64^2$. b) Autoregressive test errors with and without fractional pairing fine-tuning. The initial solution (model input) is updated on vertical lines according to the `AR-4` scheme (see **SM** Section C.2). These models are exceptionally trained on snapshots with time resolution $4\Delta t$ and up to time $t_{16}$. c) Autoregressive (`AR-2`) test errors with different time marching strategies. Time snapshots after the vertical line has not been seen during training and are considered as extrapolation in time.

**On temporal processing with RIGNO.** RIGNO can learn the solution operators of time-dependent PDEs and contains several elements in its design to enforce time continuity and enable accurate temporal approximation. To test these aspects, we focus on the NS-PwC and the NS-SL datasets. The underlying trajectories are available from the *PDEgym* collection [13] on a time interval of $[0, 1]$, sampled at equidistant times $t_n = n\Delta t$, with $\Delta t = 0.05$ and $0 \leq n \leq 20$. We trained RIGNO with data sampled at times $\{t_0, t_4, t_8, t_{12}, t_{16}\}$. The smallest time difference present in the training dataset is thus $\tau = 4\Delta t$. During inference, we *unroll* RIGNO using time steps $\tau \in \{\Delta t, 2\Delta t, 3\Delta t, 4\Delta t\}$ according to the `AR-4` scheme described in **SM** Section C.2. We compare the test errors at all possible time indices $1 \leq n \leq 20$. Hence, we test the generalization capability of RIGNO to a higher time resolution. The resulting test errors are shown in Figure 3b. With regular training, the errors with unseen time steps $\tau < 4\Delta t$ are significantly higher than with $\tau = 4\Delta t$, leading to the

observed jagged pattern in the plot of the error over time. The proposed *fractional pairing* fine-tuning approach effectively enhances the accuracy with small unseen time steps, leading to a monotone error accumulation with $t \geq t_4$, where the output time is in-distribution.

We also consider time marching strategies arising from the choice of parameters $\alpha, \beta$ in (3). Here, we train and infer the model using the same time resolution ($2\Delta t$). We present the results with all the three choices of time marching strategies, namely output, residual and derivative marching for the NS-SL dataset in Figure 3c. We observe a slow error growth and very good extrapolation beyond the times seen during training ($t > t_{14}$). Moreover, we see that the derivative time marching strategy with $\alpha = 1, \beta = \tau$ is clearly superior in this example to the other two, leading us to prefer it during our evaluations.

**Quantifying model uncertainties.** Not only does edge masking enable RIGNO to be (approximately) resolution invariant, but it also provides a simple method to quantify model uncertainties. As edge masking is random, invoking it during inference enables the model to provide multiple outputs for the same input. The mean of the outputs can be used as the model prediction as shown in **SM** Figure F.4. The same figure shows how the standard deviation of this ensemble of outputs is highly correlated with the test error. Thus, it can be used to provide an *adaptive time marching strategy* to reduce errors at inference, see **SM** Section F.6 for details. In **SM** Section F.7, we show that edge masking makes RIGNO more robust to noisy inputs as well.

## 4 Discussion

In this paper, we present RIGNO, an end-to-end GNN-based neural operator for learning the solution operators of PDEs on arbitrary domains. Our model follows an encode-process-decode paradigm [40]. Given a point cloud on which the input data is defined (the physical mesh), the encoder is a GNN that maps the input data into a latent space defined on a regional mesh, which can be interpreted as a downsampled version of the underlying point cloud. The input point cloud can be very general, as no relational structure or graph must be specified on it. The processor maps the latent variables on the regional mesh into learned representations through multi-scale processing, implemented through multiple stacked message-passing neural networks. Finally, the decoder transforms the processed variables from the regional mesh to physical variables on the output point cloud. Again, the only graph in this layer connects the regional mesh nodes to the points in the output cloud. No relational structure or graph needs to be defined on the output point cloud, making RIGNO very general in terms of its abilities to handle data on arbitrary point clouds.

In addition to this basic construction (see Figure 1), RIGNO introduces several novel elements to satisfy two important requirements. First, as desired in [16, 2], (approximate) resolution invariance is essential for structure-preserving operator learning. RIGNO uses overlaps between the sub-regions in the regional mesh connecting points in the physical mesh and more importantly, edge masking, to enable approximate resolution invariance, a feature which is hard to achieve for neural operators on arbitrary domains. Second, RIGNO is tailored to process time-dependent PDEs. In this context, temporal continuity and control on error growth over time are key properties that the neural operator needs to uphold. Here, we have incorporated many elements in RIGNO to achieve these requirements, namely lead time conditioning, all2all training, and a novel fractional pairing fine-tuning strategy. This allows us to maximize training steps for a given number of trajectories, while allowing for general rollout strategies with arbitrarily small time steps at inference.

We have tested RIGNO on a suite of challenging problems including PDEs with data on structured/unstructured meshes, random point clouds, and Cartesian grids. RIGNO is shown to be significantly more accurate than baselines designed for such arbitrary domains. We also observe that RIGNO is more accurate than state-of-the-art neural operators such as CNO, FNO and scOT, even for PDEs on Cartesian grids. The gains in accuracy are not marginal, and can be of an order of magnitude on many problems, demonstrating the expressive power of the proposed architecture. We attribute RIGNO's state-of-the-art accuracy and robustness primarily to the high expressive power of local message passing, which enables fine-grained manipulation and propagation of information. This advantage is especially prominent in time-dependent and advection-dominated problems. For elliptic problems, where global interactions are more crucial, the gains are less pronounced. To mitigate the limitations of purely local interactions, RIGNO incorporates multi-scale edges in the processor, introducing long-range interactions that significantly improve accuracy without incurring notable

additional computational cost. Furthermore, we show that RIGNO scales with both model and data size, while having excellent resolution invariance and temporal continuity properties. It generalizes very well to interpolated and extrapolated time levels, and maintains its accuracy with very small time steps.

**Related work.** The popular MeshGraphNet [33] served as a foundation for the formulation of RIGNO and thus admits basic architectural similarities. This approach constructs a graph on the input coordinates based on a mesh and performs message passing steps directly on the physical coordinates. Yet, as confirmed by our empirical results (see **SM** Section F.4), this approach leads to a strong dependence on the space resolution of the training data and cannot generalize to a different resolution. Furthermore, the computational cost quickly becomes prohibitive as the number of mesh (input) points increases, hindering MeshGraphNet from scaling to larger problems. This issue is addressed in [10] by downsampling the initial mesh to a coarser mesh and performing more message passing steps on the coarse mesh. However, this approach still relies on the original mesh for resolving small scales and consequently inherits the dependence on the training resolution from MeshGraphNet. As evidenced by our experimental results, the multi-scale regional mesh construction of RIGNO mitigates the reliance of such graph-based approaches on the distribution of the input points and introduces the ability to scale to larger problem sizes (see Table F.2). Radius-based aggregation for projection onto a coarse grid has been explored in prior work and shown to be effective for non-graph-based architectures. GINO [23] performs local aggregation of coordinates within a fixed radius to map between an unstructured point cloud and a uniform mesh, enabling the use of FNO as the core processor. UPT [1] employs transformers as the core processor, and thus does not require the mapped coordinates to lie on a uniform grid. In this case, local aggregation is optionally applied in the encoder to compress the information from dense point clouds to a coarser representation. Both these methods rely on a global fixed radius which is a hyperparameter and must be calibrated for each dataset. In practice, the given point clouds are often more refined in regions with sharper gradients, making a fixed-radius approach destined to produce an uneven number of edges per receiver node. To address this, additional mechanisms such as limiting the number of incoming edges per node are necessary. By contrast, the adaptive strategy used in the current work determines aggregation radii based on local resolution, thereby mitigating both these issues while automatically ensuring full domain coverage without the need for manual calibration.

**Limitations.** This work focuses on PDEs in two spatial dimensions. However, many of the considered datasets are comparable in complexity, grid resolution, and number of input-output pairs to typical 3D problems. In principle, RIGNO is readily extendable to PDEs in three dimensions, though the current paucity of publicly available 3D benchmarks limits our ability to explore this important direction. Furthermore, our focus here was on presenting the model and extensive empirical investigation. Important theoretical questions such as those of universal approximation of PDE solution operators with RIGNO will be considered in future work. Similarly, we have empirically demonstrated that RIGNO scales with the number of training samples; however, proving a rigorous generalization error bound, such as that of [18], merits further analysis. Moreover, the scaling curves of Figure 2 suggest that a large number of training samples are necessary for neural operators to achieve small error. This issue is common to all standalone neural operators [13] and can be alleviated by building foundation models such as those proposed in [13, 11, 44, 30]. Our results presented in **SM** Section F.9 show that RIGNO can largely benefit from transfer learning. Using RIGNO as the backbone for a foundation model is another avenue for further exploration. Finally, downstream applications of RIGNO to tasks such as uncertainty quantification (UQ), inverse problems and PDE-constrained optimization will be considered in future work.

## Acknowledgements

The contribution of Siddhartha Mishra to this work was supported in part by the DOE SEA-CROGS project (DE-SC-0023191).

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

# Supplementary Material
## RIGNO: A Graph-based Framework for Robust and Accurate Operator Learning for PDEs on Arbitrary Domains

## A  Graph Construction

### A.1  Encoder and decoder edges

The edges between the physical and regional nodes in the encoder and in the decoder are defined based on support neighborhoods around each regional node. We consider a circular neighborhood around each regional node (see Figures 1 and A.1a) and calculate the radii of the circles such that the union of all the circles covers the whole domain. To achieve this, we first construct a Delaunay triangulation on the regional nodes. We then calculate all medians of all the simplices in the triangulation. The minimum radius of a node is then calculated as two-thirds of the longest median that crosses that node. The minimum radius is then multiplied by an *overlap factor*, which is a hyperparameter of the model. A similar method can be applied to three-dimensional point clouds as well. Each regional node is then connected to all the physical nodes that lie inside its corresponding support sub-region. With a larger overlap factor, the regional nodes receive information from more physical nodes in the encoder. Too large overlap factors might be unnecessary since regional nodes are assumed to encode information locally. However, in the decoder, overlap factor plays a more important role. With minimum overlaps, many physical nodes will end up being connected to only a single regional node, which might result in overly biased decoded latent features. Therefore, we generally use different overlap factors in the encoder and in the decoder.

### A.2  Regional nodes

By defining the edge connections between the regional and the physical nodes based on support sub-regions in the encoder and in the decoder, RIGNO allows for unstructured physical and regional meshes. Within this framework, even for an unstructured set of physical nodes, the regional nodes can still follow a structure (e.g., lie on a uniform grid) since the choice of the regional nodes is completely arbitrary. However, constructing an unstructured regional mesh is favorable since local resolutions of the physical mesh can be inherited by the regional mesh. To this end, we always define the regional nodes by randomly sub-sampling the given physical nodes by a factor that is given as a hyperparameter. The set of regional nodes may be different during inference or in each training epoch.

The sub-sampling factor is a fixed hyperparameter, which makes the resolution of the regional mesh dependent on the resolution of the physical mesh. If a model is only trained with a fixed mesh resolution, changing the resolution at inference might affect its accuracy. However, since the choice of the regional mesh nodes is completely arbitrary, the resolution of the regional mesh can be corrected in order to correspond to the resolution of the training regional meshes. For zero-shot super-resolution inference, this can be easily done by sub-sampling the physical mesh with a higher sub-sampling factor. For zero-shot sub-resolution inference, however, a suitable mesh-refinement algorithm must be employed. In this work, we refine the mesh by first constructing a standard triangulation, randomly picking the necessary number of triangles, and adding the centroid of those triangles to the regional mesh. With this method, we ensure that local resolutions are inherited from the original mesh.

### A.3  Processor edges

The processor of RIGNO consists of a sequence of message-passing blocks with skip connections, which act on the regional nodes and the edges that connect them. We define the first level of the bidirectional edges between the regional nodes as all the unique pairs of connected vertices in a Delaunay triangulation on the regional nodes. In addition to the first level edges, we also define edges that connect long-distance regional nodes, which allows learning multiple spatial scales and enables a more efficient propagation of information in a message-passing step. The construction of these edges is illustrated in Figure 1 as *multi-scale edges*. For the second level, we randomly choose a subset the regional nodes and repeat the above process on them. We repeat this process multiple times by sub-sampling the subsets again.

The structure of the regional mesh can be regarded as a hierarchy of multiple sub-meshes with different resolutions and shared nodes (see Figures A.1b and A.1c, and the *multi-scale edges* box of Figure 1). In one message-passing step, the coarser sub-meshes, which have longer edges, allow the information to flow in larger scales, while the finer sub-meshes allow for assimilating small-scale details. All regional nodes are part of the finest sub-mesh, while only a few nodes take part in the coarsest one. In a single message-passing step, it is not possible for all the nodes to receive information from longer distances. Because of this, it is essential to have multiple message-passing steps in the processor to allow for the information to propagate through the whole domain.

## A.4  Fixed and variable physical meshes

For some applications in engineering (e.g., shape optimization), the physical mesh changes for every sample in the training dataset or at inference. Since the coordinates of the input discretization will change, the graph construction needs to be carried out every time. For training with large datasets, these graphs can be constructed only once as a pre-processing step. At inference, however, the time for constructing the graph is considerable compared to the execution of the model. On the other hand, another class of applications in science and engineering only require training and inference on a fix unstructured physical mesh. For these problems, constructing the graph can only be done once.

For the latter class of problems, RIGNO is trained using a fixed physical mesh. However, if the graph construction is only done once during training, regional mesh will consequently be fixed too. In this case, although the encoder and the decoder are invariant to the input discretization, the processor might be biased to that specific fixed regional mesh. Such a model can exhibit a poor performance when it is inferred with a different regional mesh. In order to avoid this situation, we update the regional mesh before every epoch during training.

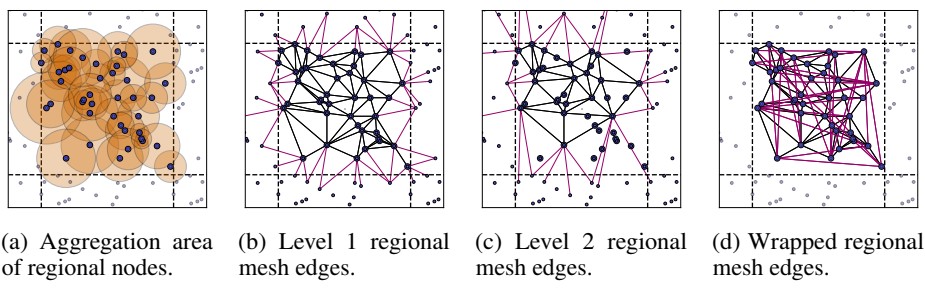

(a) Aggregation area of regional nodes.  (b) Level 1 regional mesh edges.  (c) Level 2 regional mesh edges.  (d) Wrapped regional mesh edges.

Figure A.1: Construction of regional mesh edges and support sub-regions with periodic boundary conditions. The dashed lines show the domain boundaries. Large and small purple dots represent the main regional nodes and the ghost regional nodes, respectively. Black lines represent the main edges, and the purple lines the cross-boundary edges. Note that in Figure A.1a, the parts of the domain that are close to the domain boundaries and do not lie in any circle are covered by at least one regional node on the other side of the boundary.

## A.5  Periodic boundary conditions

The majority of the problems considered in this work have periodic boundary conditions. The flexibility of graphs allows us to enforce periodic boundary conditions in the architecture without any significant computational overhead. To achieve this, we make the following three changes to our framework for problems with periodic boundary conditions: 1) Initialize the node features with a periodic encoding of the coordinates instead of raw coordinates; 2) extend the domain before defining the encoder and the decoder edges; and 3) extend the domain before defining the processor edges.

In order to motivate the above changes, let us consider two nodes, one close to the left and one close to the right boundary. With (normalized) raw coordinates, the initial features of these two nodes will be very different as one will take a value close to $0$ and the other one takes a value close to $1$. With periodic boundary conditions, however, these two points will be close to each other if the frame of the reference domain is slightly shifted. The invariance to a shift in the frame of reference means that these two nodes should be connected in the graphs. In simpler words, we assume during graph construction that the both ends of a domain with periodic boundary conditions are connected. In

one-dimensional settings, this set-up corresponds to considering a circular domain, see Figure A.2 for an illustration. For a square two-dimensional domain, connecting the two ends in both direction corresponds to a torus.

Inspired by the *ghost node* technique in classical numerical methods, we extend the domain in all directions by constructing a 3×3 tiled structure, where each tile contains the repeated coordinates of the given point cloud. All the Delaunay triangulations in the graph construction phase are built based on the extended domain. Figure A.1 shows the resulting support sub-regions, as well as the extra edges that are introduced because of the domain extensions for a minimal set of regional nodes. We refer to these extra edges as *cross-boundary* edges. Notice how all the white areas in Figure A.1a are covered by at least on circle on the other side of the domain. In order to avoid introducing extra costs to the architecture, the ghost nodes are discarded after graph construction and only the cross-boundary edges are kept. In order to keep the extra edges, we replace each ghost node with its corresponding node in the original point cloud. The result is illustrated in Figure A.1d.

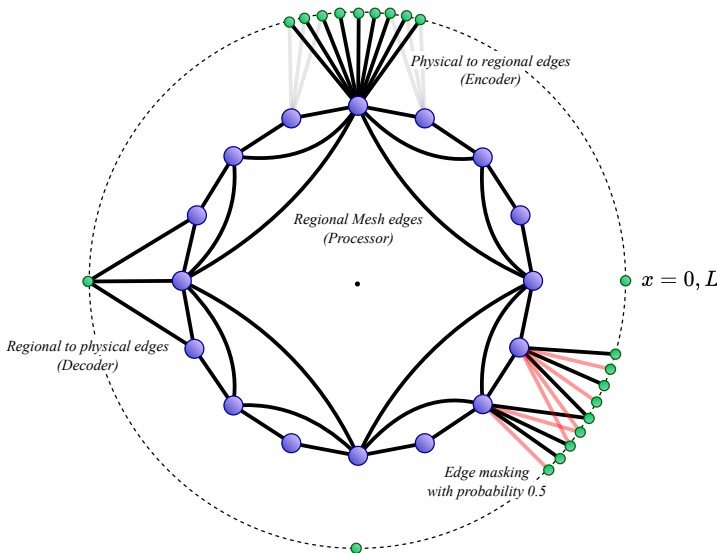

Figure A.2: Graph structure for periodic boundary conditions in one-dimensional settings. The green dots represent physical nodes, the purple circles represent regional nodes, and the solid lines represent edge connections. Note that only a few physical nodes are visualized. The bottom right part of the figure partly shows the receptive fields of two regional nodes in the encoder. Here, the red lines show the edges that are randomly masked, and hence will not be used in that particular message passing block.

## A.6 Structural features

The corresponding coordinates of the nodes are pre-processed to provide the initial structural features of the graph entities. In order to impose translation equivariance to the model, we consider linear normalized coordinates $\boldsymbol{\zeta}_i \in [-1, +1]$ and angular normalized coordinates $\boldsymbol{\alpha}_i \in [0, 2\pi]$ defined as $\boldsymbol{\zeta}_i := 2(\mathbf{x}_i - \mathbf{x}_c)/L - 1$ and $\boldsymbol{\alpha}_i := \pi(\boldsymbol{\zeta}_i + 1)$. The structural features of an edge $\boldsymbol{e}_{ij}$ connecting node $i$ to node $j$ is $\boldsymbol{\varphi}_{ij} := [\tilde{\boldsymbol{\zeta}}_{ij} \diamond \|\tilde{\boldsymbol{\zeta}}_{ij}\|_2]$; with $\tilde{\boldsymbol{\zeta}}_{ij} := (\boldsymbol{\zeta}_j - \boldsymbol{\zeta}_i)/(2\sqrt{d})$ being the relative coordinates of the connected nodes normalized by the diameter of the domain, $2\sqrt{d}$. We denote the vertical concatenation of column vectors by $\diamond$. The structural features of a node $\boldsymbol{v}_i$ with general boundary conditions are simply its normalized coordinates $\boldsymbol{\varphi}_i := \boldsymbol{\zeta}_i$. With periodic boundary conditions (see Section A.5), we concatenate sines and cosines of angular normalized coordinates with $K_{\text{freq}}$ frequencies: $\boldsymbol{\varphi}_i = [\sin(k\boldsymbol{\alpha}_i) \diamond \cos(k\boldsymbol{\alpha}_i)]$ for $1 \leq k \leq K_{\text{freq}}$.

# B  Details of the GNN Components

We consider two underlying sets of nodes (see Figure 1). $\boldsymbol{V}_{\mathfrak{P}} = \{\boldsymbol{v}_i^{\mathfrak{P}}\}_i$ denotes the physical nodes, which corresponds to the unstructured point cloud on which the input (and output) data is defined. The regional nodes are denoted by $\boldsymbol{V}_{\mathfrak{R}} = \{\boldsymbol{v}_i^{\mathfrak{R}}\}_i$ and the bidirectional edges that connect them by $\boldsymbol{E}_{\mathfrak{R} \leftrightarrow \mathfrak{R}} = \{\boldsymbol{e}_{ij}^{\mathfrak{R} \leftrightarrow \mathfrak{R}}\}_{ij}$. $\boldsymbol{E}_{\mathfrak{P} \rightarrow \mathfrak{R}} = \{\boldsymbol{e}_{ij}^{\mathfrak{P} \rightarrow \mathfrak{R}}\}_{ij}$ and $\boldsymbol{E}_{\mathfrak{R} \rightarrow \mathfrak{P}} = \{\boldsymbol{e}_{ij}^{\mathfrak{R} \rightarrow \mathfrak{P}}\}_{ij}$ denote the unidirectional edges in the encoder and the deocder, respectively.

The first step is to embed the initial features of all nodes and all edges (see Section A.6) to suitable high-dimensional features. The other model inputs (initial condition, spatial parameters, and time) are only embedded in the features of the physical nodes. For regional nodes, we also embed the radius of their support sub-region $R_i$. This is done by passing all the initial features through the embedder feed-forward blocks (*FF*, see Figure 1),

$$
\begin{aligned}
\boldsymbol{v}_i^{\mathfrak{P},(0)} &= FF_{\text{embedder}}^{\mathfrak{P}}(\boldsymbol{u}_i^t, \boldsymbol{\varphi}_i, \boldsymbol{c}_i^t, t, \tau) && \text{(Physical nodes)}, \\
\boldsymbol{v}_i^{\mathfrak{R},(0)} &= FF_{\text{embedder}}^{\mathfrak{R}}(\boldsymbol{\varphi}_i, R_i) && \text{(Regional nodes)}, \\
\boldsymbol{e}_{ij}^{\mathfrak{P} \rightarrow \mathfrak{R},(0)} &= FF_{\text{embedder}}^{\mathfrak{P} \rightarrow \mathfrak{R}}(\boldsymbol{\varphi}_{ij}) && \text{(Physical to regional edges)}, \\
\boldsymbol{e}_{ij}^{\mathfrak{R} \leftrightarrow \mathfrak{R},(0)} &= FF_{\text{embedder}}^{\mathfrak{R} \leftrightarrow \mathfrak{R}}(\boldsymbol{\varphi}_{ij}) && \text{(Regional mesh edges)}, \\
\boldsymbol{e}_{ij}^{\mathfrak{R} \rightarrow \mathfrak{P},(0)} &= FF_{\text{embedder}}^{\mathfrak{R} \rightarrow \mathfrak{P}}(\boldsymbol{\varphi}_{ij}) && \text{(Regional to physical edges)}.
\end{aligned}
\tag{5}
$$

Next, the encoder applies a single message passing step to the embedded features of the physical nodes and transmits the information to the regional mesh,

$$
\begin{aligned}
\boldsymbol{m}_{ij}^{\mathfrak{P} \rightarrow \mathfrak{R},(0)} &= FF_{\text{encoder}}^{\mathfrak{P} \rightarrow \mathfrak{R}}(\boldsymbol{v}_i^{\mathfrak{P},(0)}, \boldsymbol{v}_j^{\mathfrak{R},(0)}, \boldsymbol{e}_{ij}^{\mathfrak{P} \rightarrow \mathfrak{R},(0)}), \\
\boldsymbol{v}_i^{\mathfrak{R},(1)} &= \boldsymbol{v}_i^{\mathfrak{R},(0)} + FF_{\text{encoder}}^{\mathfrak{R}}(\boldsymbol{v}_i^{\mathfrak{R},(0)}, \tfrac{1}{|\mathcal{N}_i|} \sum_{j \in \mathcal{N}_i} \boldsymbol{m}_{ji}^{\mathfrak{P} \rightarrow \mathfrak{R},(0)}), \\
\boldsymbol{v}_i^{\mathfrak{P},(1)} &= \boldsymbol{v}_i^{\mathfrak{P},(0)} + FF_{\text{encoder}}^{\mathfrak{P}}(\boldsymbol{v}_i^{\mathfrak{P},(0)}).
\end{aligned}
\tag{6}
$$

The processor passes this latent information on the regional mesh through $P$ sequential message-passing blocks with residual updates and skip connections. The $p$-th message-passing step in the processor updates the regional nodes and the regional mesh edges as,

$$
\begin{aligned}
\boldsymbol{m}_{ij}^{\mathfrak{R} \leftrightarrow \mathfrak{R},(p-1)} &= FF_{\text{processor}}^{\mathfrak{R} \leftrightarrow \mathfrak{R},(p)}(\boldsymbol{v}_i^{\mathfrak{R},(p)}, \boldsymbol{v}_j^{\mathfrak{R},(p)}, \boldsymbol{e}_{ij}^{\mathfrak{R} \leftrightarrow \mathfrak{R},(p-1)}), \\
\boldsymbol{v}_i^{\mathfrak{R},(p+1)} &= \boldsymbol{v}_i^{\mathfrak{R},(p)} + FF_{\text{processor}}^{\mathfrak{R},(p)}(\boldsymbol{v}_i^{\mathfrak{R},(p)}, \tfrac{1}{|\mathcal{N}_i|} \sum_{j \in \mathcal{N}_i} \boldsymbol{m}_{ji}^{\mathfrak{R} \leftrightarrow \mathfrak{R},(p-1)}), \\
\boldsymbol{e}_{ij}^{\mathfrak{R} \leftrightarrow \mathfrak{R},(p)} &= \boldsymbol{e}_{ij}^{\mathfrak{R} \leftrightarrow \mathfrak{R},(p-1)} + \boldsymbol{m}_{ij}^{\mathfrak{R} \leftrightarrow \mathfrak{R},(p-1)}.
\end{aligned}
\tag{7}
$$

The decoder then transfers processed information from the regional mesh back to the physical mesh by performing a single message passing step,

$$
\begin{aligned}
\boldsymbol{m}_{ij}^{\mathfrak{R} \rightarrow \mathfrak{P},(0)} &= FF_{\text{decoder}}^{\mathfrak{R} \rightarrow \mathfrak{P}}(\boldsymbol{v}_i^{\mathfrak{R},(P_{\mathfrak{R}}+1)}, \boldsymbol{v}_j^{\mathfrak{P},(1)}, \boldsymbol{e}_{ij}^{\mathfrak{R} \rightarrow \mathfrak{P},(0)}), \\
\boldsymbol{v}_i^{\mathfrak{P},(2)} &= \boldsymbol{v}_i^{\mathfrak{P},(1)} + FF_{\text{decoder}}^{\mathfrak{P}}(\boldsymbol{v}_i^{\mathfrak{P},(1)}, \tfrac{1}{|\mathcal{N}_i|} \sum_{j \in \mathcal{N}_i} \boldsymbol{m}_{ji}^{\mathfrak{R} \rightarrow \mathfrak{P},(0)}).
\end{aligned}
\tag{8}
$$

Finally, in the second step of the decoder, the feature vectors of the physical node are independently scaled down to the desired dimension to form the output

$$
\boldsymbol{r}_i = FF_{\text{decoder}}^{\text{output}}(\boldsymbol{v}_i^{\mathfrak{P},(2)}).
\tag{9}
$$

## C Temporal Processing

### C.1 Time stepping strategies

As described in the main text, we consider three time stepping strategies, namely direct *output* stepping, *residual* stepping, and *time-derivative* stepping. It is important to normalize the outputs correspondingly in each of these strategies since the nature and the statistics of the target variables are different in each case. Therefore, we pre-compute the appropriate statistics in each strategy based on the training samples and use them to normalize and de-normalize the inputs and the outputs. With direct output stepping, the statistics of the inputs and the outputs are trivial. For residual stepping, we pre-compute the statistics of the residuals with all the lead times that are present in the training. For time-derivative stepping, we similarly consider all lead times and compute the statistics of the residual divided by the lead time. Figure C.1 illustrates flowcharts for input and output normalization with each stepping strategy.

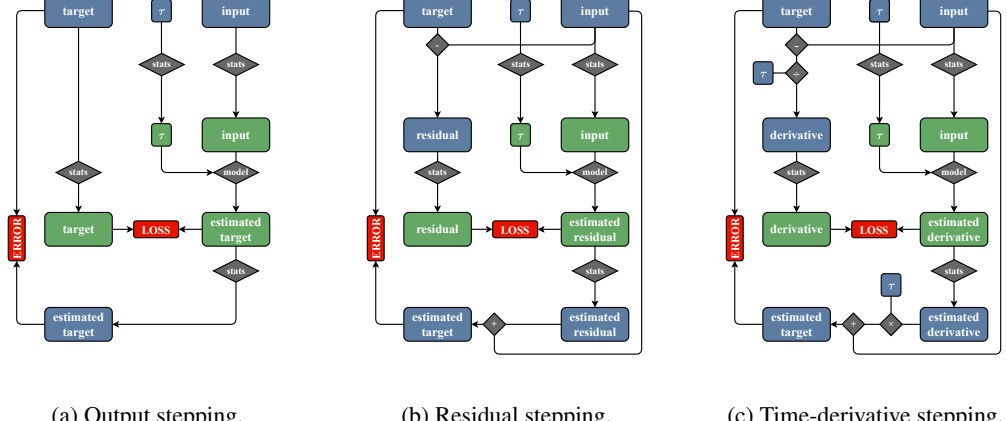

(a) Output stepping.       (b) Residual stepping.       (c) Time-derivative stepping.

Figure C.1: Flowcharts of input and output normalization strategies. Blue boxes represent raw data and green boxes represent normalized data. We apply a min-max normalization for the input time $t$, which is omitted in the flowcharts for simplicity. The loss is computed on normalized values while the errors are computed on the raw targets and de-normalized estimates. Note that the raw target and the de-normalized estimates may be normalized in a different manner or with different statistics (e.g., global equation-wide statistics as in [13]) before the error is computed on them.

### C.2 Temporal inference

Given any set of partial lead times $\{\tau_i\}_{i=1}^n$ that sum up to a time $t \in \Omega_t$, a time-continuous operator $\mathcal{G}_\theta$ can be applied sequentially to reach the solution at time $t$:

$$u(t) \simeq \mathcal{G}_\theta \left( \mathcal{G}_\theta \left( \mathcal{G}_\theta \left( \ldots, c, \sum_{i=1}^{n-3} \tau_i, \tau_{n-2} \right), c, \sum_{i=1}^{n-2} \tau_i, \tau_{n-1} \right), c, \sum_{i=1}^{n-1} \tau_i, \tau_n \right). \quad (10)$$

This gives an infinite number of ways to infer the learned operator for reaching $u(t)$. However, since $\mathcal{G}_\theta$ is not exact, the approximation errors generally accumulate and can lead to very high errors for long-time estimations. With larger lead times it is possible to decrease the number of times that the operator is applied or to completely avoid autoregressive inference by directly using $\tau = t$.

In all the time-dependent datasets considered in this work, for simplicity of exposition, we have considered $N_t = 20$ or $N_t = 21$ time snapshots $\{t_n\}_{n=1}^{N_t}$. Unless otherwise stated, only snapshots with even indices up to $t_{14}$ are used for training. Snapshots with $t > t_{14}$ are reserved for time extrapolation tests, and snapshots with $t < t_{14}$ with even indices are reserved for time interpolation tests. Error at snapshot $t_{14}$ is used as a reference for the benchmarks. Since the trained operator is continuous in time and can be inferred with different lead times, there are infinitely many ways to reach the solution at $t_{14}$ via autoregressive inference of the trained operator. The optimal autoregressive

path can be different for different operators. Figure C.2 illustrates the training and test time snapshots as well as different autoregressive paths for reaching $t_{14}$. In the presented benchmarks in Tables 1 and 2, we calculate the error with 3 different autoregressive paths: i) $\{\tau_i = 2\Delta t\}_{i=1}^{7}$ which is denoted by AR-2; ii) $\{\tau_1 = 4\Delta t, \tau_2 = 4\Delta t, \tau_3 = 4\Delta t, \tau_4 = 2\Delta t\}$ which is denoted by AR-4; and iii) $\{\tau_1 = 14\Delta t\}$, denoted by DR. We always report the lowest error among these three rollout paths.

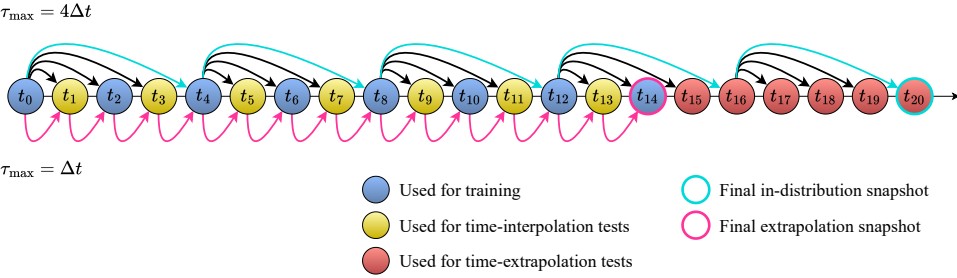

Figure C.2: Snapshots used for training, time interpolation, and time extrapolation. The in-distribution and extrapolation evaluation snapshots are marked with thick borders. The lower arrows show the autoregressive steps for evaluating a learned operator at $t_{14}$ with $\tau_{\max} = \Delta t$ (14 steps). Note that in this rollout path, a lead time $(\Delta t)$ that is smaller than the time resolution of the training dataset $(2\Delta t)$ is applied 14 times in the evaluation. The upper arrows show the autoregressive steps for evaluating a learned operator at $t_{20}$ with $\tau_{\max} = 4\Delta t$ (5 steps).

## C.3 Fractional pairing

We use the fractional pairing strategy to achieve a truly time-continuous operator that can be inferred with lead times smaller than the smallest time difference present in the training dataset. In fractional pairing, we approximate the solution at an intermediary time by inferring a model that has been trained on the original time resolution. This approximation is then given as input to the same model. By choosing a suitable intermediary time, we can have an input-output pair that is not present in the training dataset and have a smaller lead time than the time resolution of the training dataset. This idea is illustrated in Figure C.3. With this, we can either calculate the loss gradients for both of the inferences, or we can cut the gradient at the intermediary time and treat the intermediary solution as a normal model input, which is referred to as *GradCut* (see bottom left of Figure C.3). In the experiments of this work, we always use fractional pairing with *GradCut*.

If the approximation at the intermediary time is not *accurate enough*, fractional pairing strategy can lead to training instabilities. Therefore, it is recommended to use this technique only as a fine-tuning stage. Furthermore, since the approximations are made with *fractional lead times*, i.e., lead times that are not present in the training dataset, it is also essential to employ other mechanisms that help generalize to fractional lead times with regular training strategies, such as consistent time marching strategies and time-conditioned normalization layers. A further consideration that helps the stability of this strategy is avoiding *out-of-distribution* lead times in the first forward pass, i.e., avoiding lead times that are smaller than the time resolution of the training dataset. Such small lead times can be seen as an extrapolation in the seen distribution of lead times during training. It has been observed that even without fractional pairing, the models generalize better to interpolated lead times than they do to extrapolated ones. Therefore, we can approximate the intermediary solution with an interpolated lead time, and use a smaller lead time only in the second forward pass. In Figure C.3, this corresponds to only using the two right-most additional pairs in the green box. Note that none of the additional pairs of the orange box will be viable in this setting.

## D Datasets

RIGNO is tested on a suite of 13 datasets [32] which covers operators arising from compressible Euler (CE) equations of gas dynamics, incompressible Navier-Stokes (NS) equations of fluid dynamics, wave equation (WE), heat equation (HE) of thermal dynamics, Allen-Cahn equation (ACE), Poisson's

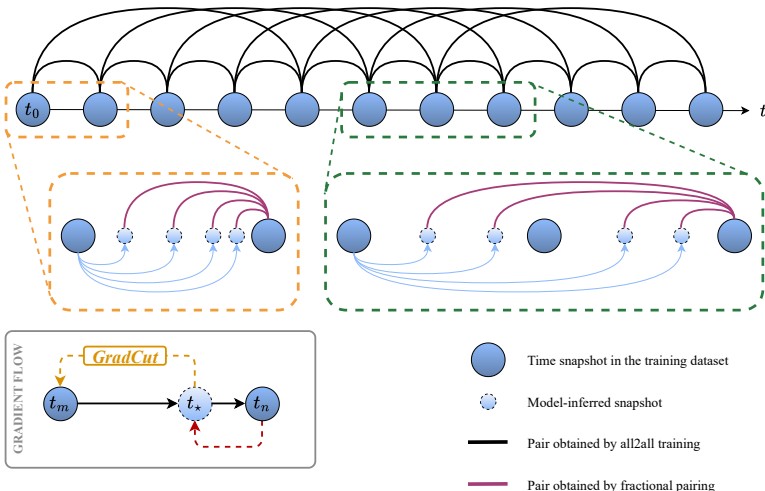

Figure C.3: Schematic of the fractional pairing strategy. The additional pairs obtained by fractional pairing correspond to smaller lead times than the smallest time difference in the original trajectory. On the bottom left of the figure, the forward passes (solid black arrows) and the backward gradient flows (dashed arrows) are illustrated for an existing pair $(t_m, t_n)$. With the *GradCut* setting, the gradient flow is stopped at $t_\star$.

equation, and force balance equation in solid mechanics (FB). Table D.1 summarizes the main characteristics of all the datasets. A random sample of each dataset is visualized in Figures D.2-D.14. This collection of datasets covers diverse multi-scale, nonlinear, complex phenomena in systems arising from PDEs such as shock waves, shear layers, mixing, and turbulence. This makes the effort to learn the underlying solution operator from data worthwhile for these datasets; since running simulations of these systems with any classical numerical method would require extensive computational resources. Moreover, by considering these datasets, we cover a diverse set of domains and meshes from Cartesian uniform grids on a unit square to structured and unstructured meshes on fixed and varying irregular geometries.

The problems in the 8 time-dependent datasets from [13] are all solved in the unit spatio-temporal domain $\Omega_t \times \Omega_x = [0, T] \times [0, 1]^2$. The final time is $T = 1$ except for the ACE dataset where $T = 0.0002$. For each dataset, a parameterized distribution of initial conditions is considered, and initial conditions are randomly drawn by sampling those distributions. The distribution of the initial conditions characterizes the dataset, the level of complexity of the dynamics captured in it, the phenomenological features exhibited in the solution function, and the corresponding solution operator. In all these datasets, $N_t = 21$ *snapshots* of each sample are collected at times $\Omega_t^\Delta = \{t_n = n\Delta t\}_{n=0}^{N_t-1}$ with uniform spacing $\Delta t$; except for the ACE and the Wave-Layer datasets where $N_t = 20$. Each snapshot originally has a $128 \times 128$ resolution, with uniform discretization in both spatial directions. In order to test the baselines on a random point cloud with these datasets, we randomly shuffle the coordinates and only keep the first 9216 coordinates in our experiments. Figure D.1 illustrates a few random point clouds obtained from this procedure.

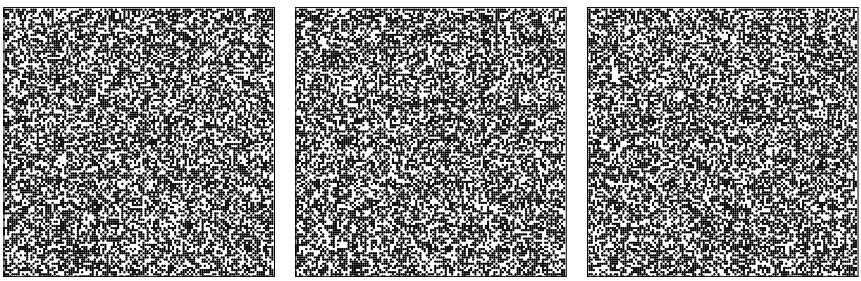

Figure D.1: Random point clouds (9216 points) obtained from a $128 \times 128$ uniform grid.

Table D.1: Abbreviations and main characteristics of all datasets. IC: initial condition; BC: boundary conditions.

| Abbreviation | PDE | BC | Domain (varies) | Time | Source | Figure |
|---|---|---|---|---|---|---|
| Heat-L-Sines | HE | Dirichlet | L-shape (✗) | ✓ | | D.2 |
| Wave-C-Sines | WE | Dirichlet | circle (✗) | ✓ | | D.3 |
| NS-Gauss | INS | periodic | square (✗) | ✓ | [13] | D.4 |
| NS-PwC | INS | periodic | square (✗) | ✓ | [13] | D.5 |
| NS-SL | INS | periodic | square (✗) | ✓ | [13] | D.6 |
| NS-SVS | INS | periodic | square (✗) | ✓ | [13] | D.7 |
| CE-Gauss | CE | periodic | square (✗) | ✓ | [13] | D.8 |
| CE-RP | CE | periodic | square (✗) | ✓ | [13] | D.9 |
| ACE | ACE | periodic | square (✗) | ✓ | [13] | D.10 |
| Wave-Layer | WE | absorbing | square (✗) | ✓ | [13] | D.11 |
| Poisson-Gauss | PE | Dirichlet | square (✗) | ✗ | [13] | D.13 |
| AF | CE | various | airfoil (✓) | ✗ | [19] | D.12 |
| Elasticity | FB | various | square with holes (✓) | ✗ | [19] | D.14 |

## D.1 Equations

### D.1.1 Incompressible Navier-Stokes equations (INS)

The incompressible Navier-Stokes equations are given by

$$\nabla \cdot \boldsymbol{v} = 0 \qquad \text{(Conservation of mass)},$$
$$\partial_t \boldsymbol{v} + (\boldsymbol{v} \cdot \nabla)\boldsymbol{v} = -\nabla p + \nu \nabla^2 \boldsymbol{v} \quad \text{(Conservation of momentum)}, \tag{11}$$

where $\boldsymbol{v} : \Omega_t \times \Omega_x \to \mathbb{R}^2$ is the velocity field with components $(v_x, v_y)^\top$, $p : \Omega_t \times \Omega_x \to \mathbb{R}_+$ is the pressure, and $\nu \in \mathbb{R}$ is the fluid viscosity. In all the INS datasets, periodic boundary conditions are considered and a small viscosity $\nu = 4 \times 10^{-4}$ is applied to the high Fourier modes only.

### D.1.2 Compressible Euler equations (CE)

The compressible Euler equations in the conservational form are given by

$$\partial_t \boldsymbol{u} + \nabla \cdot \boldsymbol{F} = \boldsymbol{S},$$
$$\boldsymbol{u} = (\rho, \rho\boldsymbol{v}, E)^\top,$$
$$\boldsymbol{F} = (\rho\boldsymbol{v}, \rho\boldsymbol{v} \otimes \boldsymbol{v} + p\boldsymbol{I}, (E + p)\boldsymbol{v})^\top,$$
$$E = \frac{1}{2}\rho\|\boldsymbol{v}\|^2 + \frac{p}{\gamma - 1}, \tag{12}$$
$$\boldsymbol{S} = (0, -\rho, 0, -\rho v_x)\frac{\partial \phi}{\partial x} + (0, 0, -\rho, -\rho v_y)\frac{\partial \phi}{\partial y},$$

with $\gamma = 1.4$, where $\boldsymbol{F}$ is the flux matrix, $E$ is the total energy density (energy per unit volume), and $\phi$ is the gravitational potential field. The gravity source term is not considered in any of the datasets of this work.

### D.1.3 Allen-Cahn equation (ACE)

The Allen–Cahn equation describes phase transition in materials science and is given by

$$\partial_t u - \nabla^2 u = \epsilon^2 (u^3 - u), \tag{13}$$

where $\epsilon$ is the reaction coefficient.

### D.1.4 Wave equation (WE)

The wave equation is considered with a fixed-in-time propagation speed $c$ that depends on the spatial location. This setting describes the propagation of acoustic waves in an inhomogeneous medium:

$$\partial_{tt} u - c^2 \nabla^2 u = 0. \tag{14}$$

The wave equation can be written in the form of (1) by considering the first time derivative $\partial_t u$ as a variable and writing it as a system of equations. The propagation speed can either be added as an input to the differential operator, or as another variable with $\partial_t c = 0$. In the former case, it should also be added to the inputs of any solution operator.

### D.1.5 Heat equation (HE)

The heat equation describes the diffusion of heat in a medium and is given by

$$\partial_t u = a^2 \nabla^2 u, \tag{15}$$

where $a$ is the diffusivity constant.

### D.1.6 Poisson's equation (PE)

Poisson's equation is given by

$$-\nabla u = f, \tag{16}$$

where $f$ is a given source function and $u$ is the solution.

### D.1.7 Force balance equation (FB)

The force balance equation of a solid body is given by

$$\rho \partial_t \boldsymbol{u} + \nabla \cdot \boldsymbol{\sigma} = 0, \tag{17}$$

where $\rho$ denotes mass density, $\boldsymbol{u}$ the displacement field, and $\boldsymbol{\sigma}$ the stress tensor. The strain field is related to the displacement field via kinematic relations. The system is closed by considering a constitutive model which describes the relation between strain and stress.

## D.2 Dataset details

### D.2.1 Heat-L-Sines

This dataset considers time-dependent solutions of the heat equation over an *L*-shaped domain which is formed by removing the rectangular portion $[1, 1.5] \times [0.5, 1]$ from the larger rectangle $[0.5, 1.5] \times [0, 1]$. We consider constant diffusivity $a = 1.0$ and zero homogeneous Dirichlet boundary conditions. We prescribe a superposition of sinusoidal modes as the initial condition:

$$u_0(x_1, x_2, \boldsymbol{\mu}) = -\frac{1}{M} \sum_{m=1}^{M} \mu_m \frac{\sin(\pi m x_1) \sin(\pi m x_2)}{m^{0.5}}, \tag{18}$$

where $\boldsymbol{\mu} \in [-1, 1]^d$, $M = 10$, and each $\mu_m$ is sampled independently and uniformly from $[-1, 1]$. A Delaunay-based triangular mesh with characteristic length 0.008 is used, yielding about 14,047 nodes and 27,590 elements. We simulate the heat equation for 100 time steps with a time step size of $\Delta t = 5 \times 10^{-5}$ and keep 21 (including the initial condition) equally spaced snapshots from each trajectory.

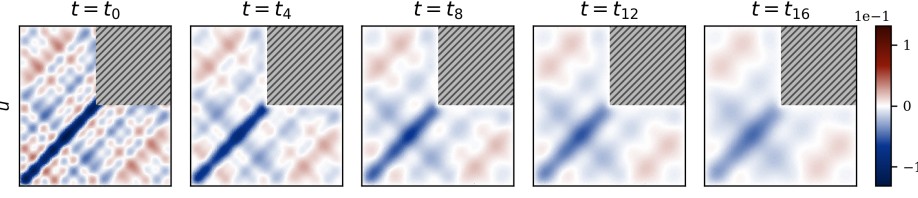

Figure D.2: Snapshots of solution trajectories of the Heat-L-Sines dataset.

### D.2.2 Wave-C-Sines

This dataset pertains to time-dependent solutions of the two-dimensional wave equation over a circular domain of radius 1, centered at $(0.5, 0.5)$. We set $c = 2.0$, constant in time and in space, and impose homogeneous Dirichlet boundary conditions ($u = 0$) on the boundary of the circle. The initial condition for $u$ at $t = 0$ is specified as a superposition of sinusoidal modes:

$$u_0(x, y) = \frac{\pi}{M^2} \sum_{i,j}^{M} a_{ij} \, (i^2 + j^2)^{-r} \, \sin(\pi i x) \, \sin(\pi j y), \tag{19}$$

where $K = 10$, $r = 0.5$, and $a_{ij} \sim \mathcal{U}_{[-1,1]}$. We employ a Delaunay-based triangular mesh with characteristic length 0.015, resulting in roughly 16,431 nodes and 32,441 elements. A time step of $\Delta t = 0.001$ is used, and solutions are computed for 100 time steps via a finite element solver. We then downsample the computed solution fields in time, keeping 21 (including the initial condition) equally spaced time frames, which constitute the dataset.

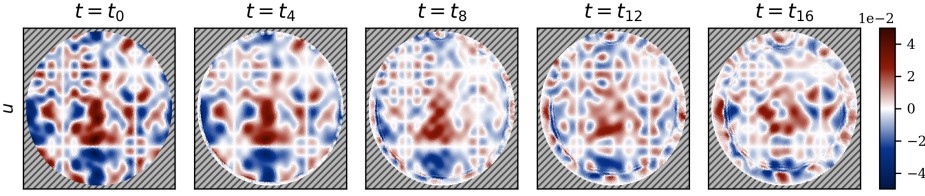

Figure D.3: Snapshots of solution trajectories of the Wave-C-Sines dataset.

### D.2.3 NS-Gauss

The initial velocities are obtained from an initialized vorticity defined by $\omega(x, y) = \partial_x v_y - \partial_y v_x$ by using the incompressibility condition [29]. The vorticity field is initialized as a linear superposition of $M = 100$ Gaussian distributions:

$$\omega^0(x, y) = \sum_{i=1}^{M} \frac{\alpha_i}{\sigma_i} \exp\left(-\frac{(x - x_i)^2 + (y - y_i)^2}{2\sigma_i^2}\right), \tag{20}$$

with $\alpha_i \sim \mathcal{U}_{[-1,1]}$, $\sigma_i \sim \mathcal{U}_{[0.01,0.1]}$, $x_i \sim \mathcal{U}_{[0,1]}$, and $y_i \sim \mathcal{U}_{[0,1]}$. A random sample of this dataset is visualized in Figure D.4.

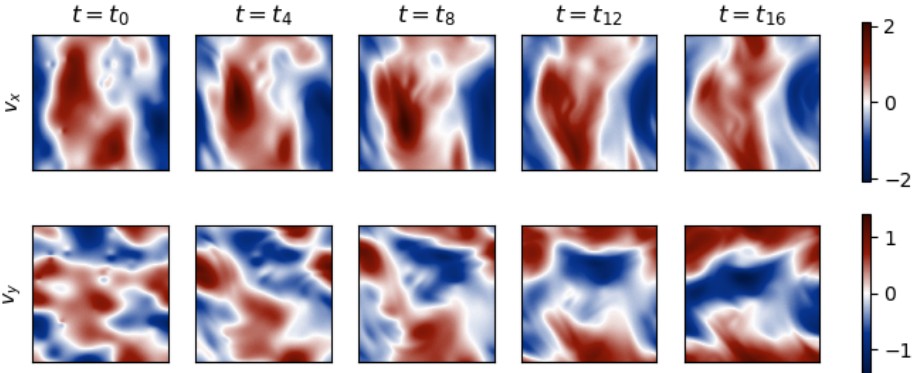

Figure D.4: Snapshots of solution trajectories of the NS-Gauss dataset. Each row corresponds to one variable.

### D.2.4 NS-PwC

The initial velocities are obtained from an initialized vorticity by using the incompressibility condition. The initial vorticity $\omega^0$ of the flow field is initialized as a collection of piece-wise constant squares

$$\omega^0(x, y) = c_{i,j} \text{ in } [x_{i-1}, x_i] \times [y_{j-1}, y_j], \tag{21}$$

with $x_i = y_i = \frac{i}{P}$ for $i = 0, 1, 2, \ldots, P$, and $c_{i,j} \sim \mathcal{U}_{[-1,1]}$. The number of squares in each direction is $P = 10$, resulting in 100 squares in total. A random sample of this dataset is visualized in Figure D.5.

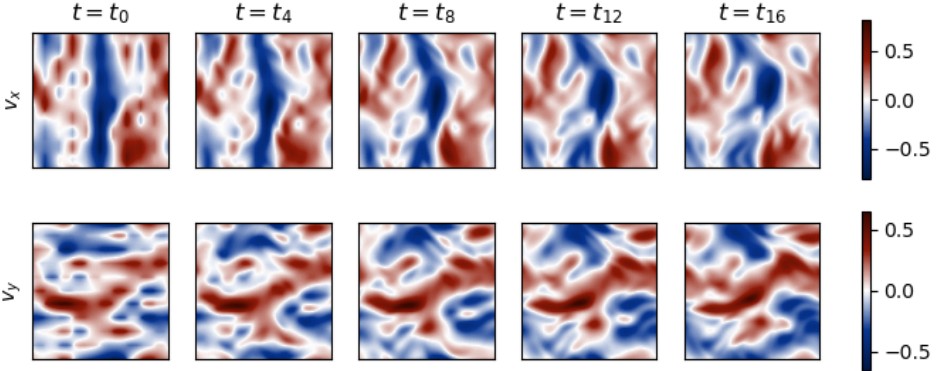

Figure D.5: Snapshots of solution trajectories of the NS-PwC dataset. Each row corresponds to one variable.

### D.2.5 NS-SL

The velocity field $\boldsymbol{v} = (v_x, v_y)$ is initialized as

$$
v_x^0(x,y) = \begin{cases} \tanh\left(2\pi \frac{y-0.25}{\rho}\right) & \text{for } y + \sigma_\delta(x) \leq \frac{1}{2} \\ \tanh\left(2\pi \frac{0.75-y}{\rho}\right) & \text{otherwise} \end{cases},
$$

$$
v_y^0(x,y) = 0,
$$

(22)

where $\sigma_\delta : [0,1] \to \mathbb{R}$ is a perturbation of the initial data given by

$$
\sigma_\delta(x) = \xi + \delta \sum_{k=1}^{p} \alpha_k \sin(2\pi k x - \beta_k).
$$

(23)

The parameters are sampled as $p \sim \mathcal{U}_{\{7,8,9,10,11,12\}}$, $\alpha_k \sim \mathcal{U}_{[0,1]}$, $\beta_k \sim \mathcal{U}_{[0,2\pi]}$, $\delta = 0.025$, $\rho \sim \mathcal{U}_{[0.08,0.12]}$, and $\xi \sim \mathcal{U}_{[-0.0625,0.0625]}$. A random sample of this dataset is visualized in Figure D.6.

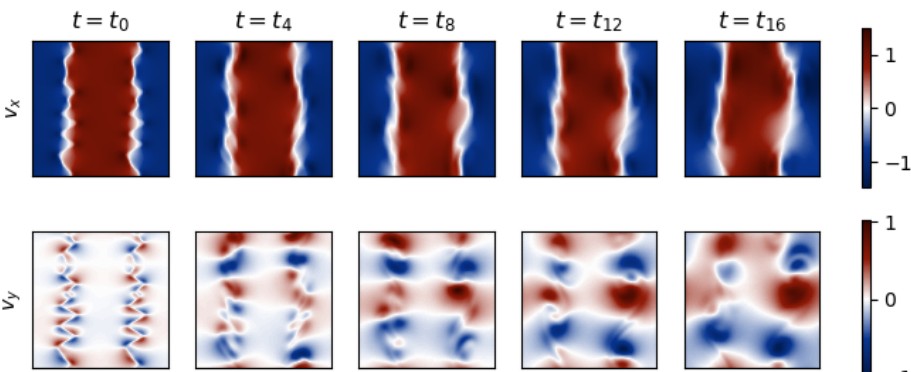

Figure D.6: Snapshots of solution trajectories of the NS-SL dataset. Each row corresponds to one variable.

### D.2.6 NS-SVS

The vorticity of the flow is initialized as

$$
\omega^0(x,y) = \phi_\rho(x,y) * \omega'(x,y),
$$

(24)

where

$$\omega^{'}(x,y) = \delta(x - \Gamma) - \int_{\mathbb{T}^2} \mathrm{d}\Gamma, \tag{25}$$

$$\phi_\rho(x,y) = \rho^{-2}\psi\left(\frac{\|x\|}{\rho}\right), \tag{26}$$

$$\psi(r) = \frac{80}{7\pi}\left[(r+1)_+^3 - 4(r + \frac{1}{2})_+^3 + 6r_+^3 - 4(r - \frac{1}{2})_+^3 + (r-1)_+^3\right], \tag{27}$$

$$\Gamma = \{(x,y) \in \mathbb{T}^2 \mid y = \frac{1}{2} + 0.2\sin(2\pi x) + \sum_{i=1}^{10}\alpha_i\sin(2\pi(x+\beta_i))\}. \tag{28}$$

The random variables $\alpha_i$ and $\beta_i$ are given by $\alpha_i \sim \mathcal{U}_{[0,0.003125]}$, $\beta_i \sim \mathcal{U}_{[0,1]}$. The parameter $\rho$ is fixed as $\rho = \frac{5}{128}$. A random sample of this dataset is visualized in Figure D.7.

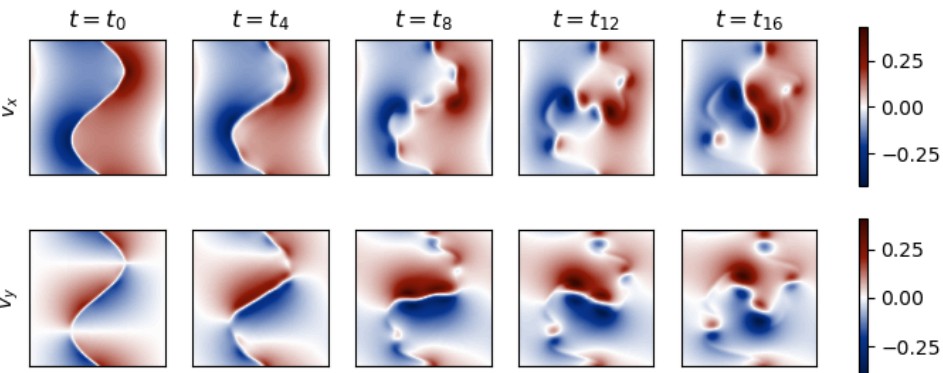

Figure D.7: Snapshots of solution trajectories of the NS-SVS dataset. Each row corresponds to one variable.

### D.2.7 CE-Gauss

The $\omega$ field is initialized as a linear combination of $M = 100$ Gaussians

$$\omega^0(x,y) = \sum_{i=1}^{M}\frac{\alpha_i}{\sigma_i}\exp\left(-\frac{(x-x_i)^2 + (y-y_i)^2}{2\sigma_i^2}\right), \tag{29}$$

where $\alpha_i \sim \mathcal{U}_{[-1,1]}$, $\sigma_i \sim \mathcal{U}_{[0.01,0.1]}$, $x_i \sim \mathcal{U}_{[0,1]}$, and $y_i \sim \mathcal{U}_{[0,1]}$. The density and pressure are initialized with $\rho = 0.1$ and $p = 2.5$, both constant in the whole domain. Similar to the NS-Gauss dataset, the initial velocity field is obtained from the initialized vorticity by using the incompressibility condition. A random sample of this dataset is visualized in Figure D.8.

### D.2.8 CE-RP

This dataset corresponds to the four-quadrant Riemann problem, which generalizes the standard Sod shock tube to two spatial dimensions. The domain is divided into a grid of $P \times P$ square subdomains $D_{i,j} = \{(x,y) \in \mathbb{T}^2 \mid \frac{i-1}{P} \le x < \frac{i}{P}, \frac{j-1}{P} \le y < \frac{j}{P}\}$. The solution on each of these subdomains is constant and initialized as

$$(\rho, v_x, v_y, p)(x,y) = (\rho_{i,j}, u_{i,j}, v_{i,j}, p_{i,j}), \tag{30}$$

where $\rho_{i,j} \sim \mathcal{U}_{[0.1,1]}$, $u_{i,j} \sim \mathcal{U}_{[-1,1]}$, $v_{i,j} \sim \mathcal{U}_{[-1,1]}$, and $p_{i,j} \sim \mathcal{U}_{[0.1,1]}$. In this dataset, $P = 2$ is considered, which constitutes 4 subdomains in total. A random sample of this dataset is visualized in Figure D.9.

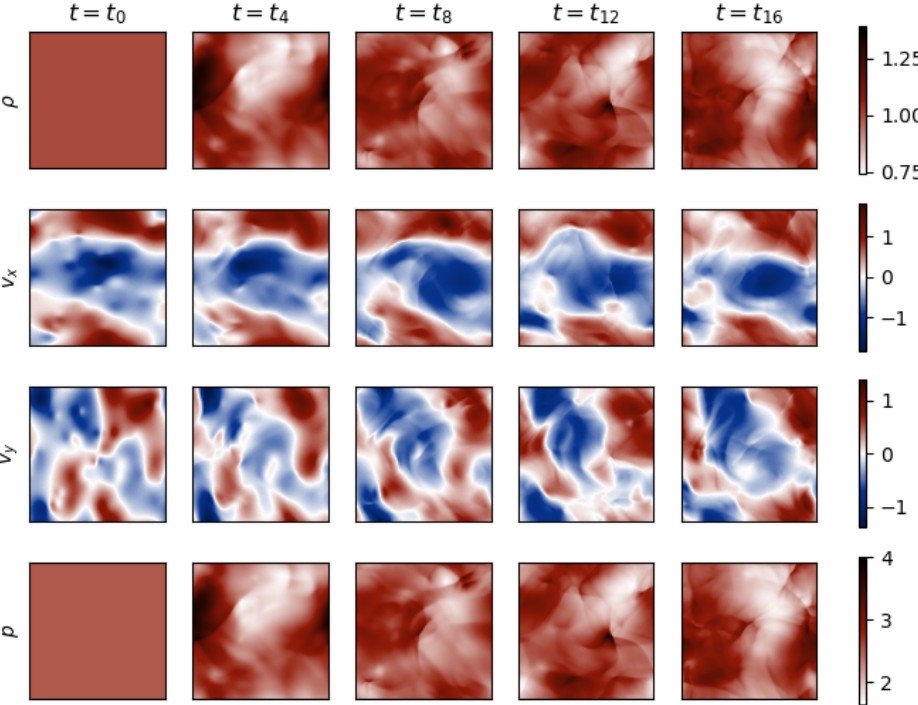

Figure D.8: Snapshots of solution trajectories of the CE-Gauss dataset. Each row corresponds to one variable.

### D.2.9 ACE

The Allen–Cahn equation (13) are considered with periodic boundary conditions and reaction coefficient $\epsilon = 220$. $u$ is initialized as

$$u^0(x,y) = \frac{1}{M^2} \sum_{i,j=1}^{M} a_{ij} \cdot (i^2 + j^2)^{-r} \sin(\pi i x) \sin(\pi j y), \tag{31}$$

where $M$ is a random integer randomly sampled from $[16, 32]$, $r \sim \mathcal{U}_{[0.7,1]}$, and $a_{ij} \sim \mathcal{U}_{[-1,1]}$. A random sample of this dataset is visualized in Figure D.10.

### D.2.10 Wave-Layer

The wave equation (14) is considered with the fixed-in-time propagation speed $c(x, y)$ as a piece-wise constant function in a vertically layered medium with $n \in \{3, 4, 5, 6\}$ layers and randomly defined intersections. More details about the definition of layers are given in SM B.2.11 of [13]. $u$ is initialized as a linear combination of $n \in 2, 3, 4, 5, 6$ Gaussian distributions centered at random locations in the domain. A random sample of this dataset is visualized in Figure D.11.

### D.2.11 AF

In this dataset, the transonic flow is considered over different airfoils. The steady-state form of the compressible Euler equations (12) govern this system. The viscous effects are ignored and the far-field boundary conditions of $\rho_\infty = 1$, $p_\infty = 1.0$, $M_\infty = 0.8$ and zero angle of attack are considered. At the airfoil boundary, no-penetration conditions are considered. We refer the reader to [19] for more details about how the airfoil geometries are drawn. The flow is computed on a C-grid mesh with $200 \times 50$ quadrilateral elements by a second-order implicit finite volume solver. The original dataset is enhanced by adding a distance field from the boundary of the airfoil which is used as the input of the model. The target value used in this work is the density $\rho$.

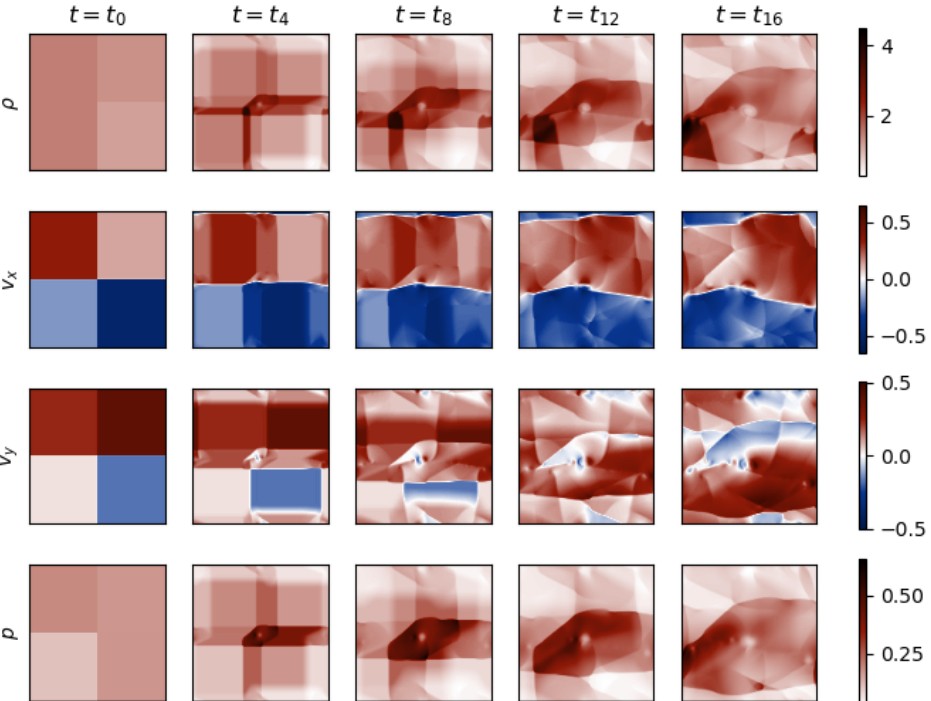

Figure D.9: Snapshots of solution trajectories of the CE-RP dataset. Each row corresponds to one variable.

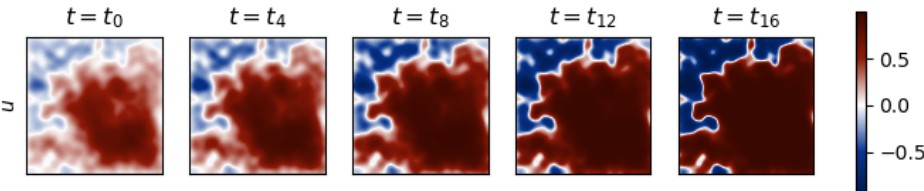

Figure D.10: Snapshots of solution trajectories of the ACE dataset. Each row corresponds to one variable.

### D.2.12 Poisson-Gauss

Poisson's equation (16) is considered in a unit square with homogeneous Dirichlet boundary conditions. the source term $f$ is composed of a superposition of $M$ random Gaussians as

$$f(x,y) = \sum_{i=1}^{M} \exp\left(-\frac{(x - \mu_{x,i})^2 + (y - \mu_{y,i})^2}{2\sigma_i^2}\right), \tag{32}$$

where $M$ is a random integer drawn from a geometric distribution, $\mu_{x,i}, \mu_{y,i} \sim \mathcal{U}_{[0,1]}$, and $\sigma_i \sim \mathcal{U}_{[0.025,0.1]}$. A few input-output pairs of this dataset are visualized in Figure D.13.

### D.2.13 Elasticity

The solid body force balance equation (17) is considered for a unit square hyper-elastic incompressible Rivlin-Saunders specimen with a hole at its center. The geometry of the hole is randomly sampled from a certain distribution (see [19] for more details) such that the radius always takes a value between $0.2$ and $0.4$. The specimen is clamped at the bottom boundary (zero displacement) and is under a constant vertical tension traction on its top boundary. The stress field is computed for different

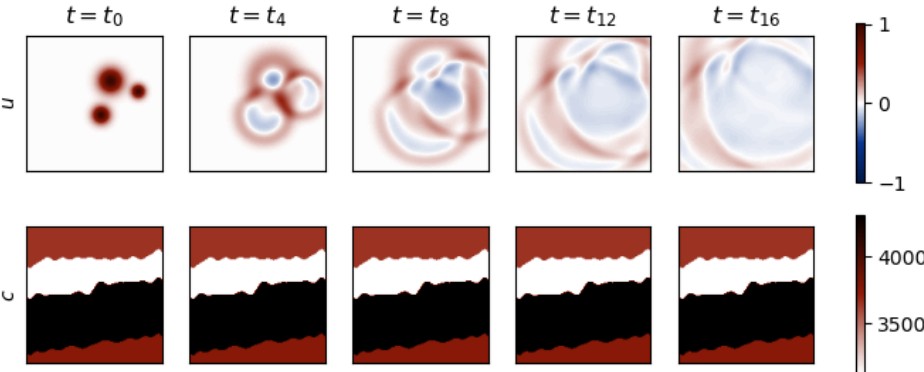

Figure D.11: Snapshots of solution trajectories of the Wave-Layer dataset. Each row corresponds to one variable.

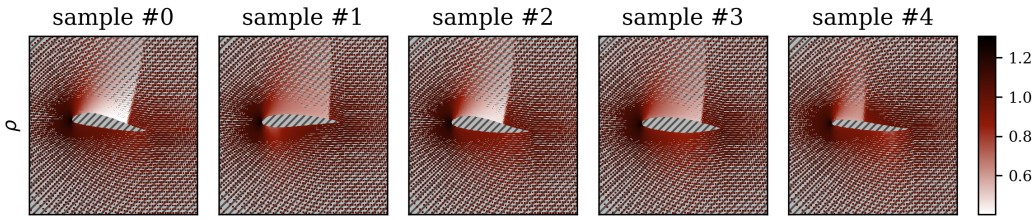

Figure D.12: Solution samples of the AF dataset (only density). The full domain is much larger and the mesh is more sparse than the visualized frame around the airfoil.

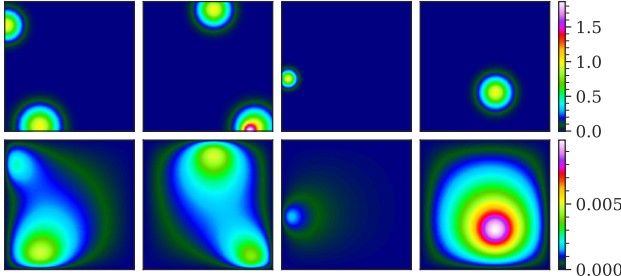

Figure D.13: Inputs (top row, source function) and outputs (bottom row, solution) of the Poisson-Gauss dataset. Each column corresponds to one sample in the dataset.

geometries with a finite elements solver with about 100 quadratic quadrilateral elements. The target function in this dataset is the stress field, which is available at 972 unstructured coordinates. The original dataset is enhanced by adding a distance field from the boundary of the hole that is given as input to the models.

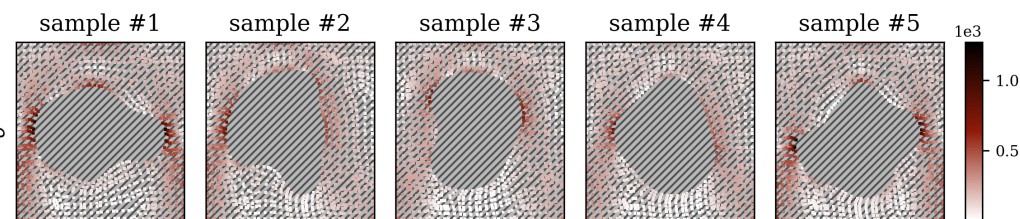

Figure D.14: Solution samples of the Elasticity dataset.

# E Details of the Experiments

A central design principle of RIGNO is to minimize the need for extensive hyperparameter tuning by employing a fixed set of hyperparameters across all tasks. To this end, we identified a robust configuration by tuning on a single internal dataset and subsequently applied the same hyperparameters to all other datasets without further adjustment. The size and performance of RIGNO are primarily influenced by only two key architectural hyperparameters: the number of processor blocks and the latent dimensionality of edge and node features. A comprehensive list of the hyperparameters used in RIGNO-18 and RIGNO-12 is provided in Table E.1. Since most of these hyperparameters require no task-specific tuning, the same configuration can be readily adopted for new problems or datasets. The only distinction between the two architectures lies in the number of message-passing blocks. Unless otherwise stated, the settings described in this section are used for all experiments. As shown in Table 1, both RIGNO-18 and RIGNO-12 achieve consistently strong performance across diverse benchmarks without the need for task-specific tuning.

The last feed-forward block in the decoder that independently acts on physical nodes does not have normalization layers, i.e., it is only composed of a two-layer MLP. A nonlinear activation function is applied on the output of the hidden layer in all MLPs. For the main MLPs of the feed-forward blocks, we use the *swish* activation function [37]; whereas for the MLPs of the conditioned-normalization layers, the *sigmoid* function is used.

Table E.1: Architectural hyperparameters of RIGNO and their corresponding values for RIGNO-18 and RIGNO-12.

| RIGNO-18 | RIGNO-12 | Description |
|---|---|---|
| 2.7M | 1.9M | Total number of trainable parameters. |
| 18 | 12 | Number of message passing blocks. |
| 4 | 4 | Factor for sub-sampling the given coordinates. As an example, with `16,000` input coordinates, we get `4,000` coordinates in the regional mesh. |
| 1.0 | 1.0 | Overlap factor of the support sub-regions in the encoder. |
| 2.0 | 2.0 | Overlap factor of the support sub-regions in the decoder. |
| 6 | 6 | Number of regional edge levels. |
| 128 | 128 | Dimension of the latent node features. |
| 128 | 128 | Dimension of the latent edge features and messages. |
| 1 | 1 | Number of the hidden layers in the MLP of the feed-forward blocks. |
| 128 | 128 | Dimension of the hidden layers in the MLP of the feed-forward blocks. |
| 16 | 16 | Dimension of the hidden layers in the MLPs of conditional normalization layers. |
| 4 | 4 | Number of frequencies for encoding coordinates, i.e., $K_{\text{freq}}$. |
| 0.5 | 0.5 | Probability for edge masking. |

We train our models with 1024 training samples for all datasets; except for the AF dataset where 2048 samples are used in the trainings. We use a different set of 128 samples for validation, and 256 samples for testing. The test errors reported Table 2 are exceptionally calculated on 240 test samples. We train our models for 2500 epochs on the time-dependent datasets, and for 3000 epochs

on the time-independent datasets. We calculate the errors on the validation dataset regularly and keep the model parameters that give the lowest validation error. We use batch size 16, the AdamW [27] optimizer with weight decay $10^{-8}$, the mean squared error loss function, and cosine learning rate decay with linear warm-up period. The learning rate is warmed up in the first 2% training epochs from $10^{-5}$ to $2 \times 10^{-4}$, and decays with cosine schedule to $10^{-5}$ in the next 88% of the epochs. In the last 10% epochs, it exponentially decays to $10^{-6}$. For all the time-dependent datasets, the *time-derivative* time stepping scheme is used due to its better performance (see Figure 3c). We use the *all2all* training strategy [13] with a warm-up period. We first train the model with all the pairs with lead time $2\Delta t$ for a few epochs, then add the pairs with lead time $4\Delta t$ and train for more epochs, and continue until all lead times are covered. The warm-up period for the *all2all* training takes 20% of the training epochs.

Most of the experiments with RIGNO have been done using 2 NVIDIA GeForce RTX 3090 GPUs (24GB). Some experiments have been carried out with more than 2 (up to 8) GPUs in order to speed up the trainings. We estimate a total of 9'000 GPU hours for reproducing the presented experiments with RIGNO (including **SM**).

We evaluate the estimates of the models against the ground-truth target functions following the metric used in [13]. We first normalize both the model estimate and the ground-truth function values by shifting and scaling them with global equation-wide mean and standard deviations. We then compute the relative $L^1$ error per each variable (group) as

$$
e_k^m = \frac{\sum_i \left| u_{k,i}^m - \widehat{u}_{k,i}^m \right|}{\sum_i \left| u_{k,i}^m \right|},
\tag{33}
$$

where $u_{k,i}^m$ and $\widehat{u}_{k,i}^m$ are respectively model estimate and ground-truth function values for sample $m$, variable group $k$, and spatial coordinate $i$. For all the Navier-Stokes and the Compressible Euler datasets, we also sum the components of the the velocity vector here. Hence, we get a scalar relative error $e_k^m$ per each target variable (group) and per each sample. We average over $k$ to get a scalar relative error per sample, and report the median error over all samples in the validation or the test dataset.

## F   Further Experimental Results

There are many hyperparameters that can affect the training stability as well as different aspects of the performance of a trained RIGNO. In the experiments presented in this work, we fix a set of architectural choices that perform fairly well for all the datasets and training sizes. These hyperparameters are listed in Table E.1. Note that these hyperparameters do not correspond to the best-performing architecture, but rather they guarantee a reasonable performance with all the desired properties. In Section F.3, we show that the performance can be improved by using larger models. In the experiments of this chapter, we perturb one or a few of these hyperparameters while leaving the rest fixed in order to study how the performance of the model changes with respect to the changed hyperparameters. The trainings are carried out using the settings described in Section E, unless otherwise stated.

### F.1   Training randomness

In order to quantify the dependence of the final results on training randomness, we train RIGNO five times, each time with a different seed for the pseudo-random number generator. We store the model with the lowest validation error during training as in all experiments, and check for fluctuations in the test error across all five repetitions of each experiment. In these experiments, we use the unstructured version of the datasets (random point clouds), we train using 512 training samples with batch size 8, and do not fine tune with fractional pairing. The results are summarized in Table F.1, where the maximum and minimum test errors across the repetitions are reported for each dataset. These fluctuations are attributed to the randomness in processes such as weight initializations, dataset shuffling, edge masking, and random sub-sampling of the coordinates. In the rest of the experiments, we only train each model once with a fixed training random seed, keeping in mind that the errors can slightly deviate because of training randomness.

Table F.1: Statistics of relative $L^1$ test errors with different random seeds. We train RIGNO-12 on 512 samples from each datasets. Each experiment is repeated 5 times.

| Dataset | Error [%] |
| --- | --- |
| | Mean $\pm$ Standard deviation |
| NS-PwC | $2.50 \pm 0.157$ |
| NS-SVS | $1.05 \pm 0.040$ |

## F.2  Number of regional nodes

The number of regional nodes is independent from the given physical nodes and can be chosen arbitrarily. With more regional nodes, the support sub-regions will be smaller on average, leading to fewer edges in the encoder and the decoder. However, the processor steps become significantly more expensive. In order to investigate the effect of different numbers of regional nodes on the performance, we train a slightly modified RIGNO-18 on the NS-SL and the NS-SVS datasets. The test results are presented in Table F.2. With only 1024 regional nodes, the model already achieves a reasonable error for both datasets. With more regional nodes, the error follows a decreasing trend with both datasets, although less significant with NS-SVS. However, the inference times show that fewer regional nodes are favorable in terms of computational efficiency. RIGNO-18 is 5 times more time-efficient with 1024 regional nodes as compared to 16384 regional nodes. Moreover, the decreasing trend of the error breaks with too many regional nodes. With sub-sample factor 1.0 (16384 regional nodes), the support sub-regions become so small that most of them only contain one physical node and hence cannot reflect a good representation of the whole spatial region. In the main experiments of the current work, we have chosen a sub-sample factor of 4.0 to maintain a balance between accuracy and efficiency.

Table F.2: Test errors and inference times with different number of regional nodes. The experiments are conducted with a variation of RIGNO-18, trained on 256 training trajectories for 2000 epochs without all2all pairing. Although the sub-sampling factor for obtaining the regional nodes changes, multi-scale edges are always obtained based on sub-meshes of factor 2.0 until the regional mesh saturates.

| Regional nodes (sub-sample factor) | Median relative $L^1$ error [%] NS-SL | NS-SVS | Inference time [ms] |
| --- | --- | --- | --- |
| 1024 (16.0) | 4.32 | 2.90 | 7.0 |
| 2048 (8.0) | 3.96 | 2.93 | 9.0 |
| 4096 (4.0) | 3.49 | 2.84 | 12.9 |
| 8192 (2.0) | 3.33 | 2.83 | 28.3 |
| 16384 (1.0) | 4.12 | 3.05 | 35.8 |

## F.3  Scaling with model size

In order to evaluate how the performance of RIGNO scales with model size, we start with RIGNO-12 as the base architecture and change the latent sizes (all at the same time), the number of message passing steps in the processor, the overlap factors (both at the same time), and the number of times that the regional mesh is sub-sampled when processor edges are built (number of regional edge levels). We train these models on 256 samples of the NS-PwC dataset without fractional pairing and calculate the error at $t_{14}$ on 128 test samples by using the AR-2 rollout scheme (defined in Section C.2). The results are plotted in Figure F.1.

We observe that the latent sizes and number of processor steps have the largest effect on the model capacity. The error consistently falls down when these hyperparameters are scaled, and we expect to get even lower errors with larger RIGNO architectures. With larger overlap factors, although the number of edges (and hence the computational cost) increases rapidly, the rate of convergence is relatively low. Therefore, the authors recommend using the minimum overlap, especially for the encoder. With more regional mesh levels, on the other hand, the number of edges only increases

logarithmically, which is also reflected in the inference times. With 4 regional mesh levels, we obtain 3.75% error, which is extremely lower than the 17.10% error with only a single regional mesh level. The inference time, however, is only increased by 14%. These results show the effectiveness of longer edges in the construction of the regional mesh. However, depending on the space resolution of the inputs, there is saturation limit for this hyperparameter. If we consider a downsampling factor of 4, with 16'000 input coordinates, for instance, we end up with a mesh with only 4 nodes after 6 downsampling steps. With so few nodes we can only get 6 additional edges at most, which explains why the decreasing trend in the error stops at 4 downsampling steps.

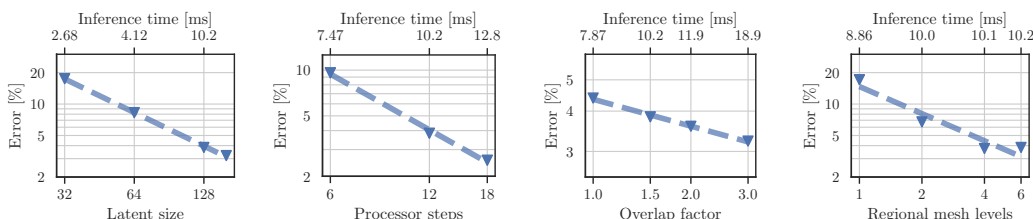

Figure F.1: Test errors at $t = t_{14}$ with different strategies for scaling model size. The experiments are carried out on the NS-PwC dataset.

## F.4 Spatial continuity: resolution invariance

In Figure 3a, we presented results of zero-shot super-resolution and zero-shot sub-resolution inference of RIGNO. The results show that RIGNO maintains its accuracy when inferred with higher resolutions but the accuracy drops with lower resolutions. When trained with edge masking, zero-shot generalization to lower resolutions is considerably improved. Generalization to higher resolutions is often a desirable feature since the model can be trained with lower resolutions with considerably less training costs, and be deployed for high-resolution inputs and outputs. In Figure F.2, we present the results of the resolution invariance test on MeshGraphNet. The results show that MeshGraphNet admits great dependence on the training resolution, with very large errors on any resolution except for the training resolution. This dependence is alleviated when MeshGraphNet is trained with edge masking. The accuracy is much less affected both with lower and higher resolutions. However, edge masking alone is not enough to make MeshGraphNet resolution invariant. With only 4 times higher resolutions, the error is almost doubled (55.16%) as compared to the error with the training resolution (30.36%).

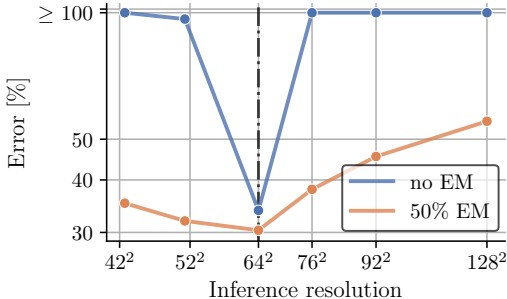

Figure F.2: Resolution invariance test if MeshGraphNet with $64^2$ training resolution on the NS-PwC dataset. EM stands for edge masking. The y-axis shows the relative test error (at $t = t_{14}$) when the spatial resolution of the inputs and outputs varies from the training resolution.

## F.5 Temporal continuity: generalization to unseen time steps

In order to assess the generalization of the trained models to unseen lead times $\tau$, we train RIGNO on the NS-PwC dataset with $2\Delta t$ time resolution, and test it on the test dataset with $\Delta t$ time resolution. With the incorporated mechanisms, namely time-conditioned normalization layers, all2all training, and fractional pairing fine-tuning, we expect to see a monotone increase in the error as we increase

the lead time; which is very well captured in our tests as visualized in Figure F.3. Here, the lead time $\tau$ is fixed at each line in the plot, and the input time snapshot $t$ is changed. This explains why the first entries are missing for larger lead times. In the uppermost line ($\tau = 9\Delta t$), for instance, the first entry corresponds to $t = 0$, $\tau = 9\Delta t$; the second entry to $t = 1\Delta t$, $\tau = 10\Delta t$; and so on.

In Figure F.3, the errors increase with larger lead times, which is expected since estimating the solution of the PDE with a higher lead time is inherently more difficult. In these experiments, we have removed the pairs with $\tau > 8\Delta t$ from the all2all training pairs. Hence, $\tau = 9\Delta t$ is considered as extrapolation in lead time. Since lead times $\tau = 3\Delta t$, $\tau = 5\Delta t$, and $\tau = 7\Delta t$ are also not present in the all2all pairs, they are considered as interpolation in lead time. We can see that the monotonicity of the increase in the error is preserved for the extrapolated as well as the interpolated lead times. Moreover, the error increases in a *controlled* manner. The authors believe that this property is attributed to incorporation of time-conditioned normalization layers in the architecture and the use of time-derivative time-stepping strategy. However, with the smallest lead time $\tau = \Delta t$, it was observed that the monotonicity was not preserved without fine-tuning with the fractional pairing strategy (see Figure 3b).

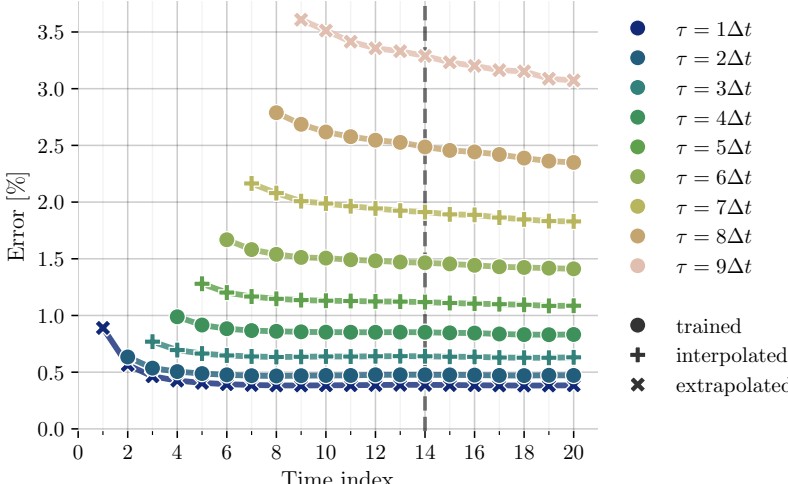

Figure F.3: Direct test errors with different $\tau$ for the NS-PwC dataset. The vertical line indicates $t = t_{14}$, which is the last time snapshot seen by the model during training. Different lines denote different lead times. The gray lines correspond to lead times present in the training dataset; the green lines highlight interpolation in lead time; and the red lines indicate extrapolation in lead time. The model is trained on 512 samples with batch size 4.

### F.6 Model uncertainty

The edge masking technique randomly *disables* a subset of the edges in each message-passing block. By adding randomness to the model, edge masking allows inferring the same model multiple times with the same input in order to generate an ensemble of estimates at a later time. The standard deviation of the generated ensemble can be interpreted as an indicator of model uncertainty on each output coordinate.

Figure F.4 visualizes the mean and the standard deviation of an ensemble of estimates at $t = t_{14}$ generated from input at $t = t_0$, along with the absolute error of the mean with respect to the ground-truth solution. More such visualizations are available in Section H. These visualizations show how the regions with the highest errors largely overlap with the regions where the ensemble has the highest deviations; which supports the interpretation of the deviations as a measure for model uncertainty, and shows that they are good indicators for inaccuracies in the estimates generated by the model.

Generally, the uncertainty of the model increases over autoregressive rollouts or with larger lead times. The model uncertainty can therefore be used to optimize the set of lead times in an autoregressive path. A possible algorithm would be to adaptively choose the largest lead time $\tau$ that results in uncertainties smaller than a threshold. Since model inference is fast and differentiable, grid-search

algorithms can also be implemented. Similarly, the input discretization can adaptively be refined at each time step $t$ in the areas where the uncertainty of the model in its output is the highest.

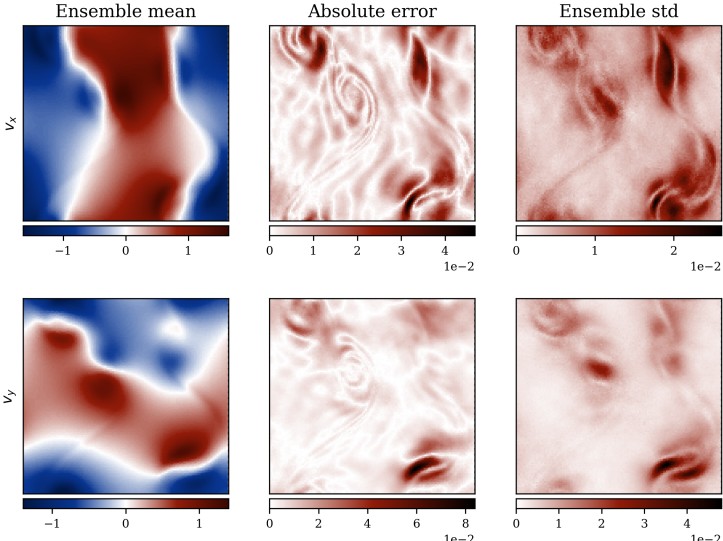

Figure F.4: Mean and standard deviation of an ensemble of autoregressive estimates at $t_{14}$ for a test sample of the NS-SL dataset.

## F.7 Robustness to noise

In many applications, a learned operator needs to be inferred with inputs that are collected experimentally. Since measurement noise is inevitable in any experimental setup, robustness to noisy inputs is vital for such solution operators. Ideally, the operator should dampen the noise in the input; or at least the noise should not propagate in an uncontrolled way. To evaluate the robustness of RIGNO to input noise, we train it using *clean*, noise-free data, infer it with noisy inputs, and compute the error of the output with respect to clean ground-truth data. This setting emulates applications where the operator has been trained on noise-free high-fidelity simulation data and is applied on noisy experimental data.

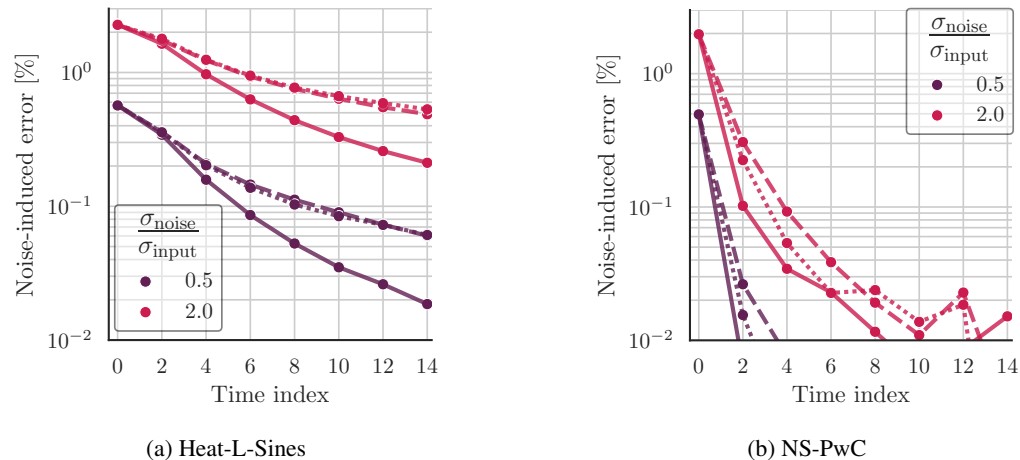

Figure F.5: Autoregressive test errors introduced by the noise in the input (relative). The y-axes show the difference in the relative error with noisy and clean inputs. The solid lines (best) show the results with edge masking and overlap factors `2.0`. The dashed lines show the results with the same overlap factors but without edge masking during training. The dotted lines show the results with edge masking and overlap factors `1.0`. All models are trained on 256 trajectories, without all2all training, and without fractional pairing fine-tuning.

For each input in the test dataset, we compute its standard deviation $\sigma_{\text{input}}$, and add a Gaussian noise with standard deviation $\sigma_{\text{noise}} = C_{noise}\sigma_{\text{input}}$, with a *noise level* $C_{noise} \in \{0.005, 0.02\}$. Starting with a noisy input, we apply the operators autoregressively with $\tau_{\max} = 2\Delta t$. The errors in the resulting trajectories are partially due to the input noise, and partially to the approximation error of the operator. In order to distinguish between these two, we also unroll the operator with clean input ($C_{noise} = 0$), and subtract the resulting errors from the errors with noisy inputs. This procedure gives a measure for the part of the error that is introduced by the noise in the input, denoted by *noise-induced error*. We present the relative noise induced error with respect to the approximation error: $|e_{\text{noisy}} - e_{\text{clean}}|/e_{\text{clean}}$, where $e$ represents the autoregressive test error at $t = t_{14}$. The results are visualized in Figure F.5. With $2\%$ input noise at $t_0$, the noise is very well dampened to only $0.5\%$ at $t_{14}$ of the Heat-L-Sines dataset. With larger support sub-regions, the regional nodes aggregate data from a larger pool of physical nodes. Therefore, they are less sensitive to noisy inputs, and a stronger noise dampening is observed. Edge masking also plays an important role in noise dampening. When larger support sub-regions are used along with edge masking, $2\%$ input noise is dampened below $0.2\%$ at $t_{14}$.

Furthermore, noise dampening of RIGNO is not limited to Gaussian noise, as there has been no indication of this type of noise during training of the model. The authors believe that RIGNO is also robust to the errors that are introduced during unrolling; if we consider the introduced errors as noise with a certain distribution. In other words, RIGNO *corrects itself* within successive autoregressive steps. Therefore, for certain problems, small -but not too small- lead times might be advantageous to larger ones due to noise dampening.

### F.8 Out-of-distribution generalization

A central challenge in operator learning, and in supervised machine learning more broadly, is achieving strong generalization to out-of-distribution (OOD) inputs and outputs. These models often experience a degradation in performance when evaluated on samples that deviate from the training distribution. While some degree of performance decay is inevitable, a robust framework should maintain reasonable accuracy under mild to moderate distribution shifts. To evaluate RIGNO's robustness in this regard, we train it on three datasets taken from [38], namely Smooth Transport, Darcy Flow, and Compressible Euler Equations. The trained models are then tested on 128 in-distribution (ID) and 128 OOD samples. The OOD samples differ from the training distribution in the location of perturbations, characteristic length scales, or geometry complexity. All models are trained for at least 2000 epochs using a batch size of 4 to ensure convergence, and all evaluation metrics are computed on unnormalized (raw) test data.

The results are summarized in Table F.3. Interestingly, RIGNO exhibits strong robustness to OOD samples in the Darcy Flow and Compressible Euler problems, but experiences a pronounced performance drop for the Smooth Transport problem when evaluated on OOD inputs. In the case of the Compressible Euler problem, the OOD samples feature airfoils with more complex geometries, while in Darcy Flow, the OOD inputs differ primarily in their characteristic length scales. Both cases can be interpreted as involving local variations in the structure of the input and output functions, to which RIGNO generalizes well. In contrast, the Smooth Transport problem introduces a non-local shift, as the center of the Gaussian perturbation changes significantly in the OOD samples, leading to the observed performance degradation.

Table F.3: Median relative $L^1$ errors [%] on 128 in-distribution (ID) and 128 out-of-distribution (OOD) test samples. The errors are calculated on unnormalized samples.

| Dataset | Training samples | ID | OOD |
|---|---|---|---|
| Darcy Flow | 256 | 2.57 | 3.18 |
| Smooth Transport | 512 | 0.72 | 6.50 |
| Compressible Euler | 512 | 0.51 | 0.62 |

### F.9 Pre-training and transfer learning

PDE foundation models have been proposed [13, 11, 44, 30] as a solution to overcome the high data demands of neural operators. In this context, we have assessed the generalization capabilities of

RIGNO to new input and output distributions. To this end, we pre-train RIGNO on 1024 trajectories (2000 epochs, without all2all pairing) of one dataset (e.g., NS-Gauss) and transfer it to a different dataset (e.g., NS-PwC). We fine-tune the pre-trained RIGNO with only a few trajectories of the target dataset and compare the results with a randomly initialized RIGNO. We have chosen the NS-Gauss, NS-PwC, and NS-SL datasets, which have the same underlying PDE (incompressible Navier-Stokes equation) but correspond to very different data distributions (see Figures D.4, D.5, and D.6). The results are depicted in Figure F.6. For the NS-PwC dataset, we find that with only 4 trajectories, the transferred RIGNO pre-trained on NS-Gauss has an error of 13.1%, while a RIGNO trained from scratch requires 128 trajectories to achieve the same error, providing a 32x gain in sample efficiency with transfer learning. Notably, the gain is smaller (roughly 4x) when the model is pre-trained on the NS-SL dataset. Similarly, for the NS-SL dataset as the target dataset, transfer learning with 32 samples lead to an error of 4.7%, which is 7 times less than the trained-from-scratch RIGNO (33.4%) with 32 samples. These examples illustrate the ability of RIGNO to transfer very well out of distribution.

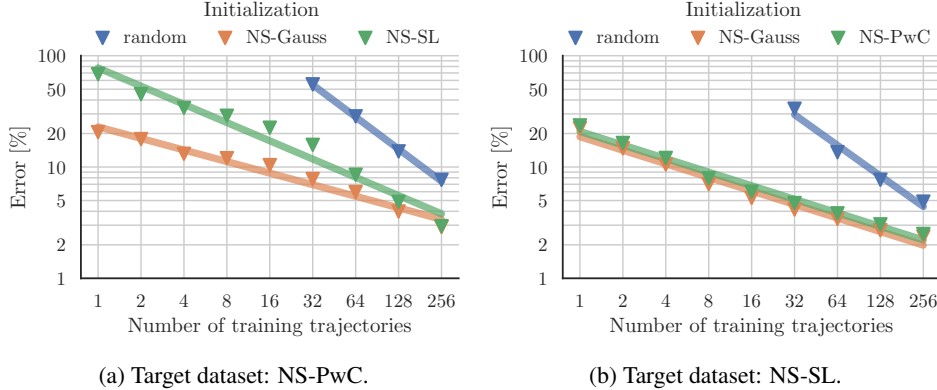

(a) Target dataset: NS-PwC.          (b) Target dataset: NS-SL.

Figure F.6: The effect of pre-training on data efficiency of RIGNO-12 with the unstructured NS datasets. All errors are relative test errors on 256 test samples.

## F.10   Performance benchmarks

We benchmarked RIGNO against the best-performing baseline, GINO, in terms of training time, accuracy, inference time, and peak memory usage during inference. For training, we used the maximum feasible batch size for RIGNO and a fixed batch size of 64 for GINO. In cases involving varying geometries across training samples, such as in the Elasticity dataset, the original GINO implementation does not support batch processing, requiring a reduced batch size of 1. We evaluated two RIGNO-18 configurations with 96 and 128 latent features, respectively, and two GINO variants comprising 3 and 7 FNO layers.

Table F.4: Performance benchmarks on two unstructured datasets evaluated on a single NVIDIA GeForce RTX 3090 GPU (24GB). TT: training time [s]; TE: test error [%]; IT: inference time [ms]; and PM: peak memory usage [MB].

| Architecture | Size | Poisson-Gauss | | | | Elasticity | | | |
|---|---|---|---|---|---|---|---|---|---|
| | | TT | TE | IT | PM | TT | TE | IT | PM |
| GINO (3) | 3.6M | 1.33 | 5.24 | 7.8 | 370 | 18.2 | 3.36 | 4.1 | 80 |
| GINO (7) | 8.4M | 1.72 | 8.01 | 10.1 | 406 | 27.1 | 3.94 | 6.4 | 116 |
| RIGNO-18 (96) | 1.4M | 24.9 | 2.23 | 8.1 | 121 | 2.4 | 4.33 | 1.5 | 51 |
| RIGNO-18 (128) | 2.5M | 28.4 | 2.09 | 12.7 | 159 | 3.1 | 3.71 | 1.9 | 64 |

The results, summarized in Table F.4, indicate that RIGNO trains more slowly than GINO on datasets with fixed geometries and input coordinates, but trains faster on datasets with variable geometries, primarily due to its support for batch processing. Although longer training times are a drawback,

training constitutes a one-time cost, whereas inference speed and memory usage are typically more critical in practice. In this regard, RIGNO and GINO exhibit comparable inference times, yet RIGNO consistently achieves substantially higher accuracy. Furthermore, RIGNO is up to three times more memory-efficient than GINO. Owing to its multi-scale architecture and computational efficiency, RIGNO stands out as a powerful end-to-end GNN that delivers state-of-the-art accuracy and robustness, while remaining competitive with strong non-GNN baselines in both speed and memory consumption.

# G  Baseline Architectures

## G.1  CNO

A *Convolutional Neural Operator* (CNO) is a model that (approximately) maps bandlimited functions to bandlimited functions [38]. Let $\mathcal{B}_w$ be the space of bandlimited functions with the bandlimit $w$. A CNO is compositional mapping between function spaces $\mathcal{G} : \mathcal{B}_w(D) \to \mathcal{B}_w(D)$ and is defined as

$$\mathcal{G} : u \mapsto P(u) = v_0 \mapsto v_1 \mapsto \ldots v_L \mapsto Q(v_L) = \bar{u}, \tag{34}$$

where

$$v_{l+1} = \mathcal{P}_l \circ \Sigma_l \circ \mathcal{K}_l(v_l), \quad 1 \le l \le L - 1, \tag{35}$$

where $L$ is the number of CNO blocks and $D$ is the domain.

Here, $P$ is the *lifting* operator, $Q$ is the *projection* operator, $\mathcal{P}_l$ is either the *upsampling* or *downsampling* operator, $\mathcal{K}_l$ is the *convolution* operator and $\Sigma_l$ is the (modified) *activation* operator. For continuous time conditioning, we used lead time-conditioned instance normalization, defined by

$$IN_{\alpha(t),\beta(t)}(\mathbf{v})(x) = \alpha(t) \odot IN(\mathbf{v})(x) + \beta(t) \tag{36}$$

where $\mathbf{v}$ is an input function, $IN(\mathbf{v})$ is a regular instance normalization, and $\alpha(t)$ and $\beta(t)$ are (small) MLPs.

In all the benchmarks, the CNO architectures and training details utilized in this work are taken from the paper [13] and are as follows:

| | |
|---:|:---|
| Trainable parameters: | 39.1M |
| Number of epochs: | 400 |
| Lifting dimension: | 54 |
| Up/downsampling layers: | 4 |
| Residual blocks in the bottleneck: | 6 |
| Residual blocks in the middle layers: | 6 |
| Batch size: | 32 |
| Optimizer: | AdamW |
| Weight decay: | $10^{-6}$ |
| Initial learning rate: | $5 \cdot 10^{-4}$ |
| Scheduler: | Linear with decreasing factor of 0.9 every 10 epochs |
| Early stopping: | If the validation loss does not improve for 40 epochs |

## G.2  scOT

The scalable Operator Transformer (scOT) is a multiscale transformer model mapping discretized functions on a regular cartesian grid to discretized functions [13]. It patches the input functions and

projects these visual tokens linearly into a latent space. A SwinV2 [25] encoder-decoder structure with ConvNeXt [26] residual blocks transforms the latent tokens in a hierarchical manner. SwinV2 transformer blocks apply cosine attention not on the full token grid, but only on windows of tokens, effectively reducing the computational burden of full attention. Downsampling operations in the encoder after each level merge four tokens into one, therefore coarsen the computational grid; upsampling in the decoder reverses these operations. After the decoder, the latent tokens are mapped back linearly into the physical space.

The model accepts continuous time or lead time inputs. It is conditioned through every single layer norm module by that lead time. The conditioned layer norm $\mathcal{N}$ applied to a latent token $\mathbf{v} \in \mathbb{R}^C$ at some level is implemented as follows:

$$\mu(\mathbf{v}) = \frac{1}{C} \sum_{l=1}^{C} (\mathbf{v})_l, \quad \sigma^2(\mathbf{v}) = \frac{1}{C} \sum_{l=1}^{C} ((\mathbf{v})_l - \mu(\mathbf{v}))^2 \tag{37}$$

$$\mathcal{N}(\mathbf{v}, t) = \alpha(t) \odot \frac{\mathbf{v} - \mu(\mathbf{v}) \cdot \mathbf{1}_C}{\sigma^2(\mathbf{v})} + \beta(t) \tag{38}$$

with lead time $t \in \mathbb{R}_{\geq 0}$, $\mathbf{1}_C \in \mathbb{R}^C$ a vector of ones, and $\alpha(t)$, $\beta(t)$ being learnable affine maps from lead time to gain and bias.

As with CNO, we utilize the model and training setup from [13]. For the sake of completeness, they are given as follows:

| | |
|---:|:---|
| Trainable parameters: | 40M |
| Patch size: | 4 |
| Embedding/latent dimension: | 48 |
| Window size: | 16 |
| Number of levels: | 4 |
| Number of SwinV2 blocks at each level: | 8 |
| Number of attention heads in each level: | $[3, 6, 12, 24]$ |
| Number of ConvNeXt blocks in the residual connection: | 2 |
| Optimizer: | AdamW |
| Scheduler: | Cosine Decay with linear warmup of 20 epochs |
| Maximum learning rate: | $5 \cdot 10^{-4}$ |
| Weight decay: | $10^{-6}$ |
| Batch size: | 40 |
| Number of epochs: | 400 |
| Early stopping: | If the validation loss does not improve for 40 epochs |
| Gradient clipping (maximal norm): | 5 |

## G.3 FNO

The Fourier neural operator (FNO) is a model that efficiently maps bandlimited functions to bandlimited functions via convolutions in Fourier space [20] and the fast Fourier transform (FFT). Similar to CNO, the FNO is a composition of lifting, convolution, and projection operators. The convolution operator, $\mathcal{K}_l(\phi)$, may be viewed as a multiplication of data, $v_t$, and a learnable kernel function, $R_\phi$, in Fourier space. Here, $\phi$ are learnable parameters of a neural network. The convolution operator

may therefore be expressed as,

$$(\mathcal{K}(\phi)v_l)(x) = \mathcal{F}^{-1}(R_\phi \cdot (\mathcal{F}v_t))(x), \tag{39}$$

where

$$v_{l+1}(x) = (\sigma \circ \texttt{IN})(Wv_l(x) + (\mathcal{K}(\phi)v_t)(x)), \quad 1 \le l \le L-1, \tag{40}$$

$W$ is a local integral operator and $\mathcal{F}$ is the Fourier transform, $\texttt{IN}$ is a (lead time-conditioned) instance normalization. For efficiency, the Fourier transform is implemented via the fast Fourier transform (FFT) with a computational complexity of $O(n \log n)$, where $n$ is the number of points in the discretized mesh. Likewise, a low-pass filter removes high-frequency components of the Fourier transform, such that the weight tensor $R_\phi$ contains only $k_{max} < n$ modes. Because $k_{max}$ is independent of $n$, generally taken as a hyperparameter between $8$ and $32$, the inner multiplication has a complexity of $O(k_{max})$. As with CNO, we condition the model on lead time through instance normalizations.

The same setup as in [13] is used for this work:

|  |  |
|---:|:---|
| Trainable parameters: | 37M |
| Lifting dimension: | 96 |
| Number of Fourier layers: | 5 |
| Number of Fourier modes ($k_{max}$): | 20 |
| Optimizer: | AdamW |
| Scheduler: | Cosine Decay |
| Initial learning rate: | $5 \cdot 10^{-4}$ |
| Weight decay: | $10^{-6}$ |
| Number of epochs: | 400 |
| Batch size: | 40 |
| Early stopping: | If the validation loss does not improve for 40 epochs |

### G.4  Geo-FNO and FNO DSE

The FFT is limited to equispaced rectangular grids, and this restriction is inherited by the FNO. To apply the FNO to general geometries, the *geometry-aware Fourier neural operator* (Geo-FNO) [19] learns diffeomorphic deformations between general point clouds and such rectangular domains. Let $D_a$ be the physical domain, $D^c$ the computational domain, $x \in D_a$, and $\xi \in D^c$ be the corresponding mesh points, $\mathcal{T}_a = \{x^{(i)}\} \subset D_a$ the input meshes, $\mathcal{T}^c = \{\xi^{(i)}\} \subset D^c$ the computational meshes. The diffeomorphism $\phi_a$ transforms the points from the computational mesh to the physical mesh by

$$\phi_a : D^c \to D_a,$$
$$\xi \mapsto x, \tag{41}$$

with inverse $\phi_a^{-1}$ and both $\phi_a$ and $\phi_a^{-1}$ are smooth. The Fourier transform is subsequently approximated as

$$(\mathcal{F}_a v(k)) \approx \frac{1}{|\mathcal{T}_a|} \sum_{x \in \mathcal{T}_a} m(x)v(x)e^{2i\pi\langle\phi_a^{-1}(x),k\rangle}, \tag{42}$$

where $m(x)$ is a weight matrix based on the distribution of points from the input mesh.

In contrast, the *FNO with discrete spectral evaluations* (FNO DSE) [24] formulates a band-limited discrete Fourier transform based on the input mesh. Based on the observation that the number of Fourier modes employed by the FNO, $k_{max}$, is small and independent of the number of mesh points, $n$, a direct matrix-vector multiplication can be performed with $O(k_{max}n)$ complexity. For small

$k_{max}$, this operation is more efficient than FFT. The discrete Fourier transform matrix for mesh points $\mathcal{T}_a = \{x^{(k)}\} \subset D_a$ is constructed by

$$\mathbf{V}_{k,l} = \frac{1}{\sqrt{n}} \left[ e^{-2\pi i \left( k x^{(l)} \right)} \right]_{k,l=0}^{k_{max}-1,n-1}. \tag{43}$$

Using this matrix as a basis for the bandlimited Fourier transform, the FNO DSE avoids the construction of a computational mesh. Instead, it works directly on the physical mesh.

In all the benchmarks, the Geo-FNO and FNO DSE architectures and model hyperparameters are taken from [24] and fixed for all experiments as follows:

| | |
|---|---|
| Trainable parameters: | 3.68M (Geo-FNO), 3.30 (FNO DSE) |
| Number of epochs: | 1000 |
| Lifting dimension: | 48 |
| Number of Fourier modes $(k_{max})$: | 14 |
| Number of Fourier layers: | 4 |
| Computational mesh size (Geo-FNO): | $40 \times 40$ |
| Batch size: | 16 |
| Optimizer: | Adam |
| Weight decay: | $10^{-5}$ |
| Initial learning rate: | $5 \cdot 10^{-4}$ |
| Scheduler: | Linear with decreasing factor of 0.9 every 10 epochs |
| Early stopping: | If the validation loss does not improve for 40 epochs |

Based on the works [24, 19], two sets of favorable training hyperparameters were tested for each experiment. The results of the best model are reported. The first set of hyperparameters is:

| | |
|---|---|
| Initial learning rate: | $1 \cdot 10^{-3}$ |
| Scheduler: | Linear with decreasing factor of 0.5 every 50 epochs |

The second set is:

| | |
|---|---|
| Initial learning rate: | $5 \cdot 10^{-3}$ |
| Scheduler: | Linear with decreasing factor of 0.97 every 10 epochs |

### G.5 GINO

*Geometry-Informed Neural Operator* (GINO) [23] is composed of three main operators:

$$\mathcal{G} = \mathcal{E} \circ \mathcal{P} \circ \mathcal{D}, \tag{44}$$

where $\mathcal{E}$ is the encoder, $\mathcal{P}$ is the processor, and $\mathcal{D}$ is the decoder. Specifically, $\mathcal{E}$ maps the input data from an unstructured mesh to a uniform latent grid using a graph neural operator (GNO). $\mathcal{P}$ processes the latent-grid features via the FNO. $\mathcal{D}$ maps the latent features back to the desired output mesh (or query points) using GNO.

GNO layers approximate an integral operator on a local neighborhood graph. Given a ball of radius $r > 0$ around each point $x \in D$,

$$v_l(x) = \int_{B_r(x)} \kappa(x, y) \, v_{l-1}(y) \, \mathrm{d}y, \tag{45}$$

we discretize the integral by selecting $M$ neighbors, $\{y_1, \ldots, y_M\} \subset B_r(x)$, and form a Riemann sum:

$$v_l(x) \approx \sum_{i=1}^{M} \kappa(x, y_i) \, v_{l-1}(y_i) \, \mu(y_i), \qquad (46)$$

where $\mu(\cdot)$ represents the local weights for integration. These localized integral updates are performed across a graph that connects each node $x$ to its neighbors in $B_r(x)$.

In all the benchmarks, the GINO architectures and model hyperparameters are taken from [23] and fixed for all experiments as follows:

| | |
|---:|:---|
| Trainable parameters: | 3.7M |
| Number of epochs: | 500 |
| Radius for GNO layers: | 0.033 |
| GNO transform type: | Linear |
| Projection channels: | 256 |
| Lifting channels: | 256 |
| FNO input channels: | 64 |
| Number Fourier of modes: | 16 |
| Number of Fourier layers: | 3 |
| Optimizer: | Adam |
| Batch size: | 64 |
| Initial learning rate: | $1 \cdot 10^{-3}$ |
| Scheduler: | Linear decay with factor of 0.8 every 100 epochs |

### G.6 MeshGraphNet

The MeshGraphNet [33] architecture is an end-to-end graph-based neural operator that performs message passing directly on the given coordinates (physical nodes). Since the architecture shares many similarities to RIGNO, we can obtain a MeshGraphNet by making a few modifications to our architecture. We consider RIGNO-12 and perform the downscaling (encoder) and upscaling (decoder) processes by considering very small overlap factors, i.e., very small support radii. Regional nodes lose their meaning since each of them is only connected to one corresponding physical node. Removing the multi-scale edges in this architecture leads to a MeshGraphNet with 1.9M parameters. We train this MeshGraphNet with edge masking to be more consistent with RIGNO but use a larger batch size (64) to compensate for the extra training time due to removing the regional nodes. All other hyperparameters and settings are the same as reported in Section E.

### G.7 UPT

The *Universal Physics Transformer* (UPT) is a unified neural operator framework designed for flexible and scalable learning across a wide range of spatio-temporal problems, including both Eulerian (grid-based) and Lagrangian (particle-based) simulations [1]. UPT consists of three components: an encoder that compresses the input into a latent representation, a transformer-based approximator that propagates dynamics entirely in the latent space, and a decoder that queries this representation at arbitrary spatial locations. The model is conditioned on time and boundary conditions via DiT-style feature modulation.

We adopt a lightweight configuration of UPT in our experiments. Specifically, we use 64 latent tokens and 64 hidden dimensions. For time-dependent datasets, this results in 2.18M trainable parameters, and for time-independent datasets (without time conditioning), the number of parameters is 0.74M. A larger variant with 256 latent tokens and 192 hidden dimensions with 10.7M parameters has also been considered. We found that the large UPT strongly suffers from optimization difficulties on the datasets considered in this work, leading to higher errors than the ones reported in Table 1 with all

datasets. In all the presented benchmarks, UPT architecture and model hyperparameters are fixed for all experiments as follows:

| | |
|---:|:---|
| Trainable parameters: | 2.18M or 0.74M (see above) |
| Number of epochs: | 500 |
| Number of latent tokens: | 64 |
| Hidden dimensions: | 64 |
| Number of supernodes: | 2048 |
| Encoder and decoder depth: | 4 layers each |
| Approximator depth: | 4 layers |
| Number of attention heads: | 4 |
| Optimizer: | AdamW |
| Learning rate: | $1 \cdot 10^{-3}$ |
| Scheduler: | Cosine decay |
| Batch size: | 32 |
| Early stopping: | If validation loss does not improve for 40 epochs |

### G.8 Transolver

Transolver [45] is an attention-based neural operator framework that learns global latent tokens from input features and processes them through self-attention mechanisms. In our experiments, we observed that this architecture scales poorly to large-scale datasets. This limitation arises from its use of global interactions implemented as independent MLPs between input points and sparse tokens. Consequently, a very large number of MLP evaluations are required, quickly exhausting GPU memory. For instance, on an RTX 4090 GPU, the maximum feasible Transolver model size during training is approximately 0.5M parameters. For time-independent datasets, we were able to increase the model size to 0.9M parameters. The corresponding results are summarized in Table G.1. While both RIGNO variants exhibit clear advantages on time-dependent problems, the accuracy is comparable for elliptic problems.

Table G.1: RIGNO and Transolver benchmarks on datasets with unstructured space discretizations. All training datasets have 1024 trajectories except for the AF dataset (2048).

| Dataset | Median relative $L^1$ error [%] | | |
|:---:|:---:|:---:|:---:|
| | **RIGNO-18** | **RIGNO-12** | **Transolver** |
| Wave-C-Sines | 5.35 | 6.25 | 27.7 |
| CE-RP | 3.98 | 4.92 | 17.8 |
| Poisson-Gauss | 2.26 | 2.52 | 2.02 |
| AF | 1.00 | 1.09 | 1.21 |
| Elasticity | 4.31 | 4.63 | 4.92 |

## H Visualizations

Visual inspection allows getting more insights about several aspects of the performance of RIGNO. Estimates and statistics of ensembles of estimates are visualized in this section for all datasets. All architecture hyperparameters and training details correspond to Section E. All the presented samples are from the test datasets, and have been obtained by applying the operator autoregressively with $\tau_{\max} = 2\Delta t$ (where applicable).

Model estimates and their absolute errors, along with model input and the ground-truth output are visualized in Figures H.1-H.21. Visual inspection of model estimates for the CE-Gauss and CE-RP

datasets reveals that there are two main reasons that explain the high errors for these datasets. First, model estimates are generally smoother than the ground-truth solutions. This inability to produce high-frequency estimates is reflected in the high errors where vortices are present. This is evident in the high-frequency details of velocity components in these model estimates. Second, although the model is able to capture shocks quite well at the right locations, it generates a smoother transition with smaller gradients. This is very well witnessed in the absolute estimate errors of Figure H.19, where the highest errors are located in small neighborhoods around shocks. The statistics of ensembles of estimates are visualized in Figures H.22-H.25. Estimate ensembles are generated by unrolling the operator randomly 20 times. Because of random edge masking, the estimates will slightly be different every time. The statistics of the estimate ensembles can act as indicators for model uncertainty. See Section F.6 for a broader discussion.

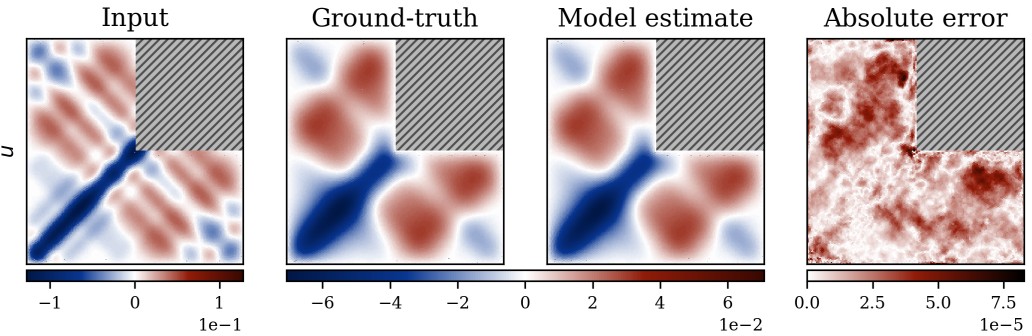

Figure H.1: Estimation of RIGNO-18 for a random test sample of the Heat-L-Sines dataset at $t_{14}$.

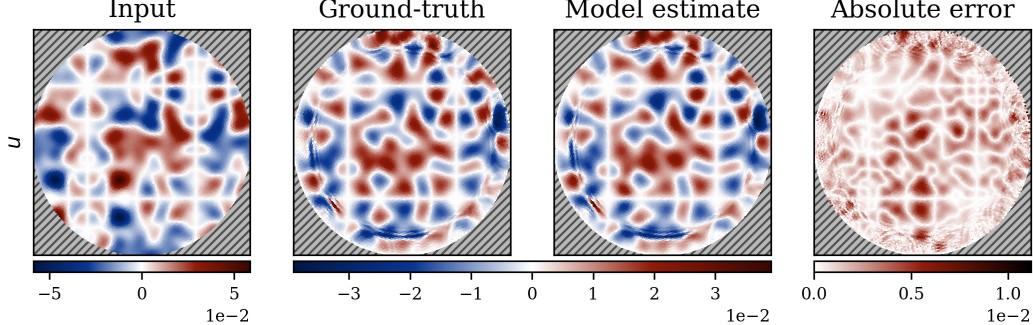

Figure H.2: Estimation of RIGNO-18 for a random test sample of the Wave-C-Sines dataset at $t_{14}$.

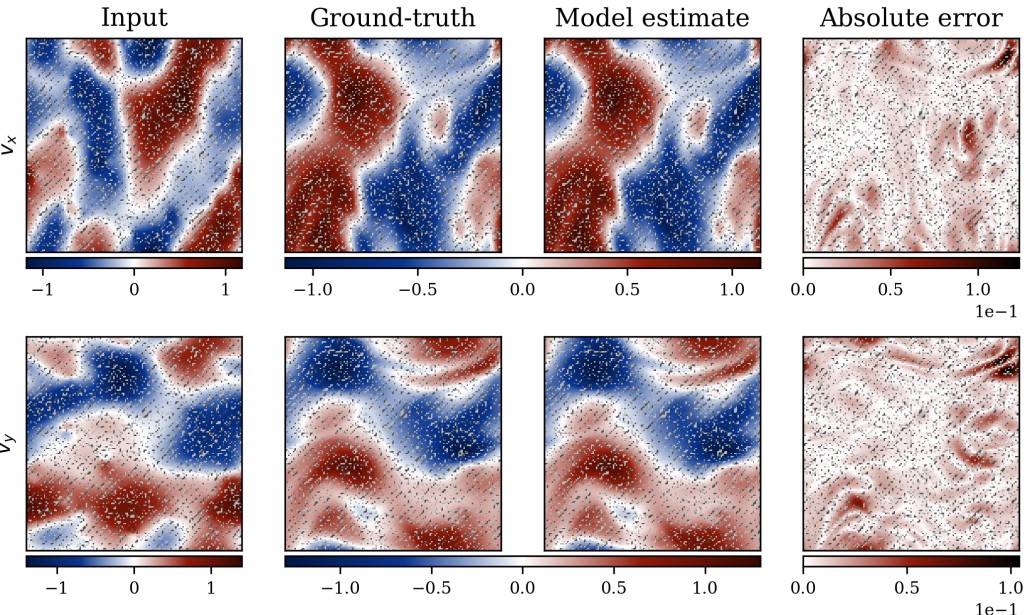

Figure H.3: Estimation of RIGNO-18 for a random test sample of the unstructured NS-Gauss dataset at $t_{14}$.

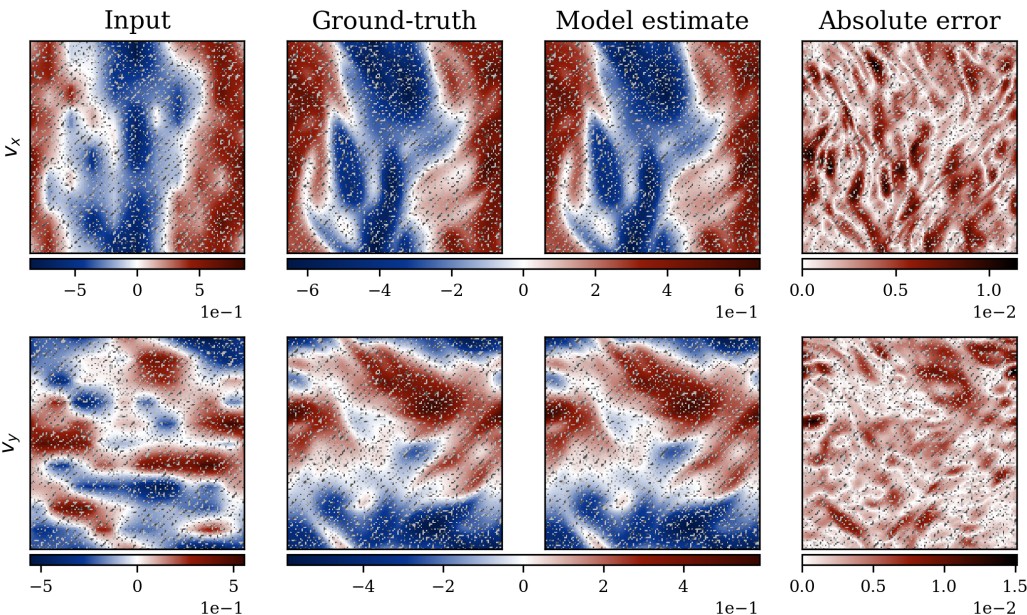

Figure H.4: Estimation of RIGNO-18 for a random test sample of the unstructured NS-PwC dataset at $t_{14}$.

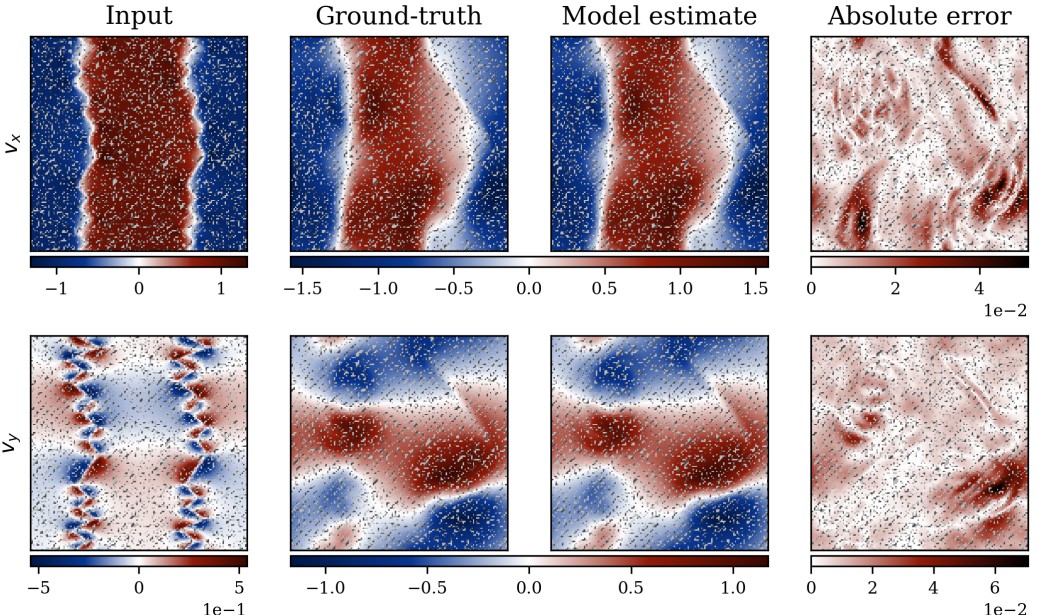

Figure H.5: Estimation of RIGNO-18 for a random test sample of the unstructured NS-SL dataset at $t_{14}$.

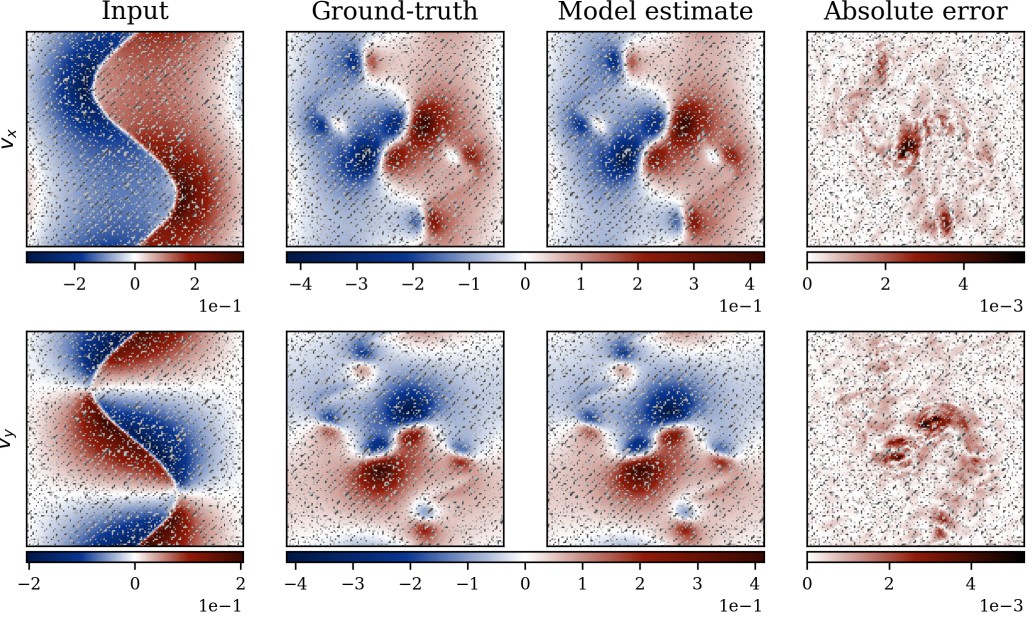

Figure H.6: Estimation of RIGNO-18 for a random test sample of the unstructured NS-SVS dataset at $t_{14}$.

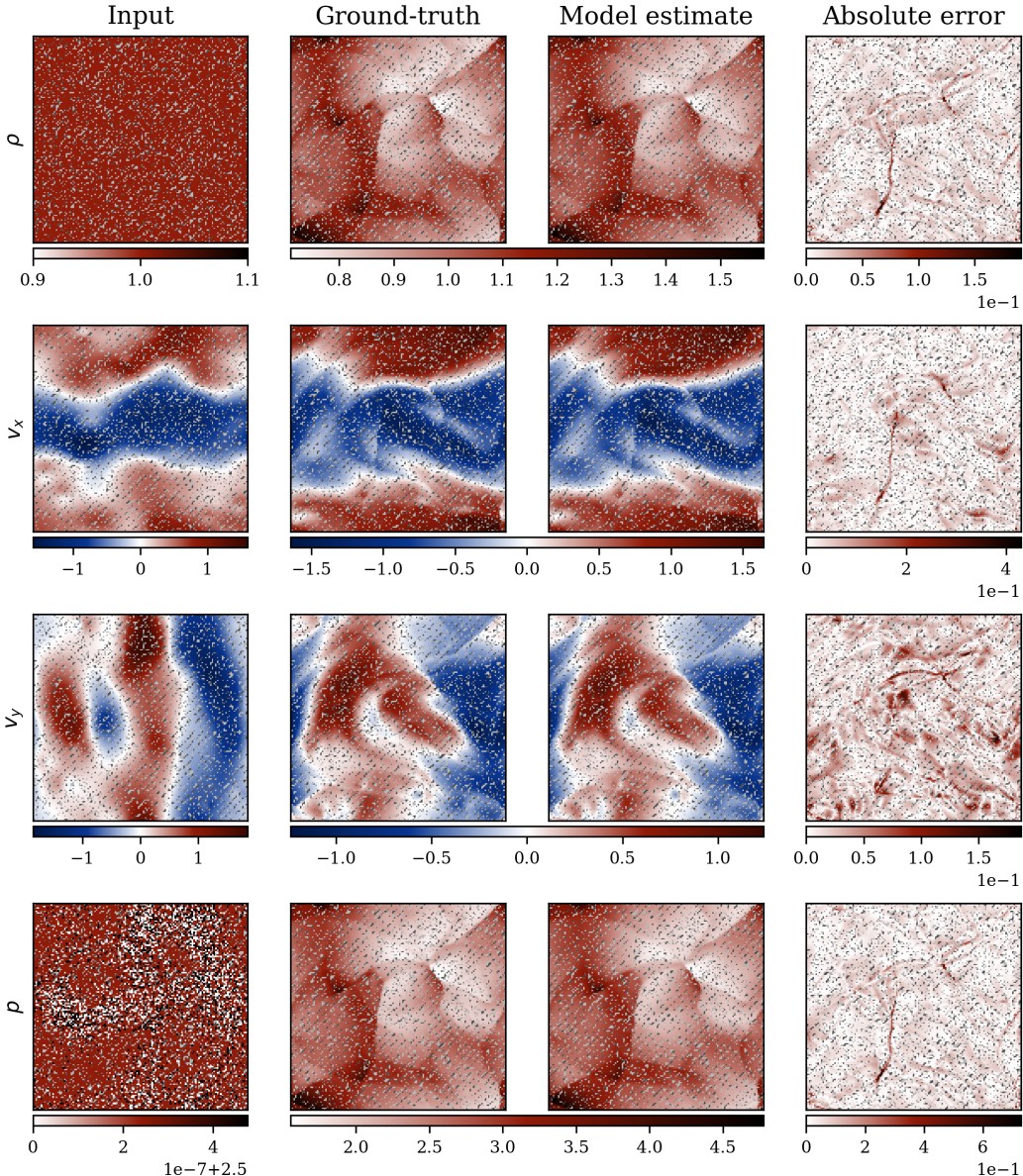

Figure H.7: Estimation of RIGNO-18 for a random test sample of the unstructured CE-Gauss dataset at $t_{14}$.

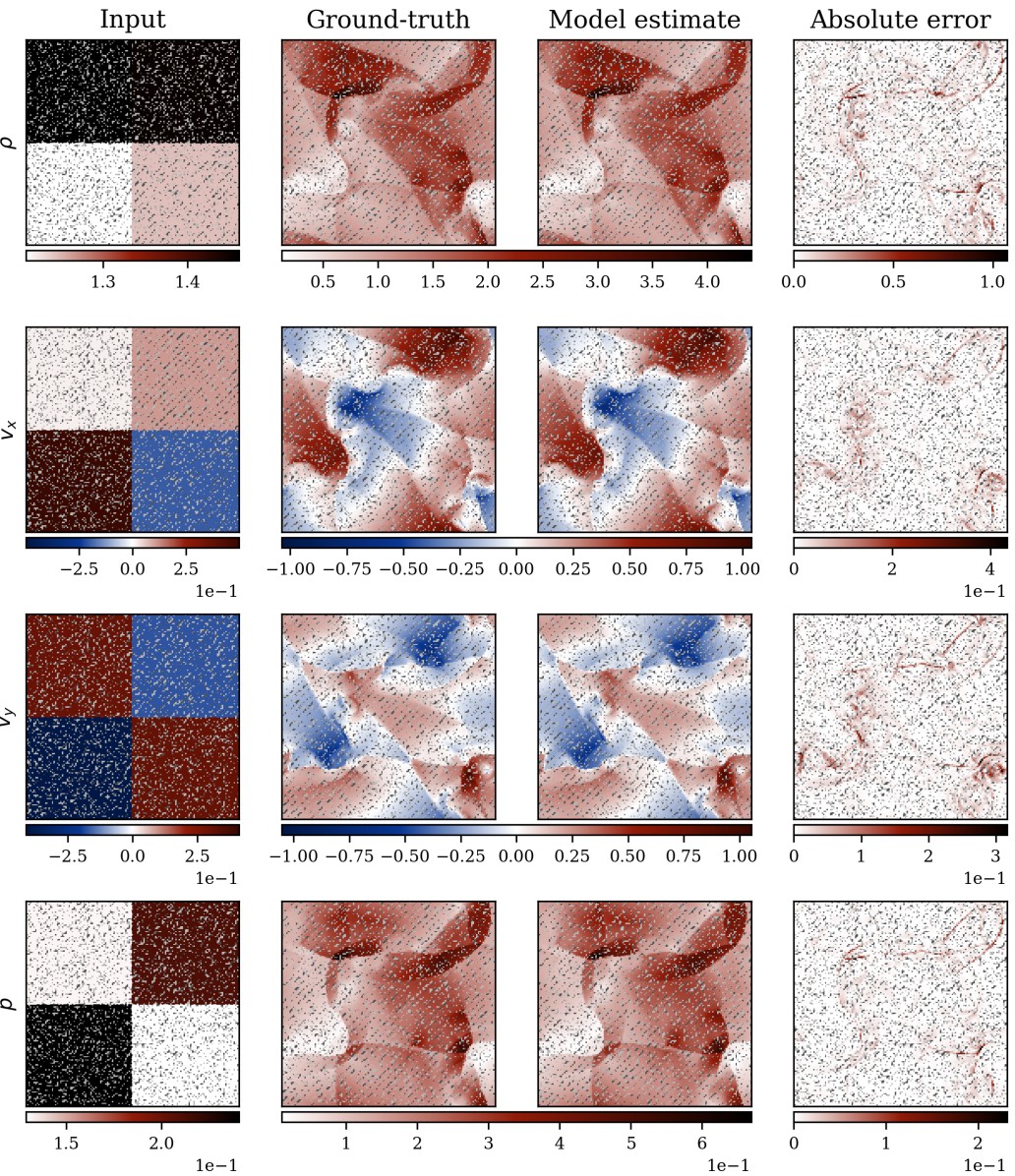

Figure H.8: Estimation of RIGNO-18 for a random test sample of the unstructured CE-RP dataset at $t_{14}$.

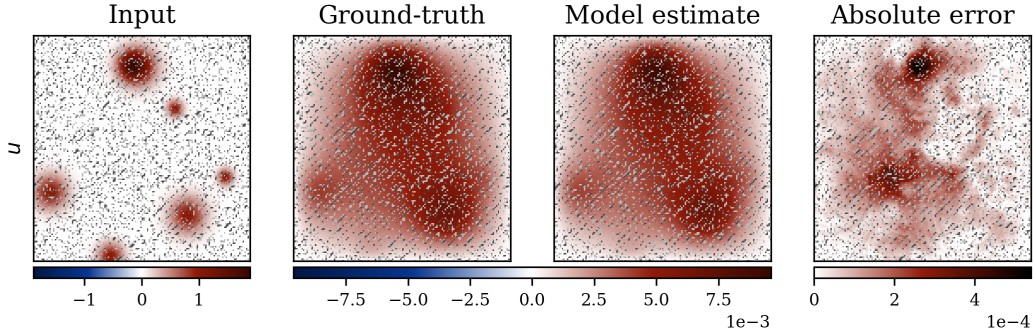

Figure H.9: Estimation of RIGNO-12 for a random test sample of the unstructured Poisson-Gauss dataset.

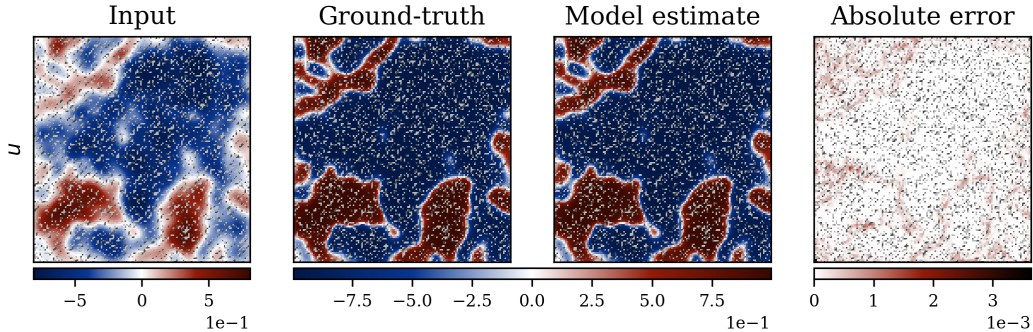

Figure H.10: Estimation of RIGNO-18 for a random test sample of the unstructured ACE dataset at $t_{14}$.

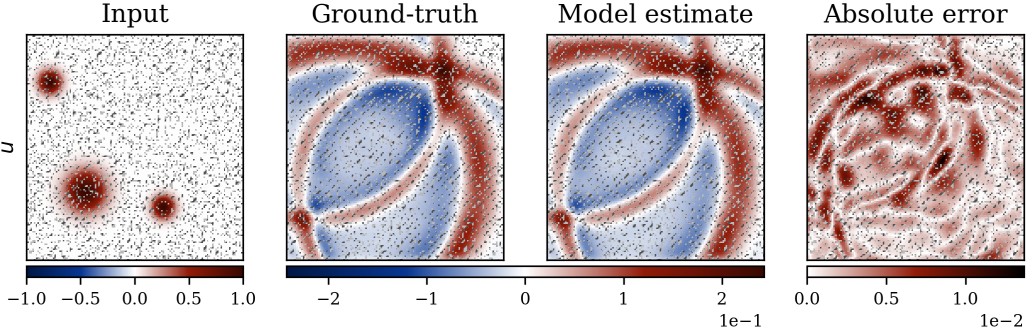

Figure H.11: Estimation of RIGNO-18 for a random test sample of the unstructured Wave-Layer dataset at $t_{14}$.

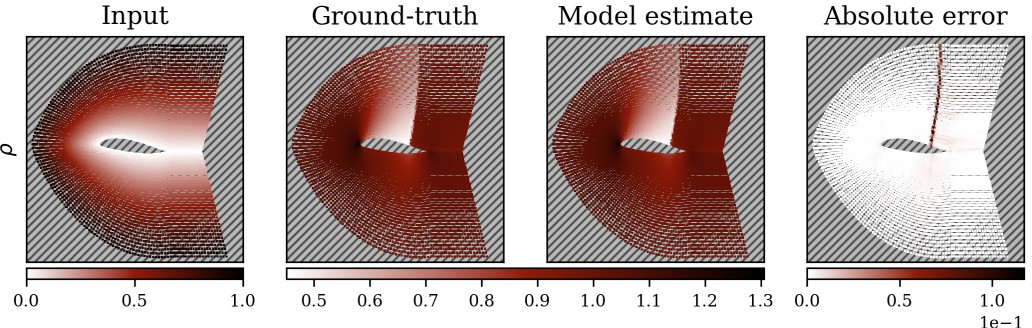

Figure H.12: Estimation of RIGNO-18 for a random test sample of the AF dataset.

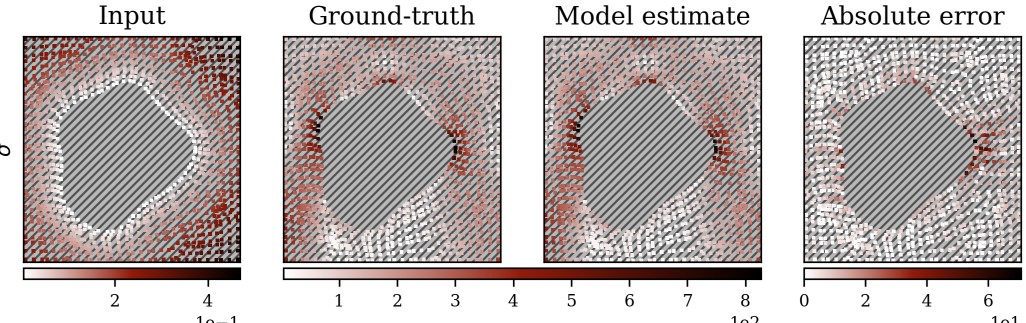

Figure H.13: Estimation of RIGNO-18 for a random test sample of the Elasticity dataset.

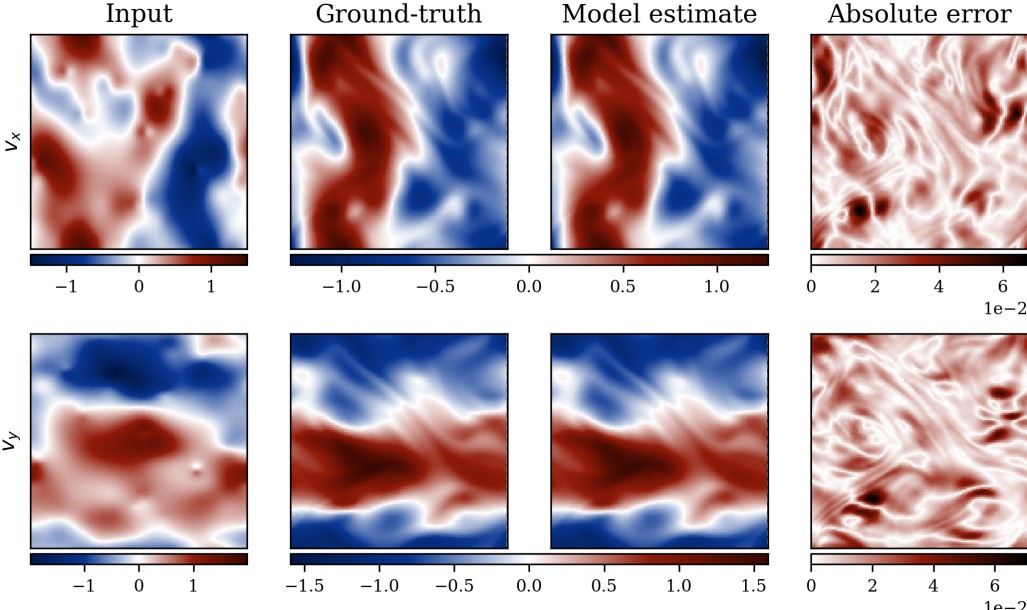

Figure H.14: Estimation of RIGNO-18 for a random test sample of the NS-Gauss dataset at $t_{14}$.

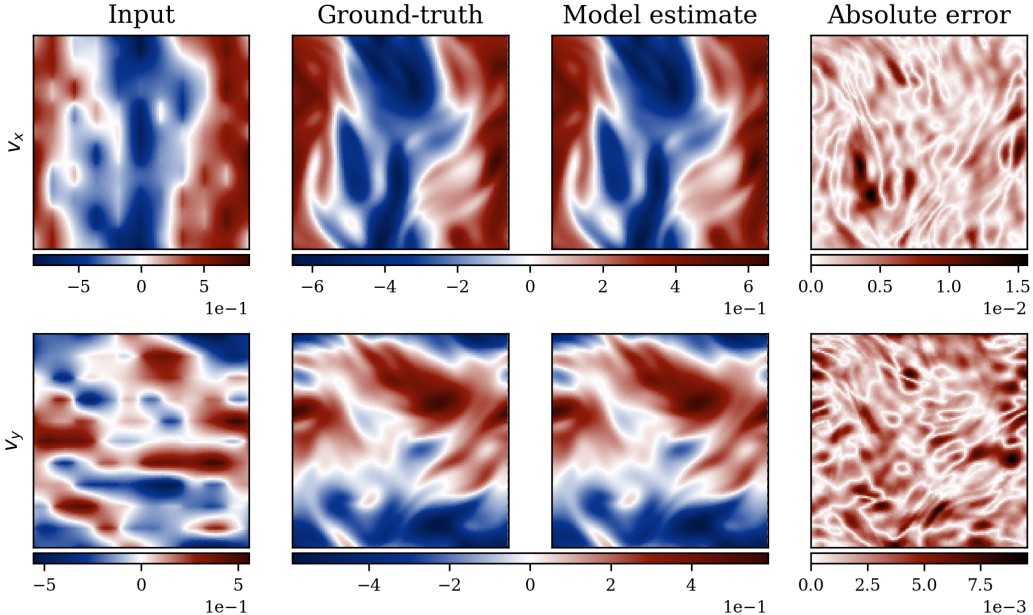

Figure H.15: Estimation of RIGNO-18 for a random test sample of the NS-PwC dataset at $t_{14}$.

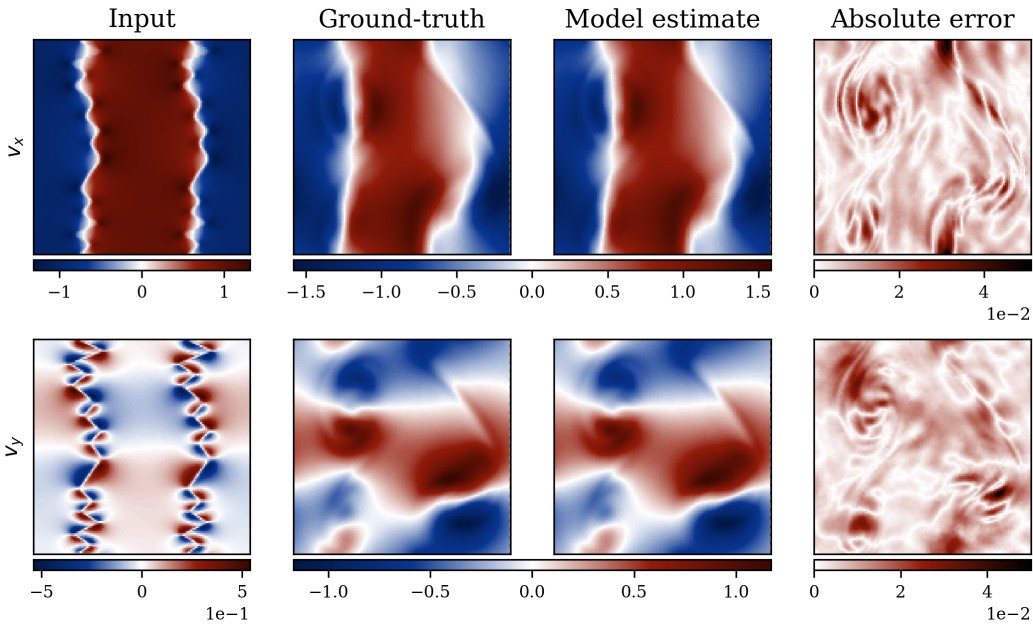

Figure H.16: Estimation of RIGNO-18 for a random test sample of the NS-SL dataset at $t_{14}$.

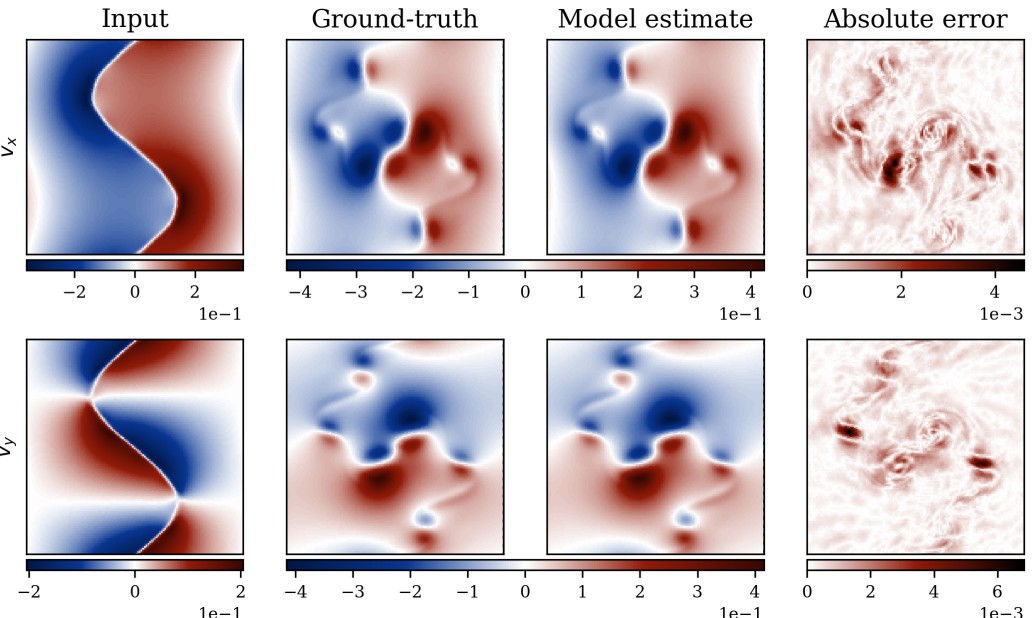

Figure H.17: Estimation of RIGNO-18 for a random test sample of the NS-SVS dataset at $t_{14}$.

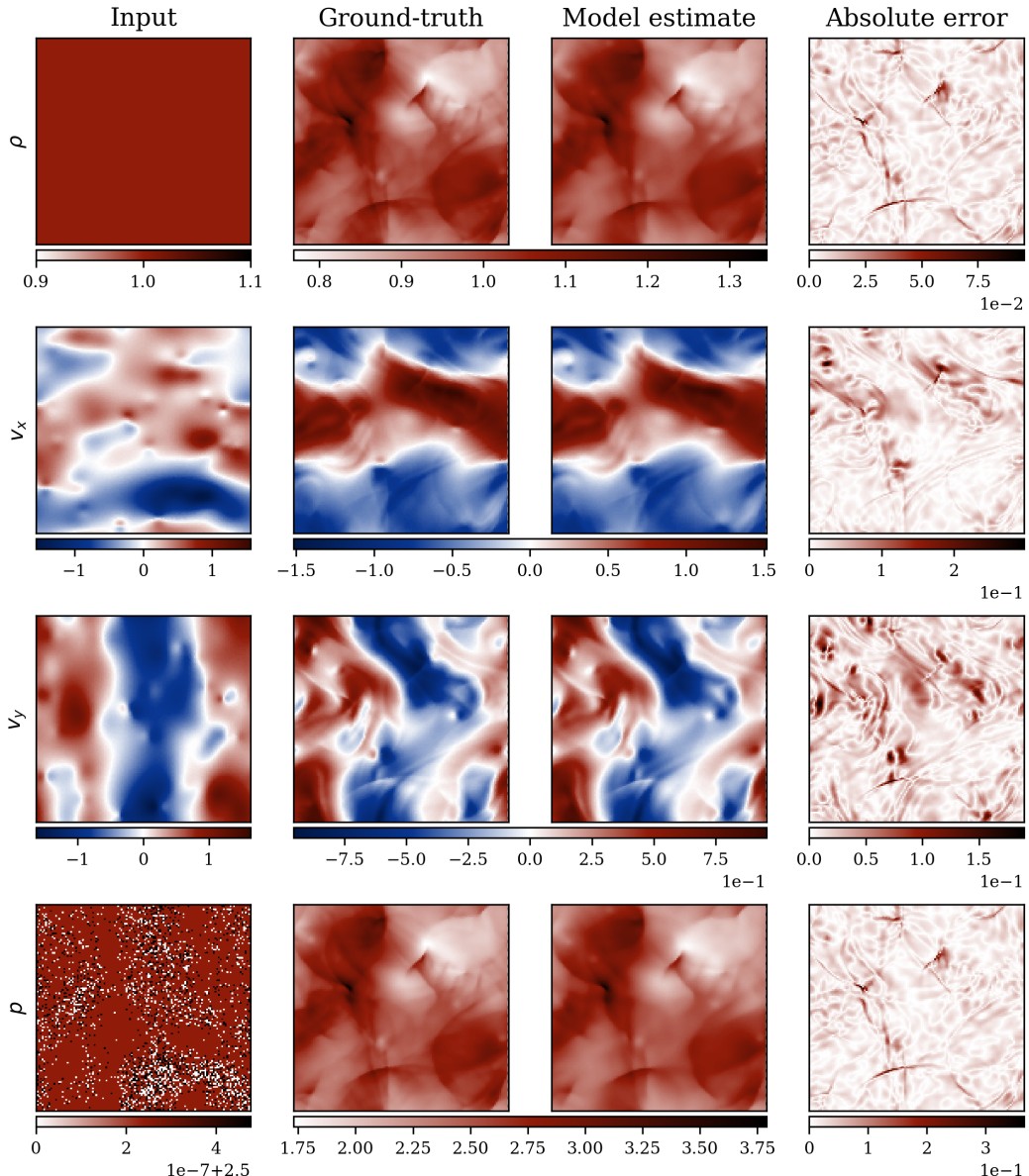

Figure H.18: Estimation of RIGNO-18 for a random test sample of the CE-Gauss dataset at $t_{14}$.

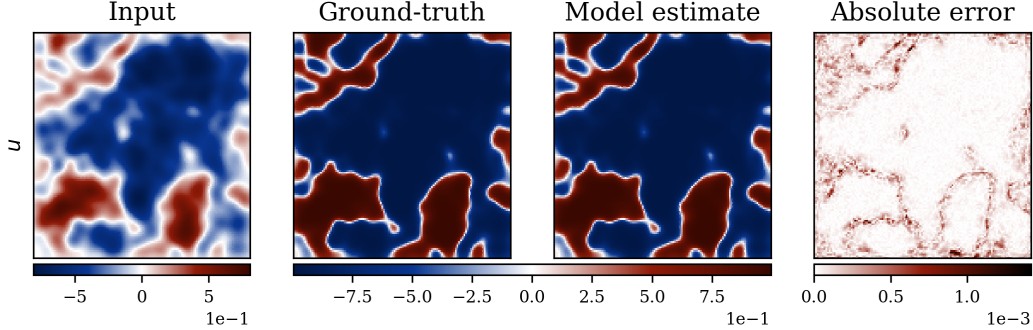

Figure H.19: Estimation of RIGNO-18 for a random test sample of the CE-RP dataset at $t_{14}$.

Figure H.20: Estimation of RIGNO-18 for a random test sample of the ACE dataset at $t_{14}$.

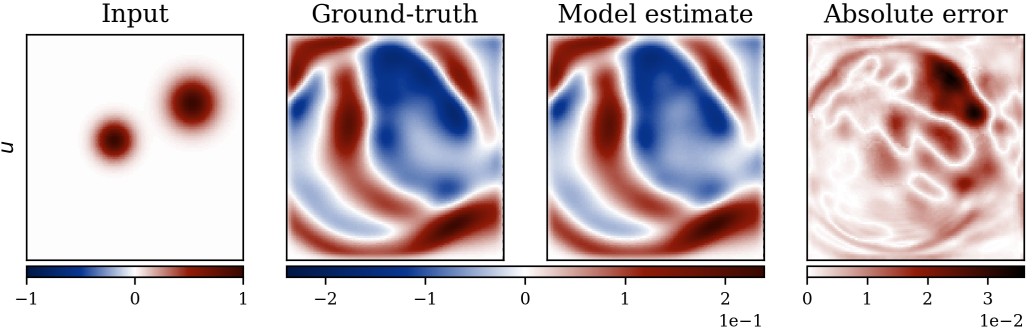

Figure H.21: Estimation of RIGNO-18 for a random test sample of the Wave-Layer dataset at $t_{14}$.

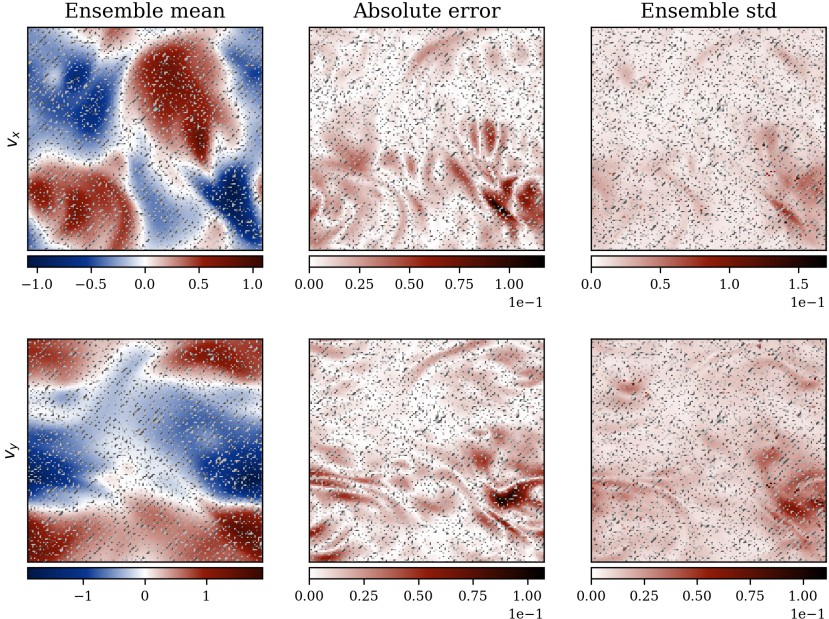

Figure H.22: Model uncertainty with RIGNO-18 for a random test sample of the NS-Gauss dataset.

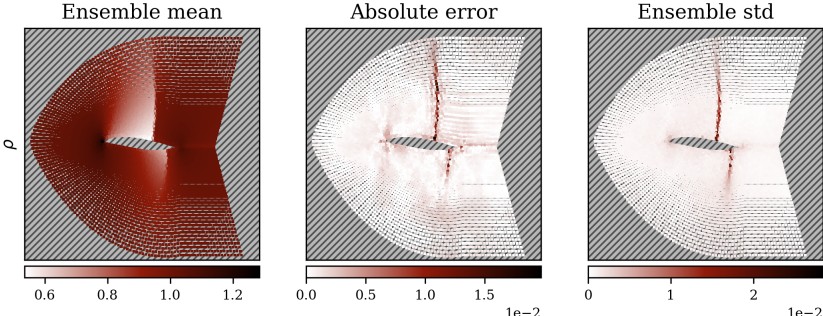

Figure H.23: Model uncertainty with RIGNO-18 for a random test sample of the AF dataset.

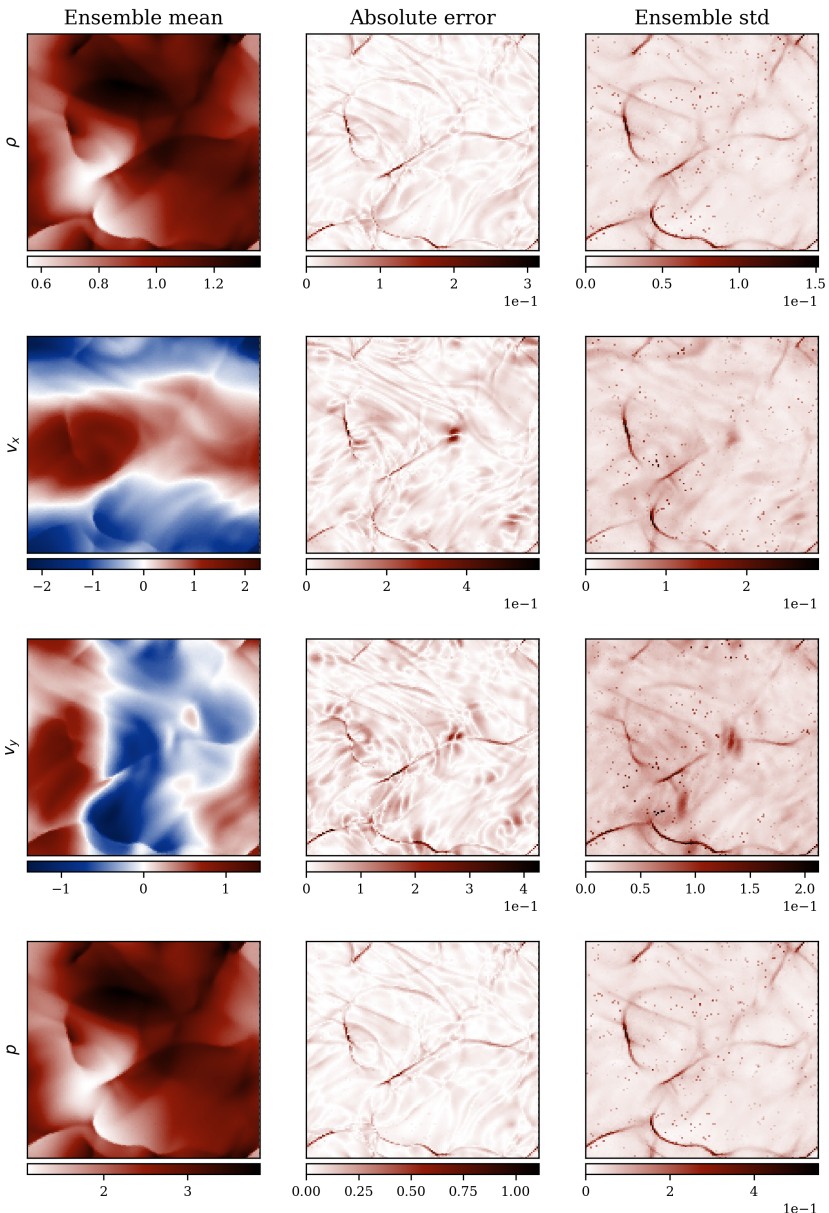

Figure H.24: Model uncertainty with RIGNO-18 for a random test sample of the CE-Gauss dataset.

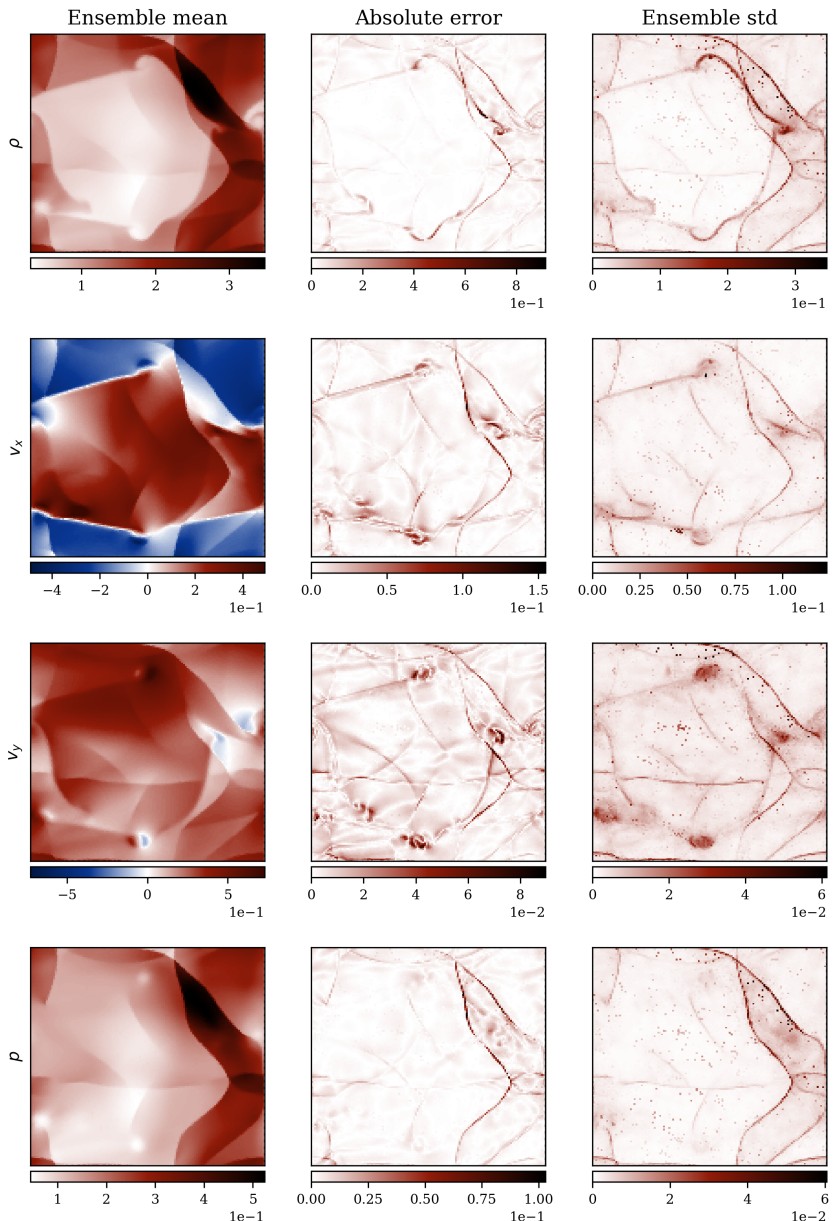

Figure H.25: Model uncertainty with RIGNO-18 for a random test sample of the CE-RP dataset.

