# OpenReview forum: "RIGNO: A Graph-based Framework For Robust And Accurate Operator Learning For PDEs On Arbitrary Domains"
_NeurIPS.cc/2025/Conference — NeurIPS 2025 poster_

### Official Review · Reviewer_AxpQ · 2025-06-30

**Clarity:** 2
**Significance:** 2
**Originality:** 2
**Rating:** 4
**Confidence:** 3

**Summary:**

This paper presents RIGNO, a new graph-based neural operator framework for solving PDEs on arbitrary domains. RIGNO proposes a new approach for encoding sets of points via a regional mesh approach, which enable to compress the physical information. It then uses a message passing architecture to predict future time-steps via an all2all training approach. This allows to generalize to different timestep intervals. To enable continuous timestep evaluation, they propose temporal fractional pairing to allow generalization to intermediate timesteps not accessible in the training set. Their approach is evaluated on a range of PDEs and exhibit competitive or sota performance.

**Questions:**

Did you try to test your model on out-of-domain data, e.g. on new PDE coefficients? As you give the pde coefficients as input, it could be interesting to check how the model behaves on unseen coefficients.

In the all2all training, you only provide future timesteps to predict right? Could the model actually learns to solve an inverse problem ? e.g. predicting the past frames?

**Ethical Concerns:**

["NO or VERY MINOR ethics concerns only"]

**Final Justification:**

Most of my concerns have been addressed by the authors. Thus, I have upgraded my score to a borderline accept. I think the novelty of the paper is limited, but the provided experiments are consequent.

- the authors provided the computational efficiency and acknowledge it as a limitation of the paper.
- the INR literature will be discussed and CORAL has been compared to RIGNO.
- they did out-horizon experiments for OOD.

They proposed to do OOD evaluation on unseen PDE parameters. I think it could be valuable.

**Limitations:**

Limitations have been mentioned. Negative societal impacts have not been mentioned in the work.

**Paper Formatting Concerns:**

No major formatting issues.

**Quality:**

3

**Strengths And Weaknesses:**

Strengths:

I think the paper is clearly written, in particular, there are very nice illustrations in appendix that clearly help for the comprehension of the paper.

The method has been evaluated on a large range of PDEs with substantial experiments. On all datasets, Rigno-18 outperforms the considered baselines. the proposed framework is a simple but effective encode-process-decode approach, which allows to process arbitrary geometries.

I think the paper also tackles an important problem: generalization on arbitrary geometries. The authors could other important works that differ from graphs or transformers such as INRs, which is another direction for learning on arbitrary geometries [1, 2].

I also appreciated the ablations studies and additional experiments provided, notably for the time processing component.

Weaknesses:

Regarding the novelty of the paper, the method is relatively simple and follows a similar direction than existing approaches. It follows the encode-process-decode approach that is particularly effective for graph-based neural operators and make use of the training strategy proposed in Poseidon. Their contributions include the edge masking and the Fractional pairing strategy which are relatively simple. However, they provide ablations showing the advantage of their incorporation.

For the ablations on the dataset and model size, it would have been nice to test until when model size or data size reaches a plateau. It has been on relatively small/medium setups only.

There are other important baselines that could have been used such as [1] and [3] for instance.

As you mentioned in the introduction, efficiency is a key property for learning efficient neural operators on arbitrary geometries. Graphs are known to be particularly costly. It is important to add additional experiments to show how RIGNO behaves in terms of computation with respect to considered baselines.

[1] Serrano et al. Operator Learning with Neural Fields: Tackling PDEs on General Geometries, NeurIPS 2023.

[2] Wang et al. GridMix: Exploring Spatial Modulation for Neural Fields in PDE Modeling, ICLR 2025.

[3] Wu et al. Transolver: A Fast Transformer Solver for PDEs on General Geometries, ICML 2024.

---

> ### Author Rebuttal · Authors · 2025-07-30
>
> We would like to thank the reviewer for their excellent and constructive suggestions. We hope to address all their concerns in detail below.
>
> 1. **Novelty and impact**:
> We appreciate the reviewer’s observation that the proposed contributions, edge masking and fractional pairing, are grounded in simple and intuitive ideas. Nonetheless, these play a critical role in RIGNO's ability to generalize to unseen spatial and temporal resolutions. We see the relative simplicity of these approaches as an advantage, as these methods are easily transferable across frameworks. Thus, this work contributes a broader impact in the operator learning community than a standalone neural operator. For example, edge masking can be readily integrated into any GNN-based PDE solver, while fractional pairing offers a lightweight and practical strategy for addressing generalization across temporal resolutions in **any** operator learning framework.
>
> 2. **Dataset and model scaling experiments:**
> Performance plateaus can arise from either a large generalization gap (i.e., insufficient data) or limited model expressivity (i.e., small model capacity). In Figure 2, our primary aim is to show that both issues can be effectively mitigated within the proposed framework; by increasing the amount of data in the former case, and scaling the model size in the latter. Specifically, we used 1024 training samples for the time-independent datasets and 1024 trajectories for the time-dependent ones (7168 input-output pairs). For single-task operator learning, this represents a relatively data-rich regime, especially considering the high resolution of the input/output functions. In this setting, we do not expect to observe a plateau unless the model becomes exceedingly large. To further support this within the limited time of the rebuttal period, we conducted a small-scale experiment using only 128 samples/trajectories from two datasets. The results (see table below) indicate that although the training error continues to decrease (overfit), the test error plateaus around 1.8M parameters. Linearly extrapolating this result to the training conditions of the paper (outlined in Fig. 2), suggests that a plateau would occur at approximately 16M parameters or more. As the reviewer rightfully suggests, a more extensive version of these experiments would be a valuable addition. We will add these to a camera-ready version (CRV), if accepted.
>
>
> ---
> >**Table 1: Scaling behavior of test and training errors (%) with model size for Poisson-Gauss (PG) and NS-Gauss (NS) datasets, using 128 training samples for PG and 128 trajectories for NS.**
> >
> >| Parameters [M] | PG Test | PG Train | NS Test | NS Train |
> >|:--------------:|--------:|---------:|--------:|---------:|
> >| 0.1            | 19.93   | 13.06    | 39.28   | 31.46    |
> >| 0.5            | 16.77   | 9.05     | 31.03   | 17.39    |
> >| 1.1            | 13.73   | 5.97     | 15.29   | 11.12    |
> >| 1.8            | 13.14   | 4.56     | 13.01   | 9.57     |
> >| 2.8            | 13.45   | 3.99     | 10.79   | 8.21     |
> >| 4.1            | 13.23   | 3.18     | 11.46   | 7.14     |
>
> ---
>
>
> 3. **Alternative approach:** INRs offer an alternative approach for learning on arbitrary geometries by encoding both the input and output functions into single latent feature vectors, with the mapping learned in this latent space. Given both the popularity and effectiveness of transformers and graph-based models, as well as RIGNO's position as an advancement among these methods, we have focused on these baselines. Nonetheless, we would like to point out that among the 8 models used for comparison, we have used the FNO and CNO, which are not graph or transformer based, as well as the GeoFNO and FNO DSE, two popular approaches for data on general geometries. These baselines cover graph, convolution, attention, and Fourier-based frameworks. Nonetheless, since INRs also address one of the core problems considered in RIGNO, we agree that they warrant discussion in the broader context of operator learning. We thank the reviewer for this suggestion and will include a brief discussion of INRs in the CRV, if the paper is accepted.
>
> 4. **Further benchmarks (Transolver):**
> Based on the reviewer’s helpful suggestion, we evaluated Transolver as an additional baseline. However, we found that this model scales poorly, severely hindering its ability to learn on large-scale datasets. This limitation stems from its use of global interactions, implemented via independent MLPs acting between each input point and a set of sparse tokens. The resulting large number of MLP evaluations leads to substantial memory consumption, limiting the model size to at most 0.5M parameters on an RTX 4090 GPU. We trained Transolver on two representative datasets, CE-RP and Wave-C-Sines, and observed relatively high test errors of 17.8% and 27.7%, respectively. These are substantially higher than the corresponding RIGNO-12 errors of 4.92% and 6.25%, achieved under similar computational constraints. We thank the reviewer for this valuable suggestion and will include Transolver results for all datasets in the CRV, if the paper is accepted.
>
> 5. **Comparison with CORAL:**
> Following the reviewer's suggestion, we investigated RIGNO's performance in comparison to CORAL. Given the time constraints and major differences in the training strategies, we do not train CORAL, and rather rely on the test results in Table 3 of [1]. The NACA-Euler dataset of [1] is the same airfoil dataset that we use in RIGNO (AF), with the difference that the target variable used there is the Mach number. We conducted new experiments with RIGNO with the settings of [1] (1000 training samples, 200 test samples), and used the same metric as in [1] (relative L2 error without normalization) to obtain the following results in Table 2, where we also report the test errors for the second-best and third-best baselines in [1]. Based on the results in Table 1 of the submission, RIGNO's accuracy is most striking for hyperbolic and parabolic problems, and does not have a large margin with the other baselines on elliptic problems. Nonetheless, RIGNO-18 achieves 0.35% error on 200 test samples, outperforming CORAL (0.59%) and the other baselines considered in [1] by a considerable margin.
>
> ---
> >**Table 2: Relative L2 errors (\%) without normalization on 200 test samples. All models are trained on 1000 samples.**
> >
> >|  | NACA-Euler |
> >|:--------------:|-----------:|
> >| RIGNO-18       | 0.35  |
> >| RIGNO-12       | 0.40    |
> >| CORAL          | 0.59   |
> >| Factorized-FNO | 0.62   |
> >| Geo-FNO        | 1.58       |
> ---
>
>
> 6. **Further performance benchmarks:**
> We benchmarked GINO, RIGNO, and MeshGraphNet [1] in terms of training and inference times across various model configurations. The results indicate that RIGNO’s training times are consistently slower than those of GINO on datasets with fixed geometry and consistent input coordinates across training samples. We acknowledge this as a limitation of our model; however, since training is a one-time cost, inference speed is ultimately more critical for deployment. In this regard, RIGNO and GINO exhibit comparable inference times, while RIGNO consistently achieves higher accuracy. Moreover, unlike GINO, RIGNO supports batch processing for datasets with variable geometries (e.g., Elasticity), where it is also faster at training. Our benchmarks show that RIGNO is also consistently more memory-efficient than GINO, enabling it to be inferred with ~3x more samples. Due to space constraints, we kindly refer the reviewer to point 6 in our response to Reviewer Z5X2 for a broader discussion regarding training and inference times, and point 6 in our response to Reviewer 9Nmc regarding memory usage. We thank the reviewer for highlighting this point and will include these results, along with additional performance benchmarks, in a CRV if the paper is accepted.
>
> 7. **Out-of-domain experiments:**
> We explore out-of-domain performance in the form of solutions at **unseen time steps**. Figure 3c in the main text shows the performance of RIGNO on NS-SL for time steps 0-14, seen during training, as well as for *unseen time steps* 16-20. These later time steps have stronger vortices and different dynamics, yet the error does not diverge from the general trend, indicating strong out-of-domain performance. SM Section F.5 and SM Figure F.3 offer a broader evaluation and discussion regarding this test with the NS-PwC dataset. For variation in the parameters of the PDEs, there exist several datasets which include *out-of-distribution* test sets, available in [2]. Given the limited time of the rebuttal period, we have been unable to perform these experiments; however, we will gladly add these experiments to the CRV, if accepted.
>
> 8. **Inverse problems:**
> In all2all training, the reviewer is correct: we only provide future time steps. Neural operators can learn arbitrary maps between parameters; however, inverse maps are often ill-posed. For this reason, approaches for learning inverse problems typically use an invertible architecture, but this may impose challenges in learning expressive maps with high-frequency features [3]. We believe studying the use of neural operators for inverse problems deserves a thorough investigation beyond the scope of the current work, and we reserve this task for future work.
>
> We hope that our responses have satisfactorily addressed all of the reviewer’s comments and concerns, and we will happily provide any additional clarifications. We kindly request the reviewer to consider updating their assessment in light of these remarks.
>
> *[1] Serrano et al, Operator Learning with Neural Fields: Tackling PDEs on General Geometries. NeurIPS 2023.*
>
> *[2] Raonic et al, Convolutional Neural Operators for robust and accurate learning of PDEs. NeurIPS 2023.*
>
> *[3] Long et al, Invertible Fourier Neural Operators for Tackling Both Forward and Inverse Problems. PMLR 2024.*

---

> > ### Comment · Reviewer_AxpQ · 2025-08-03
> >
> > I would like to thanks the authors for their rebuttal. Most of my concerns have been addressed.
> >
> > Please, refer to INRs in the related works, as it is linked to your work. Concerning the experiments, it would be very nice to add the suggested experiments in the final version.
> >
> > I will upgrade my rating score to 4.

---

> > > ### Author Response · Authors · 2025-08-04
> > > **Thanking the Reviewer**
> > >
> > > We sincerely thank the reviewer for appreciating our response and for raising the score. Your suggestions will help us improve our paper.

---

### Official Review · Reviewer_9Nmc · 2025-07-02

**Clarity:** 4
**Significance:** 3
**Originality:** 2
**Rating:** 5
**Confidence:** 4

**Summary:**

This paper presents the Region Interacting Graph Neural Operator (RIGNO), a multi-scale, resolution-invariant graph neural operator that builds upon established architectural principles while introducing several interesting enhancements. The work follows a solid foundation using latent space representations and adopts the encoder-processor-decoder structure familiar from MeshGraphNet, enhanced with a novel multiscale regional mapping component.

**Questions:**

Based on my understanding, the paper proposes to use graph neural operators for the flexibility of graph. Which component from RIGNO makes it signficiantly outperform previous state-of-the-art model such as CNO and Poseidon/SCOT on regular grids? Are these components applicable to other models?

**Ethical Concerns:**

["NO or VERY MINOR ethics concerns only"]

**Limitations:**

Yes

**Quality:**

4

**Strengths And Weaknesses:**

## Strengths
The paper demonstrates several commendable qualities. The experimental evaluation is particularly thorough, encompassing a comprehensive suite of PDE test problems and comparing against a diverse set of baseline methods. The reported performance improvements are impressive and suggest the method's practical value. The incorporation of edge masking for regularization and temporal processing through adaptive layer normalization with gating mechanisms shows thoughtful engineering.

## Weaknesses and Questions
While the work shows strong empirical results, I believe it could benefit from deeper analysis in a few key areas:
- Novelty Clarification: The paper skillfully combines several existing techniques, but it would strengthen the contribution to more clearly articulate which specific innovations drive the state-of-the-art performance. A more explicit discussion of the novel aspects beyond the multiscale regional mapping would help readers understand the core contributions.
- Ablation study: The work would significantly benefit from ablation studies examining each major component's contribution to overall performance. This would help identify which design choices are most critical and provide valuable insights for future work in this area.
- Computational Efficiency: Including runtime and memory usage comparisons would provide important practical insights, especially given the multi-scale nature of the approach, which readers would find valuable for implementation considerations.

---

> ### Author Rebuttal · Authors · 2025-07-30
>
> We begin by thanking the reviewer for their thoughtful comments, constructive suggestions, and their recognition of the value of our work. We address their concerns in detail below.
>
> 1. **Contributing factors to RIGNO's performance:**
> Among the baselines presented in Table 1, most (excluding CNO and MeshGraphNet [1]) rely on global interactions; either through convolutions in the Fourier space or multi-head attention mechanisms. In contrast, we attribute the state-of-the-art accuracy and robustness of RIGNO primarily to the high expressive power of *local* message passing for fine-grained manipulation and propagation of information. This advantage is especially prominent in time-dependent and advection-dominated problems, whereas for elliptic problems (AF, Elasticity, and Poisson-Gauss), where global interactions are more crucial, the gains are less pronounced. Compared to MeshGraphNet [1], which also utilizes message passing, RIGNO introduces several key architectural components that enhance both efficiency and expressivity. The introduction of regional nodes facilitates multi-scale aggregation of input data and yields significant computational benefits (see SM Table F.2 and Point 2 in our response to Reviewer 1KFB). While we benchmarked MeshGraphNet with the same number of message-passing blocks, SM Figure F.1 indicates that such models benefit substantially from deeper message-passing pipelines. RIGNO enables this depth efficiently by performing message passing on coarse regional graphs, which are less costly to process. To mitigate the limitations of purely local interactions, RIGNO incorporates *multi-scale edges* in the processor. These long-range connections significantly enhance accuracy, without a notable increase in computational cost, as shown in the right-most plot of SM Figure F.1. We appreciate the reviewer’s suggestion to provide a broader discussion of the novel components driving RIGNO’s performance, and will include this analysis in the camera-ready version (CRV), if accepted. Below, we further elaborate on two of our key contributions, *edge masking* and *fractional pairing*, whose design and benefits will also be described in greater detail in the CRV. We emphasize that both techniques are framework-agnostic and easily transferable. Edge masking can be integrated into any GNN-based PDE solver to improve generalization across spatial resolutions. Similarly, fractional pairing provides a lightweight, effective fine-tuning strategy for mitigating temporal generalization gaps and can easily be **employed in any operator learning framework**.
>
> 2. **Intuitions behind edge masking:**
> Considering the graph structure in the encoder, a regional node receives information from multiple physical nodes. With high-resolution inputs, many of these nodes are *redundant* in the sense that the underlying function can often be accurately reconstructed (e.g., via interpolation) using fewer nodes. During training, the model may implicitly ignore some of these redundant nodes and focus on a subset of dominant or central connections. At test time with different resolutions, these dominant nodes may no longer be present, leading to degraded performance. We identify this phenomenon as a key contributor to the resolution sensitivity observed in many GNN-based operator learning models and address it with the proposed method. By randomly masking edges during training, the model cannot consistently rely on the same high-correlation connections. In the absence of these edges, the model is forced to use the weaker correlations. This principle extends naturally to both the processor and decoder stages. Additionally, learning from weaker or secondary correlations in the processor enables the model to better integrate information across multiple spatial scales, rather than overfitting to dominant patterns in the training resolution.
>
> 3. **Intuitions behind fractional pairing:**
> Similarly, the effectiveness of the fractional pairing strategy can be understood through a simple observation. In conventional training setups (including all2all pairing), the model is only exposed to lead times that are integer multiples of the dataset’s time resolution Dt. As a result, any smaller lead time (e.g., Dt/2) is out-of-distribution (OOD) for the model. During inference, this distribution shift can lead to noticeable performance degradation when the model is queried on shorter, unseen lead times. Our proposed fractional pairing fine-tuning strategy directly addresses this issue by explicitly extending the distribution of lead times seen during training. In simple terms, it increases the richness of our data distribution by approximating continuous time dynamics, thereby reducing generalization gaps and improving accuracy during inference.
>
> 4. **Ablation studies:**
> We have conducted a comprehensive ablation study to evaluate the core components of the proposed framework and to identify their individual contributions to overall performance. While we present the most critical results in the main text, additional ablations and supporting evidence are provided in the Supplementary Materials (SM). In Figure 3a and the accompanying discussion on page 7, we ablate the proposed *edge masking* strategy, highlighting its role in enhancing resolution invariance. We also briefly discuss its computational advantages. Further, in SM Section F.4, we demonstrate the broader applicability of edge masking by incorporating it into MeshGraphNet [1], where it similarly leads to a substantial reduction in the test error with unseen resolutions. In both figures, it is evident that edge masking also reduces the test error with the training resolution by facilitating more effective optimization. The impact of the fractional pairing strategy and various time-marching schemes is ablated in Figures 3b and 3c. We also ablate the regional aggregation mechanism in the encoder and decoder in SM Section F.2 (see SM Table F.2). These results confirm that regional aggregation is a key contributor to RIGNO's performance. Notably, the trend in decreasing test error breaks when regional aggregation is omitted (at 16,384 regional nodes, same as the number of input points), underscoring its necessity for accurate learning. Additionally, in SM Section F.3 (see SM Figure F.1), we analyze the sensitivity of RIGNO’s accuracy and inference time to four major hyperparameters. From this analysis, we observe that neither the overlap factors nor the number of regional mesh levels require tuning: minimal overlaps are recommended due to their limited benefit relative to their computational cost, while maximal mesh levels are encouraged given their accuracy gains with negligible overhead. Moreover, we find that the most effective way to scale RIGNO’s capacity is through increasing either the number of processor blocks or the latent dimensionality of node and edge features. Both of these options yield consistent improvements in performance. We believe that all essential aspects of the framework have been carefully studied and reported. Other hyperparameters listed in SM Table E.1 (e.g., the number of hidden layers in MLPs) are included for completeness, though we find they have minimal impact on model accuracy or generalization.
>
> 5. **Training/inference time benchmarks:**
> We benchmarked GINO, RIGNO, and MeshGraphNet [1] in terms of training and inference times across various model configurations. The results indicate that RIGNO’s training times are consistently slower than those of GINO on datasets with fixed geometry and consistent input coordinates across training samples. We acknowledge this as a limitation of our model; however, since training is a one-time cost, inference speed is ultimately more critical for deployment. In this regard, RIGNO and GINO exhibit comparable inference times, while RIGNO consistently achieves higher accuracy. Moreover, unlike GINO, RIGNO supports batch processing for datasets with variable geometries (e.g., Elasticity), where it is also faster at training. Due to space constraints, we kindly refer the reviewer to point 6 in our response to Reviewer Z5X2 for a broader discussion. We thank the reviewer for highlighting this point and will include these results, along with additional performance benchmarks, in a CRV if the paper is accepted.
>
> 6. **Memory usage benchmarks:**
> Following the reviewer’s suggestion, we evaluated the memory usage of RIGNO and the best-performing baseline, GINO, in terms of peak inference memory usage on the Elasticity and Poisson-Gauss datasets using a standard 24GB NVIDIA GPU. The results are summarized in the table below. For the Poisson-Gauss dataset, GINO requires approximately 370MB per sample, which limits the maximum batch size to 66. In contrast, RIGNO uses only 120MB per sample, allowing batch sizes up to 200. A similar trend is observed on the Elasticity dataset, where the smaller problem size permits even larger batches for both models. If the paper is accepted, we will gladly include these results, along with a comparison to additional baselines, in a CRV.
>
> ---
> >**Table 1: Peak memory usage of RIGNO and GINO on the unstructured Poisson-Gauss and Elasticity datasets with a NVIDIA GeForce RTX 3090 (24GB). The two best results are highlighted at each column.**
> >
> >||Parameters [M]|Poisson-Gauss|Elasticity|
> >|:---|:---:|:---:|:---:|
> >|GINO (3)|3.6|370.0MB|80.3MB|
> >|GINO (7) |8.4|406.1MB|116.4MB|
> >|RIGNO (96)|1.4|**120.9MB**|**50.6MB**|
> >|RIGNO (128)|2.5|**159.3MB**|**64.6MB**|
> ---
>
> We hope to have addressed all of the reviewer’s comments and concerns in this response, and we kindly ask the reviewer to consider updating their assessment in light of these clarifications.
>
> *[1] Tobias Pfaff et al., Learning mesh-based simulation with graph networks. NeurIPS (2020).*

---

> > ### Comment · Reviewer_9Nmc · 2025-08-04
> >
> > Thanks the author for the response. If I understand correctly, one of the major advantage of RIGNO is its local interation via graphs, which helps it outperforms global models such as Fourier and Attention based model. These global models may be better on elliptic problems such as Heat, Elasticity, and Poisson-Gauss equation. FNO and SCot could be better than RIGNO on these elliptic problems, at least on Cartesian grid. In this case, it might be helpful to address this in the limitation section, and add the experiments of elliptic PDEs on Cartesian grid. Currently, in Table 1, elliptic PDEs only appear with point clouds, not with Cartesian grid, which may feel like cherry-picked. I believe it is totally fine even if RIGNO is not the best on all the PDEs in table 1. The contribution of the paper is still significant enough.

---

> ### Author Response · Authors · 2025-08-05
> **Elliptic problems on Cartesian grids**
>
> We thank the reviewer for appreciating our rebuttal and for acknowledging the significance of our contributions.
>
> Regarding the performance on elliptic problems, the reviewer is correct in noting that these are not RIGNO’s primary strength. However, our results on non-Cartesian grids demonstrate that the long-range reinforcements in RIGNO enable it to **match or even surpass** the performance of global-interaction models, including Fourier-based approaches (GINO, FNO-DSE, GeoFNO) and attention-based models (UPT), on such tasks. Specifically, the Heat-L-Sines and Elasticity datasets are defined on irregular domains (L-shape and square with a hole, respectively), and as such cannot be evaluated using methods like CNO, scOT, or FNO in their original form. However, we were able to evaluate these baselines on the Poisson-Gauss dataset, which uses a Cartesian grid. While RIGNO does not outperform scOT in this setting, it still achieves higher accuracy than both CNO and FNO, suggesting its competitive performance even on structured grids.
>
> Given that RIGNO is the second-best model on the Poisson-Gauss dataset (Cartesian) and outperforms strong baselines such as FNO and CNO, we do not consider this to be a major limitation of our architecture. That said, we agree with the reviewer that this aspect deserves explicit discussion in the main text, and we appreciate the suggestion. We will gladly incorporate a dedicated discussion and complementary results on elliptic PDEs over Cartesian grids into Table 1 in a camera-ready version, if accepted.
>
> We once again thank the reviewer for their thoughtful comments and hope that this addresses their concerns satisfactorily. We remain at your disposal to answer any further questions about our paper.

---

> > ### Comment · Reviewer_9Nmc · 2025-08-08
> >
> > Thanks the author for the response. I find the information regarding the runtime very helpful. I will keep my original score of 5 -- acceptance. I hope the review process is helpful and constructive.

---

> > > ### Author Response · Authors · 2025-08-08
> > >
> > > We sincerely thank the reviewer for appreciating our response and recommending our paper for acceptance. Your constructive feedback is greatly appreciated and will help us further improve the quality of our work.

---

### Official Review · Reviewer_Z5X2 · 2025-07-03

**Clarity:** 2
**Significance:** 2
**Originality:** 3
**Rating:** 4
**Confidence:** 4

**Summary:**

The authors propose RIGNO (Region Interacting Graph Neural Operator), an end-to-end graph neural network for training the solution operators of PDEs on arbitrary domains. The main motivation for the article is that many existing Neural Operators (NOs) assume that the underlying domain of the PDEs is discretized on a regular grid. The results demonstrate that RIGNO outperforms the baselines on both unstructured and structured grids.

**Questions:**

See the weaknesses

**Ethical Concerns:**

["NO or VERY MINOR ethics concerns only"]

**Final Justification:**

I get answer on my question. It helped me to better understand the paper.

**Limitations:**

The authors discussed the method limitations.

**Quality:**

2

**Strengths And Weaknesses:**

**Strengths**
* RIGNO demonstrate better performance compare with state-of-art baselines across a wide and challenging range of benchmarks.
* The paper introduces several novel techniques:
	* **edge masking**: the core idea is to randomly disable, or "mask," a fraction of the edges in the graph before each message-passing step. This process is dynamic, with a different random set of edges being masked at every step and epoch.
	* **temporal fractional pairing** is a novel fine-tuning strategy designed to improve the generalization of time-continuous neural operators to time steps smaller than those seen during training.
* The authors are also write about the limitations of their approach.
* The introduction and related work sections clearly formulated the gap in the literature (the need for a multi-scale, resolution-invariant GNN operator) and position RIGNO as the solution.


**Weaknesses**
* The paper is purely empirical. It lacks theoretical justification for its key components:
	* Are there theoretical guarantees on why edge masking improve resolution invariance? It is presented only as an empirical result.
* RIGNO contains a lot of hyperparameters. The article does not provide a sensitivity analysis for these hyperparameters. This complexity could make the model difficult to tune and apply for new problems, potentially hindering its practical application.
* For a method to be a practical surrogate solver, its speed is as important as its accuracy. Therefore, it would be beneficial to include information in the article about the training and inference times for both RIGNO and the baselines.

---

> ### Author Rebuttal · Authors · 2025-07-30
>
> We begin by thanking the reviewer for their thoughtful comments and constructive suggestions. We address their concerns in detail below.
>
> 1. **Justification for edge masking:**
> Although we have not provided a formal theoretical analysis of edge masking in our manuscript, we have a clear and intuitive idea of how it contributes to resolution invariance. Considering the graph structure in the encoder, a regional node receives information from multiple physical nodes. With high-resolution inputs, many of these nodes are *redundant* in the sense that the underlying function can often be accurately reconstructed (e.g., via interpolated) using fewer nodes. During training, the model may implicitly ignore some of these redundant nodes and focus on a subset of dominant or central connections. At test time with different resolutions, these dominant nodes may no longer be present, leading to degraded performance. We identify this phenomenon as a key contributor to the resolution sensitivity observed in many GNN-based operator learning models and address it with the proposed method. By randomly masking edges during training, the model cannot consistently rely on the same high-correlation connections. In the absence of these edges, the model is forced to use the weaker correlations. This principle extends naturally to both the processor and decoder stages. Additionally, learning from weaker or secondary correlations in the processor enables the model to better integrate information across multiple spatial scales, rather than overfitting to dominant patterns in the training resolution.
>
> 2. **Justification for fractional pairing:**
> Similarly, the effectiveness of the fractional pairing strategy can be understood through a simple observation. In conventional training setups (including all2all pairing), the model is only exposed to lead times that are integer multiples of the dataset’s time resolution Dt. As a result, any smaller lead time (e.g., Dt/2) is out-of-distribution (OOD) for the model. During inference, this distribution shift can lead to noticeable performance degradation when the model is queried on shorter, unseen lead times. Our proposed fractional pairing fine-tuning strategy directly addresses this issue by explicitly extending the distribution of lead times seen during training. In simple terms, it increases the richness of our data distribution by approximating continuous time dynamics, thereby reducing generalization gaps and improving accuracy during inference.
>
> 3. **Theoretical analysis:**
> Although mathematical formalization is lacking for much of the neural operator community, it is more straightforward to formalize the intuition behind edge masking. One approach to construct a formal definition could view edge masking as a Bayesian regularization on the input/output space of a GNN layer and follow a similar strategy as for dropout [1]. We thank the reviewer for this thoughtful suggestion and recognize it as a valuable direction for future research. We also appreciate the reviewer’s observation regarding the lack of detailed justification of the effectiveness of these methods, and we will gladly include a more comprehensive discussion in the camera-ready version (CRV), if the paper is accepted. However, providing rigorous theoretical explanations or guarantees would require significant additional effort and is beyond the scope of the current work.
>
> 4. **Many hyperparameters:**
> Apart from standard optimization hyperparameters common to most machine learning frameworks, the size and performance of RIGNO are primarily sensitive to just **two key architectural hyperparameters**: the number of processor blocks and the latent dimension of edge/node features. In fact, one of the core design principles of our framework is to minimize the need for extensive hyperparameter tuning by using **a fixed set of hyperparameters across all tasks**. To this end, we identified a robust configuration by tuning on a single internal dataset and applied the same hyperparameters across all other datasets without further adjustment. As shown in Table 1 of the submission, both the proposed RIGNO-18 and RIGNO-12 architectures deliver consistently strong performance across diverse benchmarks, in contrast to many of the baselines that require task-specific tuning. The complete list of hyperparameters is provided in SM Table E.1, and this configuration can be readily applied to new problems or datasets. Note that the only difference between RIGNO-18 and RIGNO-12 lies in the number of processor blocks. While we report all hyperparameters for transparency, most of them do not require tuning. If the paper is accepted, we will clearly emphasize this point in the CRV.
>
> 5. **Sensitivity to hyperparameters:**
> We have analyzed the sensitivity of RIGNO with respect to both accuracy and inference time across four hyperparameters in SM Section F.3 (see SM Figure F.1). In particular, we demonstrate that overlap factors and the number of regional mesh levels do not require tuning. Based on our analysis, we recommend using minimal overlaps, as they offer limited accuracy gains relative to their computational cost, and maximal regional mesh levels, which provide substantial accuracy improvements with minimal added cost. The number of regional nodes is another important hyperparameter, and is studied in SM Section F.2 (see SM Table F.2). Our results show that while using a coarser regional mesh can improve efficiency, it reduces model capacity. We find that using approximately 4000 regional nodes provides a good trade-off between accuracy and efficiency. While the optimal number may vary depending on the target function, this configuration has consistently yielded strong results across all datasets.
>
> 6. **Performance benchmarks:**
> Below, we provide two tables reporting the training and inference times for GINO, RIGNO, and MeshGraphNet (MGN) [1] across various model configurations, all evaluated on a single NVIDIA GeForce RTX 3090 GPU (24GB). GINO is included due to its consistently accurate performance across benchmarks, while MGN is selected for its architectural similarity to RIGNO. Owing to the limited time available for evaluation and the consistent accuracy advantages of RIGNO over MGN (see Table 1 in the main text), we report test errors only for RIGNO and GINO in the tables below. For training, we use the maximum possible batch size for RIGNO and MGN, and a batch size of 64 for GINO. In scenarios involving varying geometries across training samples, such as in the Elasticity experiment, the original implementation of GINO does not support batch processing, requiring us to use a reduced batch size of 1. We employ 18 message-passing blocks in the processor for both RIGNO and MGN, and evaluate configurations with 96 and 128 latent features. For GINO, we report results for two variants consisting of 3 and 7 FNO layers, respectively. The results show that the training times of RIGNO are slower than those of GINO for datasets where the geometry and input coordinates are fixed between training samples, and are faster for datasets with variable geometries, mainly due to the support of batch processing. We acknowledge the former case is a disadvantage of the proposed model; however, as training time is a fixed initial cost, the difference in speed is more critical at inference time. In this setting, RIGNO and GINO have very similar performance, yet RIGNO is still able to significantly surpass GINO's performance in terms of accuracy. While this subset of experiments is necessarily limited due to the time constraints of the rebuttal period, we believe the reported metrics are representative and consistent across settings. Admittedly, longer training times are a drawback; however, we view them as a worthwhile tradeoff for the consistent gains in accuracy and robustness achieved across a diverse range of physical systems with RIGNO over GINO (Main Text Table 1). Importantly, inference times, which are far more relevant for practical deployment, remain comparable across all models considered in this rebuttal and in the main text (Table 1). Moreover, our benchmarks show that RIGNO is up to 3x more memory-efficient than GINO. Please see point 6 in our response to Reviewer 9Nmc. Thanks to its multi-scale architecture and computational efficiency, RIGNO emerges as a powerful end-to-end GNN that achieves state-of-the-art accuracy and robustness, while remaining competitive with strong non-GNN baselines in terms of speed and memory usage. We believe this work reinforces the viability of GNNs for solving PDEs and opens the door to future developments aimed at further reducing training costs. We thank the reviewer for raising this point and will present these results along with complementary performance benchmarks in a CRV, if accepted.
>
> ---
> >**Table 1: Performance benchmarks on the unstructured Poisson-Gauss. RIGNO+ corresponds a subsample factor of 16.0, resulting in only 576 regional nodes.**
> >
> >||Parameters|Training time per epoch [s]|Test error [%]|Inference time [ms]|
> >|:---|:---:|:---:|:---:|:---:|
> >|GINO (3)|3.6|**1.33**|5.24|7.8|
> >|GINO (7)|8.4|**1.72**|8.01|10.1|
> >|MGN (96)|1.4|43.3|n/a|16.1|
> >|MGN (128)|2.5|53.1|n/a|25.5|
> >|RIGNO (96)|1.4|24.9|**2.23**|8.1|
> >|RIGNO (128)|2.5|28.4|**2.09**|12.7|
> >|RIGNO+ (96)|1.4|15.5|4.19|**5.2**|
> >|RIGNO+ (128)|2.5|17.9|3.25|**7.2**|
>
> >**Table 2: Performance benchmarks on the Elasticity dataset.**
> >
> >||Parameters|Training time per epoch [s]|Inference time [ms]|
> >|:---|:---:|:---:|:---:|
> >|GINO (3)|3.6|18.2|4.1|
> >|GINO (7) |8.4|27.1|6.4|
> >|RIGNO (96)|1.4|**2.4**|**1.5**|
> >|RIGNO (128)|2.5|**3.1**|**1.9**|
> ---
>
> We hope that our responses have satisfactorily addressed all of the reviewer’s comments and concerns, and we kindly ask the reviewer to consider updating their assessment in light of these clarifications.
>
> *[1] Pierre Baldi and Peter Sadowski, Understanding dropout. NeurIPS (2013).*

---

> > ### Comment · Reviewer_Z5X2 · 2025-08-05
> >
> > Thank you for your comprehensive response. I appreciate the answers to my questions. I will increase my score.

---

### Official Review · Reviewer_1KFB · 2025-07-16

**Clarity:** 4
**Significance:** 4
**Originality:** 4
**Rating:** 4
**Confidence:** 5

**Summary:**

This paper proposes  a novel graph-based framework for learning solution operators of PDEs on arbitrary, non-Cartesian domains. RIGNO uses an end-to-endGNN with a hierarchical "regional mesh" to achieve space-time resolution invariance and handle irregular point clouds. The framework introduces key innovations including regional mesh construction, edge masking, and temporal fractional pairing for robust and generalizable operator learning. Experiments on 13 unstructured and 8 Cartesian-grid PDE datasets show that RIGNO significantly outperforms prior methods (e.g., MeshGraphNet, FNO, CNO) in both accuracy and generalization.

**Questions:**

None.

**Ethical Concerns:**

["NO or VERY MINOR ethics concerns only"]

**Limitations:**

None.

**Paper Formatting Concerns:**

None.

**Quality:**

4

**Strengths And Weaknesses:**

**Strengths**
1. The paper is novel to the best of my knowledge. It introduces a unique combination of graph neural operators and multi-scale regional meshes to handle non-Cartesian geometries with space-time resolution invariance, which is not sufficiently addressed in prior work.

2. Writing is clear and well-structured.

**Weakness**
1. The current implementation focuses only on 2D domain. No empirical evidence or 3D experiments are shown.
2. Fig 2 is small and not clear.
3. Table1 looks strange. I believe the authors could find more baselines to fill the table.

---

> ### Author Rebuttal · Authors · 2025-07-30
>
> We begin by thanking the reviewer for their thoughtful comments and constructive suggestions. We address their concerns in detail below.
>
> 1. **Extension to 3D:**
> As the reviewer rightly points out, the current work does not include empirical results on 3D problems. However, we would like to emphasize that all components of the proposed framework are readily extendable to 3D settings, and we do not anticipate any significant technical hurdles or computational limitations in doing so. In the graph construction stage (see Section "Underlying Graphs" on page 3 and SM Section A), all key concepts naturally generalize to 3D point clouds. The encoder and decoder edges defined in 2D using a support radius can be similarly defined in 3D, where the support sub-regions become spherical instead of circular. The processor edges and the support radii of the regional nodes can be constructed using tetrahedral meshes in 3D in place of 2D triangular meshes. Importantly, once the 3D input graphs are constructed, the underlying GNN architecture remains unchanged, requiring no architectural modifications to operate on 3D data.
>
> 2. **Scalability:**
> On the other hand, the 2D datasets used in our study are large-scale, with most containing 1K trajectories and over 28K input-output pairs, each defined on 16K spatial points. This level of scale significantly exceeds that of typical 2D datasets in the literature and is comparable to some 3D datasets, which typically include fewer than 1K input-output pairs at spatial resolutions ranging from 10K to 40K points. We believe that RIGNO’s strong performance on these challenging 2D benchmarks demonstrates both its scalability and its potential for extension to 3D problems. To further support this claim, we highlight the computational advantages of RIGNO over MeshGraphNet [1]. Notably, GNNs have already been employed successfully in 3D problems using MeshGraphNet and have been scaled to even larger problems as shown in [2]. To quantify this, we benchmarked the computational cost of RIGNO-18 against MeshGraphNet with 18 message-passing blocks on the Poisson-Gauss dataset (see table below). While MeshGraphNet saturates the memory of an NVIDIA GeForce RTX 3090 (24GB) with a batch size of 8, RIGNO (with 2.3K regional nodes) requires only half the memory, supporting a batch size of 16. When using fewer regional nodes (~0.5K), the batch size can be further increased to 32. Moreover, training and inference times are reduced to one-third compared to MeshGraphNet. These computational advantages do not come at the cost of performance. On the contrary, they lead to significant improvements, as evidenced by the results in Table 1 of the paper and SM Table F.2. Across many datasets, RIGNO achieves up to an order of magnitude reduction in test error compared to MeshGraphNet with similar configurations. Given that MeshGraphNet has been successfully tested on 3D problems, RIGNO's reduced computational/memory costs further support our claim that RIGNO can readily be extended to 3D problems. We aim to do so in future work.
>
> ---
> > **Table 1: Computational cost of RIGNO and MeshGraphNet (MGN) with 18 message passing steps on the unstructured Poisson-Gauss dataset with 9216 input points. All experiments are done on a single NVIDIA GeForce 3090 (24GB). Training times are measured with 1024 training samples using the maximum batch size, and inference time corresponds to a single sample. RIGNO+ corresponds to a subsample factor of 16.0, resulting in only 576 regional nodes.**
> >
> > |  | Maximum batch size | Training time per epoch [s] | Inference time [ms] |
> > |:---:|---:|---:|---:|
> > | MGN | 8  | 53.1 | 25.5 |
> > | RIGNO | 16 | 28.4 | 12.7 |
> > | RIGNO+  | 32 | 17.9 | 7.2  |
>
> ---
>
>
> 6. **Further performance benchmarks:**
> We benchmarked RIGNO and two baselines in terms of memory usage, training time, and inference time across various model configurations. The results indicate that RIGNO is consistently more memory-efficient than the baselines, while yielding higher accuracies and comparable inference times. We kindly refer the reviewer to point 6 in our response to Reviewer Z5X2 for a broader discussion regarding training and inference times, and point 6 in our response to Reviewer 9Nmc regarding memory usage. We will include these results, along with additional performance benchmarks, in a camera-ready version (CRV) if the paper is accepted.
>
> 3. **Formatting issues:**
> We thank the reviewer for their constructive suggestions regarding Figure 2 and Table 1, and will immediately correct these issues for the CRV, if accepted. Regarding Table 1, since the main advantage of the proposed architecture is for arbitrary domains, we have included many baselines that work on point clouds and only a few that work on uniform Cartesian grids. The extra page in a CRV allows us to simply separate the tables, especially since we plan to add Transolver as a baseline following the suggestion of reviewer AxpQ.
>
> We hope that our responses have satisfactorily addressed all of the reviewer’s comments and concerns, and we kindly ask the reviewer to consider updating their assessment in light of these clarifications.
>
> *[1] Tobias Pfaff et al., Learning mesh-based simulation with graph networks. NeurIPS (2020).*
>
> *[2] Remi Lam et al., Learning skillful medium-range global weather forecasting. Science 382, 1416-1421 (2023).*

---

> ### Comment · Area_Chair_jwJB · 2025-08-06
> **Urgent request**
>
> Hi Reviewer 1KFB,
>
> You have not yet responded to the 'Flag Insufficient Review' request, nor have you provided a response to the author rebuttal.
>
> I kindly ask that you address both items as soon as possible.
>
> If you are unable to do so before the end of the discussion period, please note that your review and rating may not be considered in the final decision.
>
> Thanks,
>
> AC

---

### Decision · Program_Chairs · 2025-09-17

**Decision:**

Accept (poster)

**Comment:**

This paper presents RIGNO, a graph-based framework for robust and accurate operator learning of PDEs on arbitrary domains. The reviewers initially highlighted concerns regarding clarity of the methodology, insufficient discussion of efficiency and scaling, and the need for stronger baselines. While they found the core idea promising, they felt that these issues limited the paper’s impact in its original form.

The authors’ rebuttal directly addressed these points. They provided detailed clarifications on the role of fractional pairing and other design choices, added comparisons against additional baselines, and included new efficiency benchmarks that demonstrated competitive training and inference performance. Reviewers acknowledged that these responses resolved their questions, improved their understanding of the method, and in some cases led to higher confidence in their evaluations.

Overall, the paper now presents a clear, well-supported framework with strong experimental evidence and thorough justification of design choices. With the reviewers’ concerns adequately resolved in the rebuttal, I recommend acceptance.